# Characterizing Evolution in Expectation-Maximization Estimates for Overspecified Mixed Linear Regression

**Zhankun Luo**                                                      *luo333@purdue.edu*
**Abolfazl Hashemi**                                                 *abolfazl@purdue.edu*
*School of Electrical and Computer Engineering*
*Purdue University, West Lafayette, IN, USA*

**Reviewed on OpenReview:** *https://openreview.net/forum?id=mFdHMNFtrT*

## Abstract

Estimating data distributions using parametric families is crucial in many learning setups, serving both as a standalone problem and an intermediate objective for downstream tasks. Mixture models, in particular, have attracted significant attention due to their practical effectiveness and comprehensive theoretical foundations. A persisting challenge is model misspecification, which occurs when the model to be fitted has more mixture components than those in the data distribution. In this paper, we develop a theoretical understanding of the Expectation-Maximization (EM) algorithm's behavior in the context of targeted model misspecification for overspecified two-component Mixed Linear Regression (2MLR) with unknown $d$-dimensional regression parameters and mixing weights. In Theorem 5.1 at the population level, with an unbalanced initial guess for mixing weights, we establish linear convergence of regression parameters in $\mathcal{O}(\log(1/\epsilon))$ steps. Conversely, with a balanced initial guess for mixing weights, we observe sublinear convergence in $\mathcal{O}(\epsilon^{-2})$ steps to achieve the $\epsilon$-accuracy at Euclidean distance. In Theorem 6.1 at the finite-sample level, for mixtures with sufficiently unbalanced fixed mixing weights, we demonstrate a statistical accuracy of $\mathcal{O}((d/n)^{1/2})$, whereas for those with sufficiently balanced fixed mixing weights, the accuracy is $\mathcal{O}((d/n)^{1/4})$ given $n$ data samples. Furthermore, we underscore the connection between our population level and finite-sample level results: by setting the desired final accuracy $\epsilon$ in Theorem 5.1 to match that in Theorem 6.1 at the finite-sample level, namely letting $\epsilon = \mathcal{O}((d/n)^{1/2})$ for sufficiently unbalanced fixed mixing weights and $\epsilon = \mathcal{O}((d/n)^{1/4})$ for sufficiently balanced fixed mixing weights, we intuitively derive iteration complexity bounds $\mathcal{O}(\log(1/\epsilon)) = \mathcal{O}(\log(n/d))$ and $\mathcal{O}(\epsilon^{-2}) = \mathcal{O}((n/d)^{1/2})$ at the finite-sample level for sufficiently unbalanced and balanced initial mixing weights, respectively. We further extend our analysis in the overspecified setting to the finite low SNR regime, providing approximate dynamic equations that characterize the EM algorithm's behavior in this challenging case. Our new findings not only expand the scope of theoretical convergence but also improve the bounds for statistical error, time complexity, and sample complexity, and rigorously characterize the evolution of EM estimates.

## 1 Introduction

Mixtures of parameterized models are powerful tools to model intricate relationships in practical scenarios, such as Mixed Linear Regression (MLR) and Gaussian Mixture Models (GMM) (Beale & Little, 1975). It is common for the number of mixture components in the fitted model to differ from that of the data distribution, which can substantially slow down parameter estimation convergence rates. We specifically focus on the overspecified setting, where the number of mixture components in the fitted model exceeds that of the data distribution. Expectation Maximization (EM) algorithm (Dempster et al., 1977; Wu, 1983; De Veaux, 1989; Jordan & Xu, 1995; Wedel & DeSarbo, 1995) is notable for its computational efficiency and ease of practical implementation, overcoming the intractable problem of non-convexity and the presence of numerous spurious

local maxima in Maximum Likelihood Estimation (MLE) (Qian et al., 2022). It proceeds through two steps: in the E-step, it computes the expected log-likelihood using the current parameter estimate; subsequently, the M-step updates the parameters to maximize the expected log-likelihood computed in the E-step. These iterative steps serve to maximize the lower bound on MLE until convergence. The goal of this paper is to gain a comprehensive understanding of EM updates for overspecified mixture models.

## 1.1 Related Work

It has been shown that EM achieves global convergence for GMM with $k = 2$ components (2GMM) (Klusowski & Brinda, 2016; Xu et al., 2016; Daskalakis et al., 2017; Ndaoud, 2018; Zhao et al., 2020; Wu & Zhou, 2021). Studies (Balakrishnan et al., 2017; Klusowski & Brinda, 2016; Kwon et al., 2019; 2022) have confirmed that EM for two-component mixed linear regression (2MLR) converges from a random initialization with high probability. GMMs with $k \geq 3$ components frequently lead to EM with random initialization being trapped in local minima with high probability, while local maxima can demonstrate notably inferior likelihood than any global maximum (Jin et al., 2016). For mixture models with multiple components, such as Gaussian Mixture Models (GMM), it is well known that the negative log-likelihood—used as the objective function in the EM algorithm—can exhibit several spurious local minima that are not globally optimal, even when the mixture components are well separated (Chen et al., 2024b). Consequently, proper initialization of the parameters is crucial, as assumed in the analysis of mixtures of many linear regressions (Kwon & Caramanis, 2020). A convergence analysis for EM in MLR with multiple components has been presented (Kwon & Caramanis, 2020), but it requires careful initialization and strong separation for regression parameters. Even for the case of 2GMM, most work requires good initialization and strong separation for regression parameters. For instance, Wu & Zhou (2021) analyzed the convergence rates of 2GMM with fixed balanced mixing weights given a specific initialization and strong separation for location parameters. Alternatively, specific initialization needs to trend toward the ground truth in cases of weak separation or no separation (Weinberger & Bresler, 2022). There is a lack of understanding of the EM update rules in mixture models given weak separation or no separation of the ground truth parameters.

When ground truth parameters of some components in mixture models have no separation, this means we are using a model with more components to fit a ground truth data distribution with fewer components, which is called the overspecified setting. In particular, the 2GMM/2MLR model is considered overspecified when there is no separation between the true regression parameters or location parameters of its two components. Moreover, it has been observed that the convergence of EM can be prohibitively slow when dealing with poorly separated mixtures (Redner & Walker, 1984), while achieving rapid linear convergence or superlinear convergence in cases with strong separation (Kwon et al., 2021; Ghosh & Kannan, 2020). The convergence rate of regression parameters using EM for 2GMM in the overspecified setting, assuming known mixing weights, is analyzed in Dwivedi et al. (2020b); Dhawan et al. (2023). Kwon et al. (2019; 2021) derived iteration complexity and statistical accuracy for the known-weight case across all SNR regimes—but their results in the low-SNR setting apply only when mixing weights are known to be balanced. Kwon & Caramanis (2020) analyzed EM for MLR with more than two components under restrictive conditions—such as requiring initializations very near the ground truth and well-separated regression parameters—which do not cover cases of overspecification where the ground truth parameters coincide. Nevertheless, there is a lack of prior research formally analyzing the evolution of EM updates for 2MLR with unknown regression parameters and mixing weights in the overspecified setting, with most prior research primarily concentrating on scenarios with known mixing weights. The behavior of EM updates in overspecified settings is not fully understood when dealing with unknown mixing weights, whether they are balanced or unbalanced. In this paper, we address this gap by conducting a detailed analysis of the EM updates for overspecified 2MLR, based on the expressions of integrals with Bessel functions given in (Luo & Hashemi, 2024). Even though Luo & Hashemi (2024) provided closed-form EM update rules (using Bessel functions) for 2MLR across all SNR regimes, but they focus primarily on the high-SNR case and do not discuss convergence guarantees or the dynamics in the overspecified setting and the low-SNR regime. The statistical rates for parameter estimation of a class of overspecified finite mixture models have been studied in several studies (Chen, 1995; Ho & Nguyen, 2016; Heinrich & Kahn, 2018; Doss et al., 2020; Manole & Ho, 2020; 2022; Ho et al., 2022b; Nguyen et al., 2023a). The optimal minimax rates for 2MLR with known balanced mixing weights have been determined in Kwon et al. (2021), while those for 2GMM with known mixing weights have been established in Dwivedi et al.

(2020b) using the scheme outlined by Balakrishnan et al. (2017). However, the statistical error for 2MLR with unknown mixing weights remains unclear. We develop new techniques to sharpen the analysis for this statistical error.

## 1.2 Technical Overview

In this paper, we present a comprehensive study of overspecified mixture models. The paper is organized as follows. In section 2, we state the setting of two-component mixed linear regression (2MLR), overspecification, and the EM update rules, and introduce Bessel functions for the analysis of EM updates. In section 3, we provide two motivating examples (haplotype assembly and phase retrieval) of the 2MLR model to justify the practical implications. In section 4, we characterize the population EM update rules with expectations under the density involving Bessel functions (equation 10), show their nonincreasing property and boundedness (Facts 4.2, 4.3), and provide their approximate dynamic equations (Proposition 4.4) for the evolution of regression parameters and mixing weights. In section 5, we establish the guarantees of convergence rate for population EM (Theorem 5.1) with balanced and unbalanced initial guesses for mixing weights, and establish theoretical bounds for sublinear convergence (Proposition 5.3) with balanced initial guesses and the contraction factor of linear convergence (Proposition 5.4) with unbalanced initial guesses. In section 6, we present the tight bounds for sample complexity, time complexity, and final accuracy for finite-sample EM (Theorem 6.1) by coupling the analysis of population EM and finite-sample EM and establishing statistical errors (Propositions 6.3, 6.4) in the overspecified setting. In section 7, we discuss the differences between results of 2MLR and 2GMM, extend our analysis from the overspecified setting to the low-SNR regime, and examine the challenges of analyzing overspecified mixture models with multiple components. In section 8, we validate our theoretical findings with numerical experiments. Detailed derivations and proof techniques are provided in the Appendices. In summary, our contributions include:

- We derive the approximate dynamic equations for the regression parameters and mixing weights in Proposition 4.4 based on the population EM update rules to disentangle the relationships between regression parameters and mixing weights by establishing novel inequalities and identities for EM update rules (see equation 10 and Appendices A, H), aiding in the investigation of EM evolution in the overspecified setting and the low-SNR regime.

- Linear convergence of regression parameters with a rate of $\mathcal{O}(\log(1/\epsilon))$ is achieved with an unbalanced initial guess for mixing weights. In contrast, sublinear convergence at a rate of $\mathcal{O}(\epsilon^{-2})$ is confirmed with a balanced initial guess by replacing annulus-based localization in Dwivedi et al. (2020b) with a "variable separation" method upon the discretized differential inequality (see the proofs of Proposition 5.3 and Theorem 6.1 in Appendices D, G respectively), ensuring $\epsilon$-accuracy in Euclidean distance, as shown in Theorem 5.1 and Proposition 5.3.

- We address the gap for sufficiently balanced mixtures, where the imbalance of the initial guess for mixing weights is less than or of the same order of magnitude as $(d/n)^{1/4}$ in Theorem 6.1, improving bounds on statistical error, time complexity, and sample complexity in 2MLR by establishing the concentration inequality based on modified log-Sobolev inequality (see Section 5.3 modified logarithmic Sobolev inequalities in Ledoux (2001)) and bounds for the statistics (see Appendices E, F). Our new techniques advance the results beyond those for 2GMM (Dwivedi et al., 2020b).

By adopting the above novel techniques, we rigorously characterize the evolution of EM estimates for both regression parameters and mixing weights of overspecified MLR models by providing approximate dynamic equations (Proposition 4.4) for EM update rules and establishing convergence guarantees (Theorems 5.1, 6.1) for final accuracy, time complexity, and sample complexity at population and finite-sample levels, respectively.

## 2 Problem Setup

In this paper, we investigate the symmetric two-component mixed linear regression (2MLR) model given by:

$$y = (-1)^{z+1}\langle\theta^*, x\rangle + \varepsilon, \tag{1}$$

where $\varepsilon$ denotes the additive Gaussian noise, $s = (x, y) \in \mathbb{R}^d \times \mathbb{R}$ represents the covariate-response observation, $z \in \{1, 2\}$ is the latent variable, i.e., the label of the data, with probabilities $\mathbb{P}[z = 1] = \pi^*(1)$ and $\mathbb{P}[z = 2] = \pi^*(2)$. The ground truth values for the regression parameters and the mixing weights are expressed by $\theta^*$ and $\pi^* = (\pi^*(1), \pi^*(2))$, respectively. $\theta$ and $\pi = (\pi(1), \pi(2))$ denote the estimated values for the regression parameters and mixing weights in the fitted 2MLR model, respectively.

Consider the 2MLR model in equation 1. Let $n$ denote the number of samples $\mathcal{S} := \{(x_i, y_i)\}_{i=1}^n$ used for each EM update, and let $\{z_i\}_{i=1}^n$ be the values of the latent variable for these samples. Furthermore, $\sigma^2$ denotes the noise variance, $\bar{\theta} := \theta/\sigma$ and $\bar{\theta}^* := \theta^*/\sigma$ are the normalized parameters, and $\eta := \|\theta^*\|/\sigma$ is the signal-to-noise ratio (SNR), while $\rho := \langle \theta^*, \theta \rangle / (\|\theta^*\| \cdot \|\theta\|)$ is the cosine angle between the ground truth and the estimated regression parameters. In this paper, we focus on the case of overspecification, namely $\theta^* = \vec{0} = (0, 0, \cdots, 0) \in \mathbb{R}^d$, where the ground truth regression parameters are zero, and there is no separation between two mixtures. We note that the 2MLR and 2GMM models are standard models for establishing theoretical understanding of methods for mixture models, see, e.g., the recent works (Dwivedi et al., 2020b; Weinberger & Bresler, 2022; Dhawan et al., 2023; Luo & Hashemi, 2024; Reisizadeh et al., 2024). Thus, we adopt the 2MLR model and its standard assumptions in this paper to develop a finer understanding of the misspecification phenomenon (see also Section 2.2.2, page 6 of Balakrishnan et al. (2017); page 1 of Klusowski & Brinda (2016); page 4 of Reisizadeh et al. (2024); and page 3 of Luo & Hashemi (2024)).

**Bessel Function.** $K_0(x)$, for all $x > 0$, is the modified Bessel function of the second kind with parameter 0, defined by the integral representation $K_0(x) := \int_0^\infty \exp(-x \cosh t) \, dt$ which is also the solution $f = K_0(x)$ to the modified Bessel equation $x^2 \frac{d^2 f}{dx^2} + x \frac{df}{dx} - x^2 f = 0$. The approximations for $K_0(x)$ are: $K_0(x) \approx \ln \frac{2}{x} - \gamma$ for $x \to 0_+$, where $\gamma \approx 0.577$ is Euler's constant, and $K_0(x) \approx \sqrt{\frac{\pi}{2x}} \exp(-x)$ for $x \to +\infty$. (see equation 10.32.9 for the integral representation, equation 10.25.1 for the modified Bessel equation, and equations 10.25.2, 10.25.3, and 10.31.2 for approximations in Chapter 10 of Olver et al. (2010)). An important fact is that for the product of two independent standard Gaussian random variables $Z_1 \sim \mathcal{N}(0, 1)$ and $Z_2 \sim \mathcal{N}(0, 1)$, it has the probability density $f_X$ involving the Bessel function $K_0$, $X := Z_1 \times Z_2 \sim f_X(x) = \frac{K_0(|x|)}{\pi}$ (see page 50, Section 4.4 Bessel Function Distributions, Chapter 12 Continuous Distributions (General) of Johnson et al. (1970)).

**Notations.** The notations $\Omega(\cdot)$, $\mathcal{O}(\cdot)$ and $\Theta(\cdot)$ share the standard definitions of asymptotic notations: $f = \Omega(g)$ means $g = \mathcal{O}(f)$ for $f, g$, namely $|f(x)| \geq C \times |g(x)|$ for some constant $C > 0$ and all $x$ sufficiently large (see page 528 of Lehman (2017)). $f = \Theta(g)$ means $f = \mathcal{O}(g)$ and $g = \mathcal{O}(f)$. We also write $f \lesssim g$ if $f = \mathcal{O}(g)$, and $f \gtrsim g$ if $f = \Omega(g)$, and $f \asymp g$ if $f = \Theta(g)$. $a \vee b$ and $a \wedge b$ refer to the least upper bound $\max(a, b)$ and the greatest lower bound $\min(a, b)$ of $a, b$, respectively.

## 2.1 EM Updates

The EM algorithm estimates the regression parameters and the mixing weights from observations. Balakrishnan et al. (2017) gave the population EM update rule of regression parameters for 2MLR given the known balanced mixing weights $\pi = \pi^* = (\frac{1}{2}, \frac{1}{2})$ as follows:

$$M(\theta) = \mathbb{E}_{s \sim p(s|\theta^*, \pi^*)} \tanh\left(\frac{y\langle x, \theta \rangle}{\sigma^2}\right) yx.$$

For the more general case of unknown mixing weights $\pi$, we introduce the variable $\nu$ to characterize the imbalance $\tanh \nu = \pi(1) - \pi(2)$ of the mixing weights $\pi = (\pi(1), \pi(2))$, namely

$$\nu := \frac{\ln \pi(1) - \ln \pi(2)}{2}, \tag{2}$$

and the population EM update rule for regression parameters $\theta$ becomes

$$M(\theta, \nu) := \mathbb{E}_{s \sim p(s|\theta^*, \pi^*)} \tanh\left(\frac{y\langle x, \theta \rangle}{\sigma^2} + \nu\right) yx, \tag{3}$$

while the population EM update rule for the imbalance $\tanh(\nu)$ is given by

$$N(\theta, \nu) := \mathbb{E}_{s \sim p(s|\theta^*, \pi^*)} \tanh\left(\frac{y\langle x, \theta \rangle}{\sigma^2} + \nu\right). \tag{4}$$

The corresponding finite-sample EM update rules with $n$ observations are given by

$$M_n(\theta, \nu) = \left( \frac{1}{n} \sum_{i=1}^n x_i x_i^\top \right)^{-1} \left( \frac{1}{n} \sum_{i=1}^n \tanh \left( \frac{y_i \langle x_i, \theta \rangle}{\sigma^2} + \nu \right) y_i x_i \right), \quad N_n(\theta, \nu) = \frac{1}{n} \sum_{i=1}^n \tanh \left( \frac{y_i \langle x_i, \theta \rangle}{\sigma^2} + \nu \right).$$
(5)

To further simplify the analysis of the finite-sample EM update rules, we introduce the following easy EM update rule for regression parameters:

$$M_n^{\text{easy}}(\theta, \nu) = \frac{1}{n} \sum_{i=1}^n \tanh \left( \frac{y_i \langle x_i, \theta \rangle}{\sigma^2} + \nu \right) y_i x_i.$$
(6)

The derivation of the EM update rules for the 2GMM model is on pages 4–6 of Weinberger & Bresler (2022), and the rigorous derivation of 2MLR for the population and finite-sample EM update rules (equations 3, 4, 5) can be found in Appendix B of Luo & Hashemi (2024).

## 2.2 Auxiliary Quantities

The superscript $t$ stands for the $t$-th EM iteration. For instance, $\theta^t$ and $\pi^t$ denote the $t$-th iteration of regression parameters and mixing weights. For the ease of theoretical analysis, we denote

$$\alpha^t := \frac{\|\theta^t\|}{\sigma}, \quad \beta^t := \tanh(\nu^t) = \pi^t(1) - \pi^t(2),$$
(7)

to be the $\ell_2$ norm of the normalized regression parameters $\theta^t/\sigma$, and $|\beta^t| = \|\pi^t - \frac{1}{2}\|_1 = |\pi^t(1) - 1/2| + |\pi^t(2) - 1/2|$ represents the $\ell_1$ distance between the mixing weight $\pi^t$ and the balanced mixing weights $(1/2, 1/2)$, namely the imbalance in mixtures, to further simplify the discussions on the convergence of EM iterations. Here, $\mathbb{1} = (1, 1)$ is the vector of all ones. To further simplify the analysis, we introduce these two functions $m(\alpha, \nu)$ and $n(\alpha, \nu)$ of $\alpha = \|\theta\|/\sigma$ and $\nu = (\ln \pi(1) - \ln \pi(2))/2$ by defining them as the expectations under the density $X \sim f_X(x) = \frac{K_0(|x|)}{\pi}$ involving the Bessel function $K_0$,

$$m(\alpha, \nu) = \mathbb{E}[\tanh(\alpha X + \nu) X], \quad n(\alpha, \nu) = \mathbb{E}[\tanh(\alpha X + \nu)].$$
(8)

In particular, we write $m_0(\alpha) = m(\alpha, 0) = \mathbb{E}[\tanh(\alpha X) X]$ for the case of $\tanh \nu = \pi(1) - \pi(2) = 0$.

# 3 Motivating Examples

As motivating examples, we highlight the tasks of haplotype assembly in bioinformatics and genomics (Cai et al., 2016) and phase retrieval, which arises in numerous fields including acoustics, optics, and quantum information (Candes et al., 2015), and also learning overparameterization models and Mixture of Experts (MoE) models, to justify the practical implications of our work.

## 3.1 Haplotype Assembly

Haplotypes are sequences of chromosomal variations in an individual's genome that are crucial for determining the individual's disease susceptibility. Haplotype assembly involves reconstructing these sequences from a mixture of sequenced chromosome fragments. Notably, humans have two haplotypes, i.e., they are diploid organisms (see Cai et al. (2016) for a clear mathematical formulation of the problem). For diploids, the primary challenge is to reconstruct two distinct haplotypes (binary sequences of single nucleotide polymorphisms— SNPs) from short, noisy sequencing reads. Each read corresponds to a local window of the genome but originates from one of the two chromosomes. The ambiguity in the haplotype origin of each read can be modeled as a mixture of two linear regression models with symmetric parameters, aligning with the model discussed in our work. Following Cai et al. (2016), let $\theta^* \in \{-1, +1\}^d$ represent one haplotype, and the other haplotype is its negative, $-\theta^*$. The binary variable $z_i \in \{-1, +1\}$ indicates the haplotype origin of the $i$-th

read, where the probability of the read originating from $\theta^*$ is $\mathbb{P}[z_i = +1] = \pi^*(1)$, and the probability of it originating from $-\theta^*$ is $\mathbb{P}[z_i = -1] = \pi^*(2)$. Furthermore, the $j$-th entry, $\varepsilon_i[j]$ for $j \in [d]$, of the noise vector $\varepsilon_i$ follows a distribution defined by a fixed error probability $p_e$: specifically, the noise causes an error (a flip) with probability $p_e$, meaning $\mathbb{P}(\varepsilon_i[j] = -2z_i\theta^*[j]) = p_e$, and the noise is zero (correct reading) with probability $1 - p_e$, meaning $\mathbb{P}(\varepsilon_i[j] = 0) = 1 - p_e$. Given this framework, the read signal $y_i$ can be modeled by the following two-mixture model: $y_i = z_i\theta^* + \varepsilon_i$. The primary goal is to estimate the unknown ground truth parameters, which are the mixing probabilities $\pi^* = (\pi^*(1), \pi^*(2))$ and the haplotype $\theta^*$, using the dataset of read signals $\{y_i\}_{i=1}^n$. It should be noted that while this is also a two-mixture model, it features a distinct formulation and noise structure.

## 3.2 Phase Retrieval

Regarding the application of the phase retrieval problem, as noted in Section 3 of Klusowski et al. (2019) and discussed in Section 3.5 of Chen et al. (2015), there is an established connection between the symmetric 2MLR and the phase retrieval problem. Specifically, by squaring the response variable $y_i$ and subtracting the variance $\sigma^2$, we obtain:

$$y_i' := y_i^2 - \sigma^2 = |\langle x_i, \theta^* \rangle|^2 + \xi_i. \tag{9}$$

This formulation is essentially the phase retrieval model with a heteroskedastic error term $\xi_i := 2(-1)^{z_i+1}\langle x_i, \theta^* \rangle \varepsilon_i + (\varepsilon_i^2 - \sigma^2)$, which has zero mean, i.e., $\mathbb{E}[\xi_i] = 0$. Therefore, by leveraging our results on symmetric 2MLR, we can directly establish convergence guarantees for the corresponding phase retrieval problem. For phase retrieval problems, several theoretical guarantees for the parameter estimate $\hat{\theta}$ have been established. Regarding the convex formulation (Chen et al., 2015), for $n \gtrsim d$ samples, the relative error bound is $\|\hat{\theta} - \theta^*\|/\sigma \lesssim \sqrt{d/n}\log^4 n + \min(\sqrt{d/n}/\eta, (d/n)^{1/4})\log^4 n$, where $\eta = \|\theta^*\|/\sigma$ represents the signal-to-noise ratio, as shown on page 10 of Chen et al. (2015). In the case of agnostic estimation (Neykov et al., 2016), for $n \gtrsim d^2 \log d$ samples, the error bound of the estimate $\hat{\theta}$ satisfies $\|\hat{\theta} - \theta^*\|/\|\theta^*\| \lesssim \sqrt{(d \log d)/n}$ under the constraint $\|\hat{\theta}\| = \|\theta^*\|$, as established on pages 3 and 9 of Neykov et al. (2016). For the EM algorithm when the sample size $n$ is sufficiently greater than the dimension $d$ ($n \gtrsim d$), the relative error bound for the estimated vector $\hat{\theta}$ depends on the initialization of the mixing weight $\pi^0$. With unbalanced initialization of mixing weights $\|\pi^0 - \frac{1}{2}\|_1 \gtrsim (d/n)^{1/4}$, the relative error is bounded by $\|\hat{\theta} - \theta^*\|/\sigma \lesssim \sqrt{d/n}$. Conversely, with balanced initialization of mixing weights $\|\pi^0 - \frac{1}{2}\|_1 \lesssim (d/n)^{1/4}$, the relative error bound is less favorable, given by $\|\hat{\theta} - \theta^*\|/\sigma \lesssim (d/n)^{1/4}$ (see Theorem 6.1). Regarding sample complexity, our results require $n \gtrsim d$ samples, matching the sample complexity requirement established in Chen et al. (2015) and improving upon the $n \gtrsim d^2 \log d$ requirement in Neykov et al. (2016). In terms of error rates, with balanced initialization, our rate $(d/n)^{1/4}$ matches the second term in Chen et al. (2015), demonstrating that the EM algorithm achieves a better rate when SNR $\eta$ is sufficiently small. With unbalanced initialization, our rate $\sqrt{d/n}$ matches the leading term $\sqrt{d/n}\log^4 n$ in Chen et al. (2015) and improves upon the $\sqrt{d \log d/n}$ rate in Neykov et al. (2016) by removing the logarithmic factor when SNR $\eta \asymp 1$. These theoretical comparisons demonstrate that our EM-based approach achieves competitive or improved error rates compared to existing phase retrieval methods, while providing explicit characterization of the initialization-dependent convergence behavior.

## 3.3 Overparametrization Models

For the general setting of overparameterization models, our theoretical results provide one of the fundamental examples. In our theoretical results of the EM algorithm for the 2MLR model, we exhibit linear convergence with an unbalanced guess of mixing weights, while showing sublinear convergence ($\alpha^t \asymp 1/\sqrt{t}$) with a balanced guess of mixing weights (see Theorems 5.1, 6.1), which is consistent with the convergence rate of $\mathcal{O}(1/\sqrt{t})$ for the overparameterized Gaussian mixture model (Xu et al., 2024). Interestingly, for the problem of low-rank matrix factorization (Xiong et al., 2023) in the overparameterization regime, an exponentially faster linear convergence rate is achieved using gradient descent with an asymmetric parameterization, while gradient descent with symmetric parameterization exhibits a sublinear rate of $1/t^2$. This suggests that imbalance accelerates the convergence rate. Overparameterization also has an impact on the convergence rate of gradient descent for learning a single neuron in neural networks (Xu & Du, 2023). The method exhibits linear

convergence in the exact-parameterization regime, but shows a sublinear convergence rate of $1/t^3$ in the overparameterization regime. Therefore, overparameterization can exponentially slow down the convergence rate of Gradient Descent (GD).

### 3.4 Mixture of Experts

While the EM algorithm has been shown to be a powerful tool for learning Mixture of Experts (MoE) models (Fruytier et al., 2025), establishing the convergence rates of the Maximum Likelihood Estimator (MLE) for these complex mixture models under exact-specified and over-specified settings remains a significant open challenge (Ho et al., 2022a; Nguyen et al., 2023b; 2024). In this context, our theoretical analysis of the EM algorithm for the 2MLR model serves as a fundamental example. By establishing rigorous guarantees in this setting, our work provides the necessary theoretical groundwork to deepen the understanding of convergence behaviors in more complex architectures, such as MoE and deep mixture models.

## 4 Population EM Updates

In this section, we characterize the population EM update rules by using expectations (equation 10) under the density involving Bessel function $K_0$. We show an alternative approach to derive Identity 4.1 (Corollary 3.2 of Luo & Hashemi (2024)) that leverages the key fact that the product $X = Z_1 Z_2$ of two independent standard Gaussians follows $X \sim \frac{K_0(|x|)}{\pi}$, from which we can derive the expectations (equation 10). We further show the nonincreasing property and boundedness (Facts 4.2, 4.3) of the expectations of EM update rules and provide approximate dynamic equations (Proposition 4.4) for the evolution of regression parameters and mixing weights.

**Identity 4.1.** *(Corollary 3.2 in Luo & Hashemi (2024): EM Updates for Overspecified 2MLR) Suppose a 2MLR model is fitted to the overspecified model with no separation, where $\theta^* = \vec{0}$. The EM update rules at the population level for $\bar{\theta}^t := \theta^t/\sigma = M(\theta^{t-1}, \nu^{t-1})/\sigma$ and $\tanh(\nu^t) := \pi^t(1) - \pi^t(2) = N(\theta^{t-1}, \nu^{t-1})$ are then as follows.*

$$\bar{\theta}^t = \frac{\bar{\theta}^0}{\|\bar{\theta}^0\|} \cdot \frac{1}{\pi} \int_{\mathbb{R}} \tanh(\|\bar{\theta}^{t-1}\|x - \nu^{t-1})x K_0(|x|)\mathrm{d}x,$$

$$\tanh(\nu^t) = \frac{1}{\pi} \int_{\mathbb{R}} \tanh(\nu^{t-1} - \|\bar{\theta}^{t-1}\|x)K_0(|x|)\mathrm{d}x,$$

*where $\bar{\theta}^0 := \theta^0/\sigma$.*

**Remark.** Identity 4.1 (see our derivation in Appendix B, Subsection B.1) completely characterizes the evolution of EM updates by using Bessel functions. Note that the proposition unveils that the EM update for regression parameters at the population level must be in the same direction as the previous EM iteration (as further corroborated numerically in Fig. 1a). The numerical experiments in Fig. 1a of EM updates validate the theoretical results. Hence, we only need to focus on the reduction of length in terms of the regression parameters; therefore, we introduce the $\ell_2$ norm of the normalized regression parameters $\alpha^t := \|\theta^t\|/\sigma$ and the imbalance of mixing weights $\beta^t := \tanh(\nu^t)$ to facilitate the analysis in the following context. Furthermore, the population EM update rules for $\alpha^t, \beta^t$ can be expressed as expectations with respect to a symmetric random variable $X$, whose probability density involves the Bessel function $K_0$, namely $X \sim \frac{K_0(|x|)}{\pi}$, given by

$$\alpha^{t+1} = m(\alpha^t, \nu^t) = \mathbb{E}[\tanh(\alpha^t X + \nu^t)X], \quad \beta^{t+1} = n(\alpha^t, \nu^t) = \mathbb{E}[\tanh(\alpha^t X + \nu^t)]. \tag{10}$$

**Fact 4.2.** *(Monotonicity of Expectations) Let $m(\alpha, \nu) := \mathbb{E}[\tanh(\alpha X + \nu)X]$ and $n(\alpha, \nu) := \mathbb{E}[\tanh(\alpha X + \nu)]$ be the expectations with respect to $X \sim \frac{K_0(|x|)}{\pi}$. Then they satisfy the monotonicity properties:*

*(monotonicity of $m(\alpha, \nu)$): $m(\alpha, \nu)$ is a nonincreasing function of $\nu \geq 0$ for fixed $\alpha \geq 0$, and a nondecreasing function of $\alpha \geq 0$ for fixed $\nu \geq 0$, namely:*

$$0 = m(\alpha, \infty) \leq m(\alpha, \nu') \leq m(\alpha, \nu) \leq m(\alpha, 0) \leq \alpha \text{ for } 0 \leq \nu \leq \nu', \alpha \geq 0,$$

$$0 = m(0, \nu) \leq m(\alpha, \nu) \leq m(\alpha', \nu) \leq m(\infty, \nu) = \frac{2}{\pi} \text{ for } 0 \leq \alpha \leq \alpha', \nu \geq 0.$$

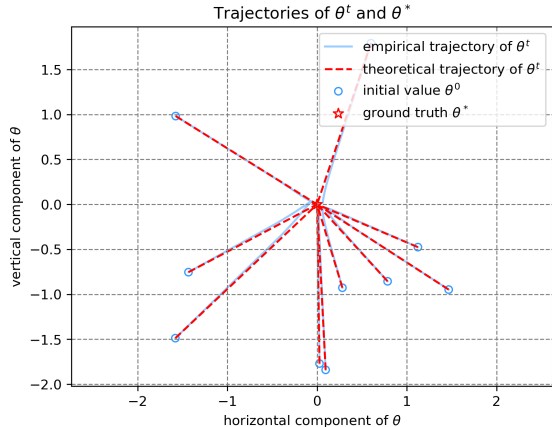

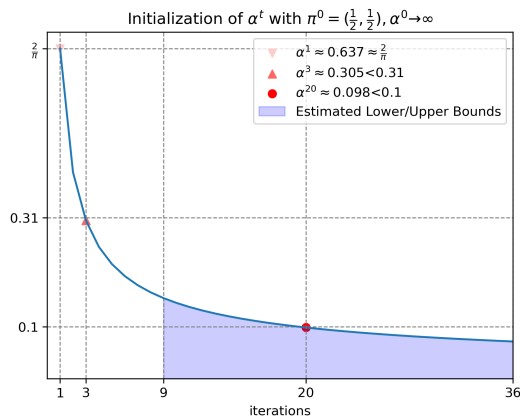

(a) Trajectories of EM iterations for regression parameters in the overspecified setting given different initial values. $d = 2$, trajectories of $\theta^t$ for 10 trials with $\theta^0$ and $\pi^0$ uniformly sampled from $[-2, 2]^2$ and $[0, 1]$, respectively in the overspecified setting $\theta^* = \vec{0}$.

(b) Initialization phase of $\alpha^t := \|\theta^t\|/\sigma$ in the worst-case scenario with a balanced initial guess $\pi^0 = (\frac{1}{2}, \frac{1}{2})$ and $\alpha^0 := \|\theta^0\|/\sigma \to \infty$. Even in the worst case, the initialization phase of $\alpha^t$ is bounded by $\alpha^1 \leq \frac{2}{\pi}$, $\alpha^3 < 0.31$, and $\alpha^{20} < 0.1$ after 1, 3, and 20 EM iterations.

Figure 1: Left: EM trajectories are nearly perfect rays from the origin to the initial point, which aligns with the theoretical results in Identity 4.1. Right: In the worst case, we show that $\alpha^t \geq 0.1$ for all $t \leq 9$ (see remark on the proof of Fact 5.2 in Appendix D, Subsection D.2) by using the theoretical matching lower bound for the worst case in Proposition 5.3. Also, we demonstrate that $\alpha^t < 0.1$ for all $t \geq 36$ (Fact 5.2) by applying the theoretical upper bound in Proposition 5.3. As $\alpha^{20} \approx 0.1$ by numerical evaluations, and $20 > 9$ and $20 < 36$, the theoretical results are consistent with the numerical results shown in the figure.

*(monotonicity of $n(\alpha, \nu)$): $n(\alpha, \nu)$ is a nonincreasing function of $\alpha \geq 0$ for fixed $\nu \geq 0$, and a nondecreasing function of $\nu \geq 0$ for fixed $\alpha \geq 0$, namely:*

$$0 = n(\infty, \nu) \leq n(\alpha', \nu) \leq n(\alpha, \nu) \leq n(0, \nu) = \tanh(\nu) \text{ for } 0 \leq \alpha \leq \alpha', \nu \geq 0,$$
$$0 = n(\alpha, 0) \leq n(\alpha, \nu) \leq n(\alpha, \nu') \leq n(\alpha, \infty) = 1 \text{ for } 0 \leq \nu \leq \nu', \alpha \geq 0.$$

**Remark.** $m(\alpha, \nu)$ is an even function of $\nu$ and $n(\alpha, \nu)$ is an odd function of $\nu$. Moreover, by using the Fact 4.2 together with equation 10, we can establish the bounded and nonincreasing properties of $\{\alpha^t\}_{t=0}^{\infty}, \{|\beta^t|\}_{t=0}^{\infty}$.

**Fact 4.3.** *(Nonincreasing and Bounded) Let $\alpha^t := \|\theta^t\|/\sigma = \|M(\theta^{t-1}, \nu^{t-1})\|/\sigma$ and $\beta^t := \tanh(\nu^t) = N(\theta^{t-1}, \nu^{t-1})$ for all $t \in \mathbb{Z}_+$ be the $t$-th iteration of the EM update rules $\|M(\theta, \nu)\|/\sigma$ and $N(\theta, \nu)$ at the population level, then $\beta^t \cdot \beta^0 \geq 0$, $\alpha^t \leq 2/\pi$, and $\{\alpha^t\}_{t=0}^{\infty}$ and $\{|\beta^t|\}_{t=0}^{\infty}$ are non-increasing.*

**Remark.** In Fact 4.3 (see proof in Appendix B, Subsection B.2), the nonincreasing property of $\|\theta^t\|$ and $\|\pi^t - \frac{1}{2}\|_1$ at the population level indicates that the estimates of regression parameters gradually approach the ground truth $\theta^* = \vec{0}$, while the estimates for mixing weights shift from "unbalanced" to "balanced". For the case of $\pi^0 = (1/2, 1/2)$, we always have $\beta^t = \beta^0 = \|\pi^0 - \frac{1}{2}\|_1 = 0$ by using the nonincreasing property of $\{|\beta^t|\}_{t=0}^{\infty}$. Additionally, the bounded $\|\theta\|/\sigma$ ensures that EM iterations remain within a bounded region, regardless of the initial distance from the ground truth $\theta^* = \vec{0}$.

**Proposition 4.4.** *(Approximate Dynamic Equations) Let $\alpha^t := \|\theta^t\|/\sigma = \|M(\theta^{t-1}, \nu^{t-1})\|/\sigma$ and $\beta^t := \tanh(\nu^t) = N(\theta^{t-1}, \nu^{t-1})$ for all $t \in \mathbb{Z}_+$ be the $t$-th iteration of the EM update rules $\|M(\theta, \nu)\|/\sigma$ and $N(\theta, \nu)$ at the population level, then the series approximations for EM update rules are*

$$\begin{aligned} \alpha^{t+1} &= \alpha^t(1 - [\beta^t]^2) + \mathcal{O}([\alpha^t]^3), \\ \beta^{t+1} &= \beta^t(1 - \alpha^t \alpha^{t+1}) + \mathcal{O}([\alpha^t]^4). \end{aligned}$$

**Remark.** Proposition 4.4 (see proof in Appendix B, Subsection B.3) characterizes the evolution of EM updates by introducing approximate dynamic equations when the regression parameters are small enough. When $\alpha^t$ is

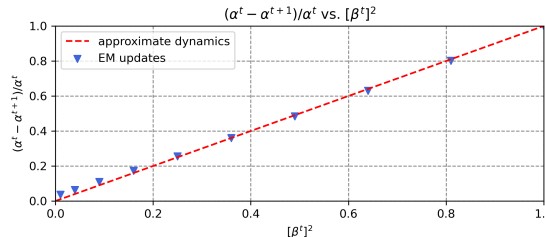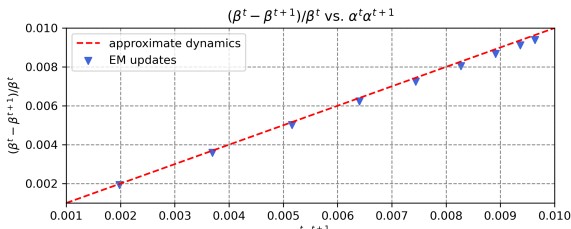

Figure 2: Top: Relation of the $(\alpha^t - \alpha^{t+1})/\alpha^t$ and $[\beta^t]^2$ given $\alpha^t = 0.1$ and $\beta^t \in \{0.1, 0.2, \cdots, 0.9, 1\}$. Bottom: Relation of the $(\beta^t - \beta^{t+1})/\beta^t$ and $\alpha^t \alpha^{t+1}$ given $\alpha^t = 0.1$ and $\beta^t \in \{0.1, 0.2, \cdots, 0.9, 1\}$. The difference between $(\alpha^t - \alpha^{t+1})/\alpha^t$ and $[\beta^t]^2$ is bounded by $(1 - [\beta^t]^2)\mathcal{O}([\alpha^t]^2)$, and the difference between $(\beta^t - \beta^{t+1})/\beta^t$ and $\alpha^t \alpha^{t+1}$ is bounded by $(1 - [\beta^t]^2)\mathcal{O}([\alpha^t]^4)$ (see remark on the proof of Proposition 4.4 in Appendix B, Subsection B.3).

sufficiently small, the higher order terms for $\alpha^{t+1}$ and $\beta^{t+1}$ are $(1 - [\beta^t]^2)\mathcal{O}([\alpha^t]^3)$ and $\beta^t(1 - [\beta^t]^2)\mathcal{O}([\alpha^t]^4)$ respectively (see the remark on the proof of Proposition 4.4 in Appendix B, Subsection B.3). We illustrate our findings by comparing the EM updates at the population level with our approximate dynamics, as shown in Fig. 2.

Further, we precisely outline the evolution of EM estimates for both regression parameters and mixing weights as opposed to just the former as done in Dwivedi et al. (2020b). Using the techniques from pages 30-33 of Dwivedi et al. (2020a), we can only establish a rough upper bound for the update $|\beta^{t+1}| := |\pi^{t+1}(1) - \pi^{t+1}(2)|$, which represents the imbalance of mixing weights. However, to fully understand the evolution of EM iterations, it is crucial to establish a lower bound for $|\beta^{t+1}|$. We achieve this by bounding expectations involving the Bessel function $K_0$ (see Lemma A.7 and Lemma C.2). By applying this lower bound for the imbalance $|\beta^t|$, we further establish Theorem 5.1, which characterizes the convergence rate at the population level.

In the special case of balanced mixing weights, namely $\beta^t = \tanh \nu^t = 0$ then $1 - [\beta^t]^2 = 1$, we have $\beta^{t+1} = \beta^t$ by using Fact 4.3, the lower/upper bounds for $\alpha^{t+1}$ when $\alpha^t$ is sufficiently small by introducing $[\alpha^t]^3$ term:

$$\alpha^t - 3[\alpha^t]^3 \leq \alpha^{t+1} \leq \alpha^t - \frac{3[\alpha^t]^3}{1 + 8[\alpha^t]}, \tag{11}$$

where the upper bound $\alpha^t - 3[\alpha^t]^3/(1 + 8[\alpha^t])$ still holds when $\beta^t \neq 0$ (see remark on the proof of Proposition 4.4 in Appendix B, Subsection B.3), and we further establish Proposition 5.3 by using the above bounds for $\alpha^{t+1}$.

# 5 Population Level Analysis

In this section, we give an analysis of population EM and establish convergence rate guarantees for population EM (Theorem 5.1) with balanced and unbalanced initial guesses for mixing weights. We show theoretical bounds for sublinear convergence (Proposition 5.3) with balanced initial guesses and the contraction factor of linear convergence (Proposition 5.4) with unbalanced initial guesses, by bounding the expectations (equation 10) under the density involving Bessel functions for EM updates in the overspecified setting.

**Main theorem 5.1.** *(Convergence Rate at Population Level) Suppose a 2MLR model is fitted to the overspecified model with no separation $\theta^* = \vec{0}$, then for any $\epsilon \in (0, 2/\pi)$:*

*(unbalanced) if $\pi^0 \neq \left(\frac{1}{2}, \frac{1}{2}\right)$, population EM takes at most $T = \mathcal{O}\left(\log \frac{1}{\epsilon}\right)$ iterations to achieve $\|\theta^T\|/\sigma \leq \epsilon$,*

*(balanced) if $\pi^0 = \left(\frac{1}{2}, \frac{1}{2}\right)$, population EM takes at most $T = \mathcal{O}(\epsilon^{-2})$ iterations to achieve $\|\theta^T\|/\sigma \leq \epsilon$.*

**Remark.** In Theorem 5.1 (see proof in Appendix D, Subsection D.4), with an unbalanced initial guess for mixing weights, that is $\beta^0 \neq 0$, we further show the linear convergence and the upper bound for the contraction factor $\alpha^{t+1}/\alpha^t$, therefore EM updates achieve the $\epsilon$-accuracy in $\mathcal{O}(\log(1/\epsilon))$ steps. Even in the worst case (the initial guess for mixing weights is balanced, $\beta^0 = 0$), we still show that $\alpha^t = \mathcal{O}(t^{-\frac{1}{2}})$ holds

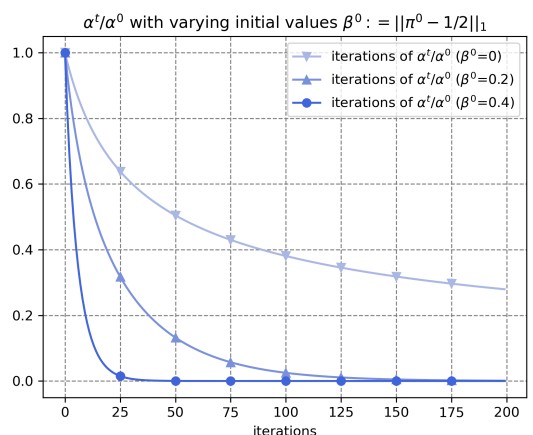
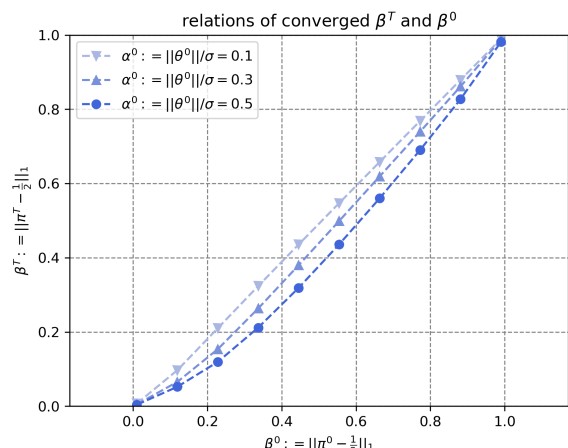

(a) Interpolation across initial mixing weight regimes $\pi^0 = (\frac{1}{2}, \frac{1}{2}), (0.6, 0.4), (0.7, 0.3)$, showing sublinear convergence for $\pi^0 = (\frac{1}{2}, \frac{1}{2})$ and linear convergence for $\pi^0 = (0.6, 0.4), (0.7, 0.3)$, with the normalized regression parameter $\alpha^0 = \|\theta^0\|/\sigma = 0.1$.

(b) Relation of the converged value $\beta^\infty \approx \beta^T = \|\pi^T - \frac{1}{2}\|_1$ and the initial value $\beta^0 = \|\pi^0 - \frac{1}{2}\|_1$ when $\beta^0 = \pi^0(1) - \pi^0(2) \geq 0$, where $\beta^0$ takes ten evenly spaced values between 0.01 and 0.99, given different initial $\alpha^0 := \frac{\|\theta^0\|}{\sigma} = 0.1, 0.3, 0.5$.

Figure 3: Comparison of convergence behavior and EM trajectories at population level.

(Proposition 5.3), and it takes at most $\mathcal{O}(\epsilon^{-2})$ iterations to achieve $\epsilon$-accuracy in $\ell_2$ norm. We validate the sublinear convergence rate with numerical results, see Fig. 4a. Compared to $\mathcal{O}(\epsilon^{-2}\log(1/\epsilon))$ steps taken to achieve the $\epsilon$-accuracy for 2GMM with known balanced mixing weights $\pi^t = (\frac{1}{2}, \frac{1}{2})$ (Equation (15), page 16 in Dwivedi et al. (2020b)), we give a better estimation $\mathcal{O}(\epsilon^{-2})$ for steps in 2MLR.

**Intuitive Explanation of Theorem 5.1.** For GMM/MLR models, running the EM algorithm is equivalent to performing a step of Gradient Descent on the negative log-likelihood (Kwon et al., 2024). When an unbalanced estimate of mixing weights is given, the negative log-likelihood retains its dominant quadratic term, $[\alpha]^2$. This quadratic form ensures that the EM algorithm behaves like a Gradient Descent step on a strongly convex function, suggesting a fast, linear convergence rate for $\alpha^t$. In contrast, when a balanced estimate of mixing weights is given, the $[\alpha]^2$ term in the negative population log-likelihood cancels out (page 22 of Luo & Hashemi (2024) and Lemma A.3). Specifically, the likelihood term can be expanded as $[\alpha]^2/2 - \mathbb{E}[\ln\cosh(\alpha X)] = [\alpha]^2/2 - \mathbb{E}[(\alpha X)^2/2 - (\alpha X)^4/12 + \ldots] \approx 3[\alpha]^4/4$. The EM algorithm, being equivalent to Gradient Descent, then follows a path approximated by $\alpha^{t+1} \approx \alpha^t - \nabla(3[\alpha^t]^4/4) = \alpha^t - 3[\alpha^t]^3$, leading to a much slower, sublinear convergence rate where $\alpha^t$ is proportional to $1/\sqrt{t}$.

**Proof Sketch. of Theorem 5.1.** By invoking Fact 5.2 of initialization at population level, it is guaranteed that $\alpha^{T_0} = \|\theta^{T_0}\|/\sigma < 0.1$ after running population EM for at most $T_0 = \mathcal{O}(1)$ iterations. By applying the nonincreasing property of $\{\alpha^t\}_{t=0}^\infty$, we show that $\alpha^t \leq \alpha^{T_0} < 0.1$ for $t \geq T_0$

For the case of $\pi^0 = (1/2, 1/2)$, by applying the sublinear convergence guarantee in Proposition 5.3 and selecting $t = \Theta(\epsilon^{-2})$, the normalized regression parameters $\alpha^{t+T_0} = \mathcal{O}(t^{-1/2} \wedge \alpha^{T_0}) = \mathcal{O}(t^{-1/2}) \leq \epsilon$ achieve $\varepsilon$-accuracy within $t + T_0 = \Theta(\epsilon^{-2}) + \mathcal{O}(1) = \Theta(\epsilon^{-2})$ iterations.

For the case of $\pi^0 \neq (1/2, 1/2)$, namely $\beta^0 \neq 0$, by using the bound for the contraction factor $\alpha^{t+1+T_0}/\alpha^{t+T_0}$ in Proposition 5.4 repeatedly, then $\alpha^{t+T_0}/\alpha^{T_0} \leq \alpha^{t+T_0}/0.1 \leq 10(1 - c[\beta^\infty]^2)^t$ for some $c > 0$. Then by selecting $t = \Theta(\log(1/\epsilon)/(-\log(1 - c[\beta^\infty]^2))) = \Theta(\log(1/\epsilon))$, it is sufficient to show that $\alpha^{t+T_0} \leq \epsilon$.

**Fact 5.2.** *(Initialization at Population Level)* *Let* $\alpha^t := \|\theta^t\|/\sigma = \|M(\theta^{t-1}, \nu^{t-1})\|/\sigma$ *and* $\beta^t := \tanh(\nu^t) = N(\theta^{t-1}, \nu^{t-1})$ *for all* $t \in \mathbb{Z}_+$ *be the t-th iteration of the EM update rules* $\|M(\theta, \nu)\|/\sigma$ *and* $N(\theta, \nu)$ *at the population level. If we run EM at the population level for at most* $T_0 = 36$ *iterations, then* $\alpha^{T_0} < 0.1$.

**Remark.** By using such nonincreasing and bounded property in Fact 4.3 and applying the EM update rules in Identity 4.1, we show that the regression parameters will converge to a small region $\|\theta^t\|/\sigma < 0.31$ within

only three iterations of EM updates. Then by invoking the following upper bound in Proposition 5.3, it takes at most 33 iterations for the value to drop below 0.1. Thus, Fact 5.2 is established (see proof in Appendix D, Subsection D.2). Similarly, we can show that in the worst case of sublinear convergence with balanced initial guess $\pi^0 = (\frac{1}{2}, \frac{1}{2})$ and $\alpha^0 = \|\theta^0\|/\sigma \to \infty$, we have $\alpha^3 = \|\theta^3\|/\sigma \approx 0.31$ and $\alpha^t \geq 0.1$ for all $t \leq 9$ by using the matching lower bound in Proposition 5.3, which also matches the numerical results in Fig. 1b.

**Proposition 5.3.** *(Sublinear Convergence Rate) Let $\alpha^t := \|\theta^t\|/\sigma = \|M(\theta^{t-1}, \nu^{t-1})\|/\sigma$ and $\beta^t := \tanh(\nu^t) = N(\theta^{t-1}, \nu^{t-1})$ for all $t \in \mathbb{Z}_+$ be the $t$-th iteration of the EM update rules $\|M(\theta, \nu)\|/\sigma$ and $N(\theta, \nu)$ at the population level. Suppose that $\alpha^0 \in (0, 0.31)$, then*

$$\alpha^t \leq \frac{1}{\sqrt{6t + \left\{8 + \frac{1}{\alpha^0}\right\}^2 - 8}} \quad \forall t \in \mathbb{Z}_{\geq 0};$$

*when the initial guess of mixing weights is balanced $\pi^0 = (\frac{1}{2}, \frac{1}{2})$, namely $\beta^0 = \tanh \nu^0 = \|\pi^0 - \frac{1}{2}\|_1 = 0$, then*

$$\alpha^t \geq \frac{1}{\sqrt{6t + 22\ln(1.2t + 1) + (\frac{1}{\alpha^0})^2}} \quad \forall t \in \mathbb{Z}_{\geq 0}.$$

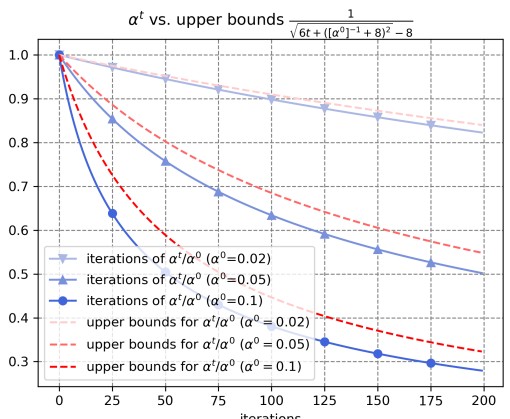
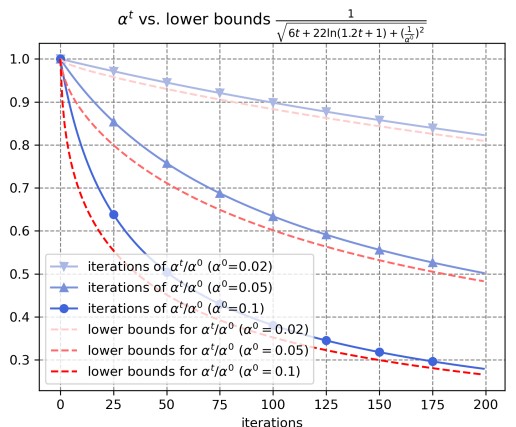

(a) Upper bound illustration for sublinear convergence with balanced initial mixing weights.

(b) Lower bound illustration for sublinear convergence with balanced initial mixing weights.

Figure 4: Sublinear convergence rate bounds for $\alpha^t := \|\theta^t\|/\sigma$ as shown in Proposition 5.3, with different initial values $\alpha^0 = 0.02, 0.05, 0.1$ and balanced initial guess $\beta^0 := \|\pi^0 - \frac{1}{2}\|_1 = 0$ over 200 EM iterations.

**Remark.** The above Proposition 5.3 (see proof in Appendix D, Subsection D.1) highlights that at the population level, when the initial guess of the mixing weights is balanced, the EM updates exhibit an excessively slow convergence rate. To establish the above tight bounds, we present much tighter lower and upper bounds for $\alpha^{t+1}$ with the same coefficients of $[\alpha^t]^3$ in equation 11 (see remark on Proposition 4.4)). Furthermore, we developed a technique of "variable separation" by interpreting the upper bound $\alpha^{t+1} \leq \alpha^t - 3[\alpha^t]^3/(1 + 8\alpha^t)$ in equation 11 as the discretized version of a differential inequality $\mathrm{d}\alpha \leq -3[\alpha]^3 \mathrm{d}t$. Consequently, we derived the simple upper bound for $\alpha^t$ in Proposition 5.3 by solving the discretized version of the differential inequality. For sufficiently small $\alpha^t$, we could further establish a tighter upper bound $\alpha^{t+1} \leq \alpha^t - 3[\alpha^t]^3/(1 + 10[\alpha^t]^2)$, and the "variable separation" technique could be applied to derive the other upper bound for $\alpha^t$ by introducing Lambert $W$ function (see Section 4.13 Lambert W-Function in Olver et al. (2010)), namely $\alpha^t \leq 1/\sqrt{6t + (\frac{1}{\alpha^0})^2 - 10\ln(6[\alpha^0]^2 t - 10[\alpha^0]^2 \ln(10[\alpha^0]^2) + 1)}$ (see remark on the proof of Proposition 5.3 in Appendix D, Subsection D.1). As $\alpha^0$ approaches 0, the logarithmic term in the above upper bound for $\alpha^t$ becomes negligible, and the upper bound can be approximated by $1/\sqrt{6t + (\frac{1}{\alpha^0})^2}$. Our lower/upper bounds for $\alpha^t$ are tight given the balanced initial guess $\pi^0 = (\frac{1}{2}, \frac{1}{2})$, since $t \gtrsim \ln(ct + c')$ for some constants $c, c' > 0$.

**Proposition 5.4.** *(Contraction Factor for Linear Convergence) Let $\alpha^t := \|\theta^t\|/\sigma = \|M(\theta^{t-1}, \nu^{t-1})\|/\sigma$ and $\beta^t := \tanh(\nu^t) = N(\theta^{t-1}, \nu^{t-1})$ for all $t \in \mathbb{Z}_+$ be the $t$-th iteration of the EM update rules $\|M(\theta, \nu)\|/\sigma$ and $N(\theta, \nu)$ at the population level. Then $\beta^\infty := \lim_{t \to \infty} \beta^t$ exists and*

*(i) if $\beta^0 = 0$, then $\beta^\infty = 0$,*

*(ii) if $\beta^0 > 0$, then $0 < \beta^\infty \leq \beta^t \leq \beta^0$,*

*(iii) if $\beta^0 < 0$, then $0 > \beta^\infty \geq \beta^t \geq \beta^0$.*

*Suppose that $\alpha^t \in (0, 0.1)$, then*

$$\frac{\alpha^{t+1}}{\alpha^t} \leq 1 - \frac{4}{5}[\beta^\infty]^2.$$

**Remark.** Proposition 5.4 (see proof in Appendix D, Subsection D.3) provides the upper bound for the contraction factor $\alpha^{t+1}/\alpha^t$ of linear convergence when $\alpha^t = \|\theta^t\|/\sigma$ is small enough, which is strictly less than 1 when the initial guess of mixing weights $\pi^0 \neq (\frac{1}{2}, \frac{1}{2})$. This upper bound for contraction factor is obtained by establishing the bounds (see Subsection A.3 and Appendix C) for expectations in Identity 4.1, thereby $\alpha^{t+1}/\alpha^t \leq 1 - (5/3)[\alpha^t]^2 - (4/5)[\beta^t]^2 \leq 1 - (4/5)[\beta^\infty]^2$. Furthermore, we can provide the lower/upper bounds for $\beta^\infty$ by showing that $-[\alpha^0]^2/(300[\beta^0]^{20}) \leq \ln |\beta^\infty| - \ln |\beta^0| \leq -[\alpha^0]^2/4$ when $\alpha^0 < 0.1, |\beta^0| < \sqrt{2/5}$ (see proof of Proposition 5.4 and its remark in Appendix D, Subsection D.3). Also, we can show the lower bound for the contraction factor $\alpha^{t+1}/\alpha^t \geq 1 - 3[\alpha^t]^2 - [\beta^t]^2 \geq 1 - \Theta(\frac{1}{t}) - [\beta^0]^2$ by using Proposition 5.3 and Fact 4.3 when $\alpha^0 < 0.1$ is sufficiently small (see Appendix C for the derivation of the first inequality). Since $\alpha^{t+1}/\alpha^t$ is lower bounded and upper bounded by some constants which are in the range of $(0, 1)$ when $t$ is sufficiently large, $\alpha^t$ converges linearly when the initial guess $\pi^0 \neq (\frac{1}{2}, \frac{1}{2})$ is unbalanced (namely $\beta^0 \neq 0$).

## 6 Finite-Sample Level Analysis

In this section, we give a finite-sample analysis and present tight bounds for sample complexity, time complexity, and final accuracy (Theorem 6.1) by coupling the analysis of population EM and finite-sample EM and establishing statistical errors (Propositions 6.3, 6.4) in the overspecified setting.

**Main theorem 6.1.** *(Convergence Rate at Finite-Sample Level for Fixed Mixing Weights)*

*Suppose a 2MLR model is fitted to the overspecified model with no separation $\theta^* = \vec{0}$, given mixing weights $\pi^t = \pi^0$ and $n = \Omega\left(d \vee \log \frac{1}{\delta} \vee \log^3 \frac{1}{\delta'}\right)$ samples:*

*(sufficiently unbalanced) if $\left\|\pi^0 - \frac{1}{2}\right\|_1 \gtrsim \left[\frac{d \vee \log \frac{1}{\delta}}{n}\right]^{\frac{1}{4}}$, finite-sample EM takes at most $T = \mathcal{O}\left(\left\|\pi^0 - \frac{1}{2}\right\|_1^{-2} \log \frac{n}{d \vee \log \frac{1}{\delta}}\right)$ iterations to achieve $\|\theta^T\|/\sigma = \mathcal{O}\left(\left\|\pi^0 - \frac{1}{2}\right\|_1^{-1} \left[\frac{d \vee \log \frac{1}{\delta}}{n}\right]^{\frac{1}{2}}\right)$,*

*(sufficiently balanced) if $\left\|\pi^0 - \frac{1}{2}\right\|_1 \lesssim \left[\frac{d \vee \log \frac{1}{\delta}}{n}\right]^{\frac{1}{4}}$, finite-sample EM takes at most $T = \mathcal{O}\left(\left[\frac{n}{d \vee \log \frac{1}{\delta}}\right]^{\frac{1}{2}}\right)$ iterations to achieve $\|\theta^T\|/\sigma = \mathcal{O}\left(\left[\frac{d \vee \log \frac{1}{\delta}}{n}\right]^{\frac{1}{4}}\right)$, with probability at least $1 - T(\delta + \delta')$.*

**Remark.** Theorem 6.1 (see proof in Appendix G, Subsection G.2) provides the final statistical errors, time complexity, and sample complexity in the specific setting of fixed mixing weights. There is a tight connection between Theorem 6.1 at the finite-sample level and Theorem 5.1 at the population level. In particular, by setting the final accuracy $\epsilon$ in Theorem 5.1 for regression parameters to be $\mathcal{O}((d/n)^{1/2})$ for the "sufficiently unbalanced" case, we intuitively obtain the required iteration complexity $T = \mathcal{O}(\log \frac{1}{\epsilon}) = \mathcal{O}(\log \frac{n}{d})$ for the "sufficiently unbalanced" case, which is consistent with the iteration complexity $\mathcal{O}(\log \frac{n}{d})$ in Theorem 6.1 at the finite-sample level. Moreover, by setting the final accuracy $\epsilon$ in Theorem 5.1 for regression parameters to be $\mathcal{O}((d/n)^{1/4})$ for the "sufficiently balanced" case, we intuitively obtain the required iteration complexity $T = \mathcal{O}(\epsilon^{-2}) = \mathcal{O}((n/d)^{1/2})$ for the "sufficiently balanced" case, which is consistent with the time complexity $\mathcal{O}((n/d)^{1/2})$ in Theorem 6.1 at the finite-sample level. This main theorem addresses all regimes where the mixing weights remain fixed, $\pi^t = \pi^0$. In contrast, Theorems 1 and 3 in Dwivedi et al. (2020b) cover only

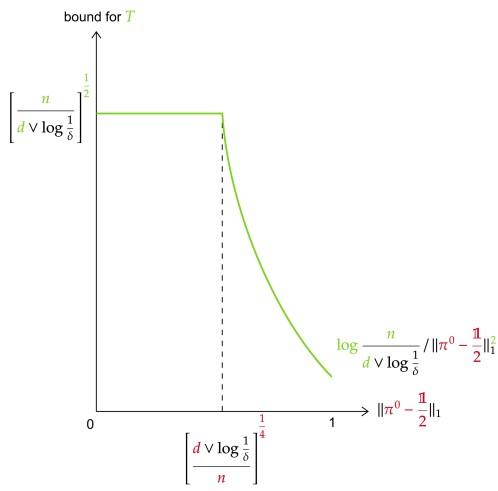

(a) Bounds of the time complexity $T$ in both sufficiently unbalanced and balanced cases.

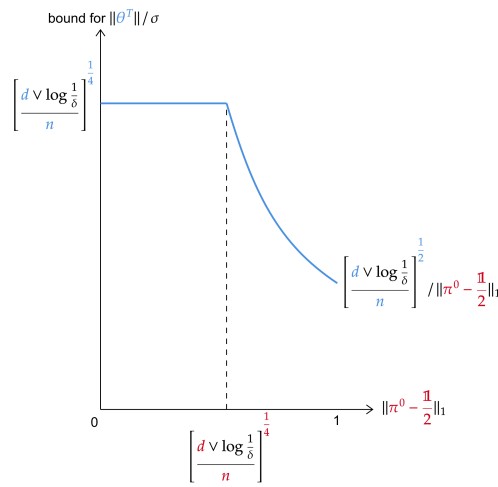

(b) Bounds of the statistical accuracy $\|\theta^T - \theta^*\|/\sigma$ in both sufficiently unbalanced and balanced cases.

Figure 5: Theorem 6.1 provides bounds on time complexity and statistical accuracy for fixed mixing weights $\pi^t = \pi^0$ in both sufficiently unbalanced and balanced cases.

the cases of sufficiently unbalanced Gaussian mixtures, where $\|\pi^0 - \frac{1}{2}\|_1 \gtrsim \mathcal{O}((d/n)^{1/4})$ and the special case $\pi^0 = (\frac{1}{2}, \frac{1}{2})$; while our main theorem covers all regimes of mixing weights, including the "sufficiently balanced" case $\|\pi^0 - \frac{1}{2}\|_1 \lesssim \mathcal{O}((d/n)^{1/4})$, which is handled by introducing our proof technique of "variable separation". In addition to developing new proof techniques, we improve the bounds in Theorem 3 of Dwivedi et al. (2020b) for statistical accuracy $\mathcal{O}\left(\left[\frac{d\vee\log\frac{1}{\delta}\vee\log\log\frac{1}{\epsilon'}}{n}\right]^{\frac{1}{4}-\epsilon'}\right)$, time complexity $\mathcal{O}\left(\left(\frac{n}{d}\right)^{\frac{1}{2}-2\epsilon'}\log\frac{n}{d}\log\frac{1}{\epsilon'}\right)$, and sample complexity $\Omega(d \vee \log\frac{1}{\delta} \vee \log\log\frac{1}{\epsilon'})$ for $\epsilon' \in (0, 1/4)$ in the case of balanced mixing weights $\pi^0 = (\frac{1}{2}, \frac{1}{2})$. The previous bounds diverged as $\epsilon' \to 0$ whereas our time and sample complexity remain stable.

**Intuitive Explanation of Theorem 6.1.** The final statistical accuracy of the MLE estimate is governed by the invertibility of the Fisher Information matrix, which is the second-order derivative of the negative population log-likelihood. With an unbalanced estimate of mixing weights, the negative log-likelihood maintains the $[\alpha]^2 = \|\theta\|^2/\sigma^2$ term, which ensures the Fisher Information matrix (w.r.t. the regression parameters) is invertible. As a well-established result (Van der Vaart, 2000), this invertibility guarantees that the MLE estimate achieves the standard parametric rate of order $n^{-1/2}$, and therefore the final accuracy of the EM algorithm is also proportional to $n^{-1/2}$. However, when a balanced estimate is given, the critical term $[\alpha]^2$ vanishes, causing the Fisher Information matrix to be singular. This singularity slows down the standard parametric rate of final accuracy (Dwivedi et al., 2020b), resulting in a converged $\alpha^T$ that exhibits a rate proportional to $n^{-1/4}$.

**Proof Sketch. of Theorem 6.1.** Initially, Fact 6.2 ensures that after at most $T_0 = \mathcal{O}(1)$ finite-sample EM iterations, we have $\alpha^{T_0} = \|\theta^{T_0}\|/\sigma < 0.1$ with high probability. Therefore, without loss of generality, we assume $\alpha^0 < 0.1, \beta^0 \geq 0$. For simplicity, we denote finite-sample and population updates for the norm of normalized regression parameters as $\alpha^t, \bar{\alpha}^t$, respectively.

We upper bound the $(t + 1)$-th finite-sample EM iteration $\alpha^{t+1}$ using the triangle inequality $\alpha^{t+1} \leq |\alpha^{t+1} - \bar{\alpha}^{t+1}| + \bar{\alpha}^{t+1}$. The first term represents the statistical error $|\alpha^{t+1} - \bar{\alpha}^{t+1}| = (\beta^t + \alpha^t)\mathcal{O}((d/n)^{1/2})$ (see Proposition 6.4), while the second term $\bar{\alpha}^t$ is the population EM update, which is bounded by the upper bounds provided in Appendix C.

For the case where $\beta^t = \beta^0 \geq \Theta((d/n)^{1/4})$, by selecting sufficiently large sample size $n$ and simplifying the expression, we derive the recurrence relation $\alpha^{t+1} - \Theta((d/n)^{1/4}/\beta^0) \leq (1 - c[\beta^0]^2)(\alpha^t - \Theta((d/n)^{1/4}/\beta^0))$ for some $c > 0$. Using the approximation $1 - c[\beta^0]^2 \leq \exp(-c[\beta^0]^2)$, we can express the above relation

recursively, leading to: $\alpha^T - \Theta((d/n)^{1/4}/\beta^0) = \mathcal{O}(\exp(-c[\beta^0]^2 T))$. By choosing $T = \Theta(\log(n/d)/[\beta^0]^2)$, we achieve $\epsilon$-accuracy for the normalized regression parameters, yielding: $\alpha^T \leq \Theta((d/n)^{1/4}/\beta^0)$.

In the case where $\beta^0 = \beta^t \leq \Theta((d/n)^{1/4})$, we similarly derive a recurrence relation by selecting sufficiently large sample size $n$ and simplifying the expression, $\alpha^{t+1} \leq \alpha^t - c[\alpha^t]^3$ for some $c > 0$ when $\alpha^t \geq \Theta((d/n)^{1/4})$. This relation can be viewed as the discretized version of the ordinary differential inequality: $d\alpha \leq -c\alpha^3 dt$. By applying the method of "variable separation" to this discretized version, we obtain $\alpha^t \leq \Theta(t^{-1/2})$ when $\alpha^t \geq \Theta((d/n)^{1/4})$. To achieve $\alpha^T \leq \Theta((d/n)^{1/4})$, we select $T = \Theta((d/n)^{1/4})^{-2} = \Theta((n/d)^{1/2})$.

**Fact 6.2.** *(Initialization at Finite-Sample Level) Let $\alpha^t := \|\theta^t\|/\sigma = \|M_n(\theta^{t-1}, \nu^{t-1})\|/\sigma$ and $\beta^t := \tanh(\nu^t) = N_n(\theta^{t-1}, \nu^{t-1})$ for all $t \in \mathbb{Z}_+$ be the $t$-th iteration of the EM update rules $\|M_n(\theta, \nu)\|/\sigma$ and $N_n(\theta, \nu)$ at the finite-sample level. If we run EM at the finite-sample level for at most $T_0 = \mathcal{O}(1)$ iterations with $n = \Omega\left(n \vee \log \frac{1}{\delta}\right)$ samples, then $\alpha^{T_0} < 0.1$ with probability at least $1 - \delta$.*

**Remark.** Fact 6.2 (see proof in Appendix G, Subsection G.1) is established in a similar way as Fact 5.2 at the population level, but requires selecting a sufficiently large number of samples $n = \Omega(d \vee \log \frac{1}{\delta})$ to guarantee that the statistical error is small enough.

**Proposition 6.3.** *(Statistical Error of Mixing Weights) Let $N(\theta, \nu)$ and $N_n(\theta, \nu)$ be the EM update rules for mixing weights $\tanh(\nu) := \pi(1) - \pi(2)$ at the population level and finite-sample level with $n$ samples, and $n \gtrsim \log \frac{1}{\delta}$, $\delta \in (0, 1)$, then*

$$|N_n(\theta, \nu) - N(\theta, \nu)| = \min\left\{\frac{\|\theta\|/\sigma}{1 + |\nu|}, 1\right\} \mathcal{O}\left(\sqrt{\frac{\log \frac{1}{\delta}}{n}}\right)$$

*with probability at least $1 - \delta$.*

**Remark.** Proposition 6.3 (see proof in Appendix F, Subsection F.1) is proved by applying the elementary inequality for tanh and the concentration inequality, which is based on modified log-Sobolev inequality in Ledoux (2001) (see Appendix E, Subsection E.1 and Subsection E.2).

**Proposition 6.4.** *(Statistical Error of Regression Parameters) Let $M(\theta, \nu)$ and $M_n(\theta, \nu)$ be the EM update rules for regression parameters $\theta$ at the population and finite-sample levels with $n$ samples, and if $n \gtrsim d \vee \log \frac{1}{\delta}$, $\delta \in (0, 1)$, then with probability at least $1 - \delta$,*

$$\|M_n(\theta, \nu) - M(\theta, \nu)\|/\sigma = \mathcal{O}\left(\sqrt{\frac{d \vee \log \frac{1}{\delta}}{n}}\right),$$

*and if $n \gtrsim d \vee \log \frac{1}{\delta} \vee \log^3 \frac{1}{\delta'}$, $\delta' \in (0, 1)$, then with probability at least $1 - (\delta + \delta')$,*

$$\|M_n(\theta, \nu) - M(\theta, \nu)\|/\sigma = \{\tanh |\nu| + \|\theta\|/\sigma\} \mathcal{O}\left(\sqrt{\frac{d \vee \log \frac{1}{\delta}}{n}}\right).$$

**Remark.** Proposition 6.4 (see proof in Appendix F, Subsection F.4) demonstrates our sample complexity $n = \Omega(d \vee \log \frac{1}{\delta})$, whereas Lemma 1 of Dwivedi et al. (2020b) presents a sample complexity $n = \Omega(d \log \frac{1}{\delta})$. The difference lies in the techniques used: their multiplicative log factor is derived through the standard symmetrization technique with Rademacher variables and the application of the Ledoux-Talagrand contraction inequality (see the proof of Lemma 1 on page 44 of Dwivedi et al. (2020b) and Appendix E of Kwon et al. (2021) for Lemma 11). In contrast, we obtain our additive log factor by applying the rotational invariance of Gaussians and expressing the $\ell_2$ norm of the orthogonal error as the geometric mean of two Chi-square distributions (see Appendix E, Subsection E.3).

## 7 Discussions on Extensions

In this section, we discuss the differences between results of 2MLR and 2GMM, extend our analysis from the overspecified setting to the low-SNR regime, and examine the challenges of analyzing overspecified mixture models with multiple components.

## 7.1 Convergence and Sample Complexity: 2MLR vs 2GMM

In the overspecified setting (signal-to-noise-ratio (SNR) $\eta := \|\theta^*\|/\sigma \to 0$), the population EM update rules for 2MLR are as follows (see equation 10) for the normalized regression parameters $\alpha^t = \|\theta^t\|/\sigma$ and imbalance of mixing weights $\beta^t = \tanh(\nu^t) = \pi^t(1) - \pi^t(2)$:

$$\alpha^{t+1} = \mathbb{E}[\tanh(\alpha^t X + \nu^t)X], \quad \beta^{t+1} = \mathbb{E}[\tanh(\alpha^t X + \nu^t)] \quad \text{where } X \sim \frac{K_0(|x|)}{\pi}$$

In contrast, for 2GMM, the population EM update rules are (see page 6 of Weinberger & Bresler (2022)):

$$\alpha^{t+1} = \mathbb{E}[\tanh(\alpha^t Z + \nu^t)Z], \quad \beta^{t+1} = \mathbb{E}[\tanh(\alpha^t Z + \nu^t)] \quad \text{where } Z \sim \mathcal{N}(0,1)$$

Here, $K_0$ denotes the modified Bessel function of the second kind, the density function of $X$ has exponential tail since $K_0(x) \approx \sqrt{\frac{\pi}{2x}}\exp(-x)$ for $x \to \infty$, and $Z$ follows a standard normal distribution, which is sub-Gaussian (Wainwright, 2019). At the population level, we still can establish the sublinear convergence rates of $\alpha^t = \mathcal{O}(1/\sqrt{t})$ with balanced initial mixing weights ($\beta^0 = 0$), and linear convergence rates with unbalanced initial mixing weights ($\beta^0 \neq 0$) for both 2MLR and 2GMM by bounding the above expectations.

However, at the finite-sample level, convergence guarantees differ between 2MLR and 2GMM. For 2MLR, Theorem 6.1 indicates that a sample size $n = \Omega(d \vee \log^3 \frac{1}{\delta})$ is sufficient to ensure convergence. In contrast, for 2GMM, we can show that a smaller sample size $n = \Omega(d \vee \log \frac{1}{\delta})$ is required to achieve the same probability $1 - T\delta$ over $T$ iterations. Intuitively, the difference arises because, for finite $n$, the sample averages $\frac{1}{n}\sum_{i=1}^{n}\tanh(\alpha Z_i + \nu)Z_i, \frac{1}{n}\sum_{i=1}^{n}\tanh(\alpha Z_i + \nu)$ converge to their expectation more rapidly when $\{Z_i\}_{i=1}^{n}$ are sub-Gaussian (in 2GMM) compared to when they have an exponential tail in probability (in 2MLR). This faster convergence in the sub-Gaussian case allows for reliable parameter estimation with fewer samples.

## 7.2 Extended Analysis in Low SNR Regime

To further extend our analysis from the limiting case of $\eta := \|\theta^*\|/\sigma = 0$ to the case of finite low SNR ($\eta \lesssim 1$) of Mixed Linear Regressions (MLR), we can still obtain the recurrence relations of $\alpha^t, \beta^t$ as follows:

$$\begin{aligned}
\alpha^{t+1} &= \mathbb{E}[\tanh(\alpha^t X + \nu^t)X] + \eta\beta^*\rho^t\mathbb{E}[\tanh(\alpha^t X + \nu^t)X^2] + \mathcal{O}(\eta^2), \\
\beta^{t+1} &= \mathbb{E}[\tanh(\alpha^t X + \nu^t)] + \eta\beta^*\rho^t\mathbb{E}[\tanh(\alpha^t X + \nu^t)X] + \mathcal{O}(\eta^2),
\end{aligned} \tag{12}$$

where $\beta^* = \tanh(\nu^*)$ is the imbalance of mixing weights of the ground truth, and $\rho^t = \langle\theta^*, \theta^t\rangle/(\|\theta^*\| \cdot \|\theta^t\|)$ is the cosine angle between the ground truth and the estimated regression parameters at $t$-th iteration.

While equation 10 gives the EM update rules of $\alpha^t, \beta^t$ in the overspecified setting of $\eta = 0$, we can also obtain the EM update rules of $\alpha^t, \beta^t$ in the low SNR regime (equation 12) with the help of introducing $\eta, \rho^t$ by using the perturbation method as discussed in Appendix H. The differences between the EM update rules of $\alpha^t, \beta^t$ in the finite low SNR regime $\eta \lesssim 1$ and the EM update rules of $\alpha^t, \beta^t$ in the overspecified setting $\eta = 0$ come from the presence of $\eta, \rho^t$ in the EM update rules. In the overspecified setting, the EM update rules of $\alpha^t, \beta^t$ are given by equation 10 with $\eta = 0$ and are independent of $\rho^t$, where the cosine angle $\rho^t = \rho^0$ remains the same as the initial value $\rho^0$ as shown in Identity 4.1 and Figure 1a. But in the finite low SNR regime $\eta \lesssim 1$, the EM update rules of $\alpha^t, \beta^t$ are given by equation 12 with $\eta$ and $\rho^t$, where the cosine angle $\rho^t$ is updated by the EM update rules of $\rho^t$ as follows when $\eta$ is sufficiently small (see details in Appendix H, Subsection H.3):

$$\rho^{t+1} = \rho^t + (1 - [\rho^t]^2) \cdot \eta\beta^*(\mathbb{E}[\tanh(\alpha^t X + \nu^t)] - \alpha^t\mathbb{E}[\tanh^2(\alpha^t X + \nu^t)X])/\mathbb{E}[\tanh(\alpha^t X + \nu^t)X] + \mathcal{O}(\eta^2). \tag{13}$$

**Remark.** The equation 12 and equation 13 (see proof in Appendix H, Subsection H.3) are proved by using the perturbation method as discussed in Appendix H, Subsection H.2. Moreover, the expectations $\mathbb{E}[\tanh(\alpha^t X + \nu^t)X^2], \mathbb{E}[\tanh(\alpha^t X + \nu^t)X], \mathbb{E}[\tanh(\alpha^t X + \nu^t)]$ and $\mathbb{E}[\tanh^2(\alpha^t X + \nu^t)X]$ in equation 12 and equation 13 can be approximated by series expansions of $\alpha^t$ around $\alpha^t = 0$ as in Appendix H, Subsection H.1. The remainder terms of $\mathcal{O}(\eta^2)$ in equation 12 and equation 13 hide the effect of $\alpha^t, \beta^t, \rho^t$ in the EM update rules. Consequently, by substituting the series expansions of the expectations into equation 12 and equation 13,

we can obtain the following approximate dynamic equations of $\alpha^t, \beta^t, \rho^t$ in the finite low SNR regime:

$$\begin{pmatrix} \alpha^{t+1} \\ \beta^{t+1} \\ \rho^{t+1} \end{pmatrix} = \begin{pmatrix} \alpha^t(1 - [\beta^t]^2) \\ \beta^t(1 - [\alpha^t]^2(1 - [\beta^t]^2)) \\ \rho^t \end{pmatrix} + \eta\beta^* \times \begin{pmatrix} \rho^t\beta^t(1 - 9[\alpha^t]^2(1 - [\beta^t]^2)) \\ \rho^t\alpha^t(1 - [\beta^t]^2) \\ (1 - [\rho^t]^2)\frac{\beta^t(1-6[\alpha^t]^2[\beta^t]^2)}{\alpha^t(1-[\beta^t]^2)} \end{pmatrix} + \mathcal{O}([\alpha^t]^3) + \mathcal{O}(\eta^2). \quad (14)$$

In particular, when $\eta = 0$, we have $\rho^{t+1} = \rho^t$ by Identity 4.1, and the above approximate dynamic equations reduce to the approximate dynamic equations of $\alpha^t, \beta^t$ in the overspecified setting, which aligns with our previous results obtained in Proposition 4.4. Also, a finer analysis shows that $|\rho^t| = 1$ implies $|\rho^{t+1}| = 1$ even when $\eta \neq 0$ (see the remark of Appendix H, Subsection H.3), therefore the EM iterations of regression parameters have the same direction as the ground truth if the initial value of the regression parameters is in the same direction as the ground truth.

### 7.3 Generalizing to Multiple Components

As shown in the numerical experiments of Dwivedi et al. (2020b), the EM algorithm for overspecified Gaussian mixtures with multiple components can also exhibit the slow convergence of final accuracy in terms of sample size $n$. The order of the final accuracy $\mathcal{O}((d/n)^{1/4})$ demonstrates slower convergence in the sufficiently balanced case compared to the final accuracy $\mathcal{O}((d/n)^{1/2})$ in the sufficiently unbalanced case for 2MLR and 2GMM in the overspecified setting, which is not merely a coincidence, but a general phenomenon. However, it remains an open problem to establish the convergence guarantees for the final accuracy, time complexity and sample complexity for overspecified mixture models with multiple components. As for a general overspecified model with multiple components, it is necessary to carefully examine the many-to-one correspondence between the components of the fitted model and those of the ground truth (see page 6 of Qian et al. (2022)). Therefore, the assumption of well-separated regression parameters and the requirement for a good initialization of the mixing weights, as in existing works such as Kwon & Caramanis (2020), no longer hold. This requires a more careful analysis and development of more advanced techniques for the analysis of overspecified mixture models with multiple components.

## 8 Experiments

In this section, we validate our theoretical findings in the previous sections with numerical experiments. The code for our numerical experiments is available at https://github.com/dassein/em_overspecified_mlr.

**Trajectory of EM Iterations.** We sample 2,000 independent and identically distributed (i.i.d.) two-dimensional covariates and additive noises from Gaussian distributions and set the true regression parameters $\theta^* = \vec{0}$. In Fig. 1a, all the iterations are nearly perfect rays from the origin to the initial point, which aligns with Identity 4.1.

**Dynamics of the EM iteration.** In the overspecified setting, we show that the approximate dynamic equations $(\alpha^t - \alpha^{t+1})/\alpha^t \approx [\beta^t]^2$ and $(\beta^t - \beta^{t+1})/\beta^t \approx \alpha^t\alpha^{t+1}$ when the regression parameters are small enough in Proposition 4.4. We demonstrate the linear correlations in Fig. 2, therefore, our experiments validate Proposition 4.4. For the experimental settings, we specify $\alpha^t = 0.1$ and consider different values of $\beta^t \in \{0.1, 0.2, \cdots, 0.9, 1\}$.

**Convergence Rate in Regression Parameters.** Fig. 3a presents the fast convergence with unbalanced initial guess and the slow convergence with balanced initial guess, which is in agreement with our theoretical results in Theorem 5.1.

**Sublinear Convergence Rate with Balanced Initial Guess.** Fig. 4a and Fig. 4b exhibit the slow sublinear convergence rate given the balanced initial guess, which aligns with our sublinear theoretical bounds. Hence, our experimental results validate our analysis in Proposition 5.3.

**Initialization Phase in the Worst Case.** In Fig. 1b, we show the iterations of $\alpha^t$ in the worst case with balanced initial mixing weights $\pi^0 = (\frac{1}{2}, \frac{1}{2})$, i.e., $\beta^0 = \tanh\nu^0 = \frac{1}{2} - \frac{1}{2} = 0$, and infinite initial regression parameters $\alpha^0 = \|\theta^0\|/\sigma \to \infty$. In the worst case, we show that $\alpha^t \geq 0.1$ for all $t \leq 9$ by using the theoretical

matching lower bound for the worst case in Proposition 5.3. Also, we demonstrate that $\alpha^t < 0.1$ for all $t \geq 36$ (Fact 5.2) by applying the theoretical upper bound in Proposition 5.3. As $\alpha^{20} \approx 0.1$ by numerical evaluations, and $20 > 9$ and $20 < 36$, the theoretical results from Proposition 5.3 and Fact 5.2 are consistent with the numerical results shown in Fig. 1b.

**Converged Mixing Weights and Initial Guesses.** In Proposition 5.4, we prove that converged mixing weights $\beta^T$ are nonzero for a nonzero initial guess $\beta^0 \neq 0$, and provide a lower bound for $\beta^T$. We observe the correlation between the converged $\beta^T$ and varying $\beta^0$ uniformly sampled from $[0.01, 0.99]$, with $\alpha^0 \in \{0.1, 0.3, 0.5\}$. These theoretical findings are supported by numerical results in Fig. 3b.

## 9    Conclusions

In this paper, we thoroughly investigated the EM's behavior in overspecified two-component Mixed Linear Regression (2MLR) models. We rigorously characterized the EM estimates for both regression parameters and mixing weights by providing the approximate dynamic equations (Proposition 4.4) for the evolution of EM estimates and establishing the convergence guarantees (Theorems 5.1, 6.1) for the final accuracy, time complexity, and sample complexity at population and finite-sample levels, respectively. Notably, with an unbalanced initial guess for mixing weights, we showed linear convergence of regression parameters in $\mathcal{O}(\log(1/\epsilon))$ steps. Conversely, with a balanced initial guess, sublinear convergence occurs in $\mathcal{O}(\epsilon^{-2})$ steps to achieve $\epsilon$-accuracy. For mixtures with sufficiently imbalanced fixed mixing weights $\|\pi^t - \frac{1}{2}\|_1 \gtrsim \mathcal{O}((d/n)^{1/4})$, we establish statistical accuracy $\mathcal{O}((d/n)^{1/2})$, whereas for those with sufficiently balanced fixed mixing weights $\|\pi^t - \frac{1}{2}\|_1 \lesssim \mathcal{O}((d/n)^{1/4})$, the accuracy is $\mathcal{O}((d/n)^{1/4})$. Additionally, our novel analysis sharpens bounds for statistical error, time complexity, and sample complexity needed to achieve a final statistical accuracy of $\mathcal{O}((d/n)^{1/4})$ with fixed sufficiently balanced mixing weights. Furthermore, we discussed the differences between results of 2MLR and 2GMM, and extended our analysis from the overspecified setting to the finite low SNR regime.

Building upon the analysis and established connections between the diffusion model objective and the classic EM algorithm in GMM (Shah et al., 2023), we foresee extending this analysis from GMM to MLR and establishing the time and sample complexities involved in learning the diffusion model objective, as discussed in recent works (Chen et al., 2024a; Gatmiry et al., 2024). The practical applications (haplotype assembly (Cai et al., 2016; Sankararaman et al., 2020) and phase retrieval (Klusowski et al., 2019; Chen et al., 2015)) of mixture models such as MLR and GMM can spur significant interest within the statistics community toward establishing rigorous theoretical foundations for generative diffusion models.

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

# Appendices

We organize the Appendices as follows:

- Appendix A, we prepare lemmas for identities of hyperbolic functions, inequalities of tanh and simple expectations under the density involving $K_0$, and lower/upper bounds for expectations with tanh under the density involving $K_0$.

- Appendix B, we provide proofs for EM update rules $M(\theta, \nu), N(\theta, \nu)$ at the population level, the nonincreasing property of $\|\theta^t\|, \|\pi^t - \frac{1}{2}\|_1$ and boundness of $\|\theta^t\|$, and the approximate dynamic equations of $\|M(\theta^t, \nu^t)\|/\sigma, N(\theta^t, \nu^t)$ when $\|\theta\|/\sigma$ is small.

- Appendix C, we present lemmas on the evolution of $\alpha^t := \|\theta^t\|/\sigma, \beta^t = \pi^t(1) - \pi^t(2)$ in population EM, providing lower/upper bounds for $\alpha^{t+1}/(\alpha^t(1 - [\beta^t]^2)), \alpha^{t+1}/\alpha^t$ and $\beta^{t+1}/\beta^t$.

- Appendix D, we provide proofs for convergence rates of regression parameters $\|\theta\|$ at the population level. For $\pi^0 = (\frac{1}{2}, \frac{1}{2})$, we show sublinear convergence rate with time complexity $\mathcal{O}(\epsilon^{-2})$ to achieve $\epsilon$-accuracy. while for $\pi^0 \neq (\frac{1}{2}, \frac{1}{2})$, we demonstrate linear convergence rate with time conplexity $\mathcal{O}(\log \frac{1}{\epsilon})$. We also establish the existence of $\beta^\infty = \lim_{t \to \infty} \beta^t$ and give an upper bound with $\beta^\infty$ for the contraction factor of linear convergence.

- Appendix E, we develop lemmas to bound errors at the finite-sample level by establishing tighter concentration inequalities using the modified log-Sobolev inequality, bounding statistics, and deriving inequalities for tanh.

- Appendix F, we establish bounds for the projected error and statistical error in regression parameters for both easy EM and standard finite-sample EM, as well as for the statistical error in mixing weights.

- Appendix G, we provide proofs for convergence rates at finite-sample level. For mixtures with sufficiently imbalanced fixed mixing weights $\|\pi^t - \frac{1}{2}\|_1 \gtrsim \mathcal{O}((d/n)^{1/4})$, the statistical accuracy is $\mathcal{O}((d/n)^{1/2})$, whereas for those with sufficiently balanced fixed mixing weights $\|\pi^t - \frac{1}{2}\|_1 \lesssim \mathcal{O}((d/n)^{1/4})$, the accuracy is $\mathcal{O}((d/n)^{1/4})$.

- Appendix H, we extend our analysis to the finite low SNR regime ($\eta = \|\theta^*\|/\sigma \lesssim 1$), and provide the approximate dynamic equations for EM iterations of $\alpha^t, \beta^t$ and the cosine angle $\rho^t := \langle \theta^t, \theta^* \rangle / \|\theta^t\| \|\theta^*\|$ in low SNR regime.

# A    Lemmas in Proofs for Results of Population EM Updates

## A.1    Identities of hyperbolic functions and expectations under the density involving $K_0$

**Lemma A.1.** (Identities of Hyperbolic Functions in Polyanin & Manzhirov (2008))

$$\frac{\tanh(a+b) - \tanh(-a+b)}{2} = \frac{[1 - \tanh^2(b)]\tanh(a)}{1 + \tanh^2(b)\tanh^2(a)}, \frac{\tanh(a+b) + \tanh(-a+b)}{2} = \frac{\tanh(b)[1 - \tanh^2(a)]}{1 - \tanh^2(b)\tanh^2(a)}.$$

*Proof.* The first identity is proved by using the identities of Hyperbolic Functions in Supplement 1, Page 698 of Polyanin & Manzhirov (2008). The second one is proved by letting $a \leftarrow b, b \leftarrow a$ in the first one. $\quad\square$

**Lemma A.2.** Let $m(\alpha, \nu) := \mathbb{E}[\tanh(\alpha X + \nu)X], n(\alpha, \nu) := \mathbb{E}[\tanh(\alpha X + \nu)]$ be the expectations with respect to $X \sim f_X(x) = K_0(|x|)/\pi$, a random variable with probability density involving the Bessel function $K_0$, and $\beta := \tanh(\nu)$, then $m(\alpha, \nu) = \frac{\partial n(\alpha, \nu)}{\partial \alpha}$ and

$$m(\alpha, \nu) = (1 - \beta^2)\mathbb{E}\left[\frac{\tanh(\alpha X)X}{1 - \beta^2 \tanh^2(\alpha X)}\right], \quad n(\alpha, \nu) = \beta\mathbb{E}\left[\frac{1 - \tanh^2(\alpha X)}{1 - \beta^2 \tanh^2(\alpha X)}\right]$$

*Proof.* By applying Leibniz integral rule or invoking the dominated convergence theorem to exchange the order of taking limit and taking the expectations (see Theorem 1.5.8, page 24 of Durrett (2019)), we obtain the relation $m(\alpha, \nu) = \frac{\partial n(\alpha, \nu)}{\partial \alpha}$. Since $X \sim f_X(x) = \frac{K_0(|x|)}{\pi}$ is a symmetric random variable, we have:

$$\mathbb{E}[f(X)] = \mathbb{E}[f(-X)] = \mathbb{E}[[f(X) + f(-X)]\mathbb{1}_{X \geq 0}] = \frac{1}{2}\mathbb{E}[[f(X) + f(-X)]].$$

By substituting $f(X)$ with $\tanh(\alpha X + \nu), \tanh(\alpha X + \nu)X$, and applying the above proven identities of hyperbolic functions, these two identities are proved. $\quad\square$

**Lemma A.3.** (Expectations in Gradshteyn & Ryzhik (2014)) For $\alpha \in [0, 1), n \in \mathbb{Z}_+$, and a random variable $X \sim \frac{K_0(|x|)}{\pi}$, then we have:

$$\mathbb{E}[\exp(-\alpha|X|)] = \frac{2}{\pi}\frac{\arccos\alpha}{\sqrt{1 - \alpha^2}}, \quad \mathbb{E}[\cosh(\alpha X)] = \frac{1}{\sqrt{1 - \alpha^2}}, \quad \mathbb{E}[|X|] = \frac{2}{\pi}, \quad \mathbb{E}[X^{2n}] = [(2n - 1)!!]^2,$$

where $(2n - 1)!! = 1 \times 3 \times 5 \cdots \times (2n - 1)$.

*Proof.* We prove the first and second identities by taking the limit $\nu \to 0$ in the third formula in table 6.611 with modified Bessel function $K_\nu$, Section 6.61 Combinations of Bessel functions and exponentials, Page 703 of Gradshteyn & Ryzhik (2014), and invoking $\cosh(\alpha x) = (\exp(\alpha x) + \exp(-\alpha x))/2$.

We prove the third and fourth identities by letting $\nu = 0$ in the 16th formula in table 6.561, Section 6.56-6.58 Combinations of Bessel functions and powers, Page 685 of Gradshteyn & Ryzhik (2014), and invoking $\Gamma(n + \frac{1}{2}) = \sqrt{\pi}2^{-n}(2n - 1)!!$. $\quad\square$

**A.2 Inequalities of** $\tanh$ **and simple expectations under the density involving** $K_0$

**Lemma A.4.** For $t \in \mathbb{R}_{\geq 0}$, we have:

$$
\begin{aligned}
t - \frac{t^3}{3} &\leq & \tanh(t) & \leq t - \frac{t^3}{3}\exp(-t) \\
t^2 - \frac{2}{3}t^4 &\leq & \tanh^2(t) & \leq t^2 - \frac{2}{3}t^4\exp(-t) \\
t^3 - t^5 &\leq & \tanh^3(t) & \leq t^3 - t^5\exp(-t) \\
t^4 - \frac{4}{3}t^6 &\leq & \tanh^4(t) & \leq t^4 - \frac{4}{3}t^6\exp(-t)
\end{aligned}
$$

*Proof.* By using Taylor expansion with remainder (see 5.15 Theorem, pages 110-111 of Rudin (1976)) for $\tanh(t), \tanh^2(t), \tanh^3(t), \tanh^4(t)$ and dropping out the non-negative remainder, we obtain the lower bounds for $\tanh(t), \tanh^2(t), \tanh^3(t), \tanh^4(t)$. The upper bounds for $\tanh(t), \tanh^2(t), \tanh^3(t), \tanh^4(t)$ are established by introducing the exponential decay in the terms of $t^3, t^4, t^5, t^6$ respectively, where the analysis on the monotonicity of the gaps between the upper bounds and $\tanh(t), \tanh^2(t), \tanh^3(t), \tanh^4(t)$ ensures the correctness of the upper bounds when $t$ is small, and the fact that $\tanh(t)$ is bounded but the upper bounds are unbounded as $t$ increases ensures the correctness when $t$ is large. $\square$

**Lemma A.5.** For any $\nu, t \in \mathbb{R}$, we have:

$$
1 + \tanh^2(\nu)\tanh^2(t) \leq \frac{1}{1 - \tanh^2(\nu)\tanh^2(t)} \leq 1 + \tanh^2(\nu)\sinh^2(t)
$$

*Proof.* By substituting $\tanh^2(\nu)\tanh^2(t) \to x$ into $1 + x \leq \frac{1}{1-x}$, we otain the lower bound for $\frac{1}{1-\tanh^2(\nu)\tanh^2(t)}$. By substituting $\tanh^2(\nu) \to r, \sinh^2(t) \to x$ into $1 \leq 1 + (r - r^2)\frac{x^2}{1+x} = (1 - r\frac{x}{1+x})(1 + rx), \forall r \in [0,1], x \in \mathbb{R}_{\geq 0}$, we establish the upper bound for $\frac{1}{1-\tanh^2(\nu)\tanh^2(t)}$. $\square$

**Lemma A.6.** For $\alpha \in [0, 0.31)$, and a random variable $X \sim \frac{K_0(|x|)}{\pi}$, we have:

$$
\begin{aligned}
\frac{9}{1+8\alpha} &\leq & \frac{\mathrm{d}^4}{\mathrm{d}\alpha^4}\mathbb{E}[\exp(-\alpha|X|)] = \mathbb{E}[X^4\exp(-\alpha|X|)] \\
\frac{225}{1+16\alpha} &\leq & \frac{\mathrm{d}^6}{\mathrm{d}\alpha^6}\mathbb{E}[\exp(-\alpha|X|)] = \mathbb{E}[X^6\exp(-\alpha|X|)] \\
\frac{18\alpha^2}{1 - \frac{25}{3}\alpha^2} &\geq & \frac{\mathrm{d}^2}{\mathrm{d}(2\alpha)^2}\mathbb{E}[\cosh(2\alpha X)] - \mathbb{E}[X^2] = \mathbb{E}[X^2(\cosh(2\alpha X) - 1)]
\end{aligned}
$$

*Proof.* By Leibniz integral rule or the dominated convergence theorem to exchange the order of taking limit and taking expectations (see Theorem 1.5.8, page 24 of Durrett (2019)), we obtain the relations between the right-hand side expectations and the simple expectations as shown above. Therefore, we obtain the closed form expressions for expectations on the right-hand side by taking derivatives of the closed-form expressions of simple expectations, which are given in Lemma of Subsection A.1. Consequently, we directly give the lower/upper bounds of rational functions on the left-hand side for these closed form expressions based on their Padé approximants, verified by analyzing the monotonicity, convexity of the gaps between the rational functions and the closed form expressions, and evaluating values at the boundaries of the interval of $\alpha$. $\square$

**A.3 Lower/Upper bounds for expectations with** $\tanh$ **under the density involving** $K_0$

**Lemma A.7.** Let $n(\alpha, \nu) := \mathbb{E}[\tanh(\alpha X + \nu)]$ be the expectation with respect to a random variable with density $X \sim K_0(|x|)/\pi$, suppose $\alpha \in [0, 0.31)$ and $\beta := \tanh(\nu) \geq 0$, then

$$
\begin{aligned}
n(\alpha, \nu) &\geq & \beta \cdot \left\{ 1 - \alpha^2(1 - \beta^2) + \alpha^4\left[(1 - \beta^2) \times \frac{6}{1+8\alpha} - \beta^2 \times 9\right] + \alpha^6\beta^2 \times \frac{300}{1+16\alpha} \right\} \\
n(\alpha, \nu) &\leq & \beta \cdot \left\{ 1 - \alpha^2(1 - \beta^2) + \alpha^4(1 - \beta^2) \times 6 \right\}
\end{aligned}
$$

*Proof.* We obtain the following lower/upper bounds for $n(\alpha, \nu)$ by applying the Lemmas of indentities in Subsection A.1 and invoking the Lemmas of inequalities in Subsection A.2.

$$
\begin{aligned}
n(\alpha, \nu) &= \beta \mathbb{E}\left[\frac{1 - \tanh^2(\alpha X)}{1 - \beta^2 \tanh^2(\alpha X)}\right] \\
&\geq \beta \mathbb{E}[(1 - \tanh^2(\alpha X))\{1 + \beta^2 \tanh^2(\alpha X)\}] = \beta \mathbb{E}[1 - (1 - \beta^2)\tanh^2(\alpha X) - \beta^2 \tanh^4(\alpha X)] \\
&\geq \beta \left\{1 - (1 - \beta^2)\mathbb{E}\left[(\alpha X)^2 - \frac{2}{3}(\alpha X)^4 \exp(-\alpha|X|)\right] - \beta^2 \mathbb{E}\left[(\alpha X)^4 - \frac{4}{3}(\alpha X)^6 \exp(-\alpha|X|)\right]\right\} \\
&\geq \beta \left\{1 - (1 - \beta^2)\left[\alpha^2 - \alpha^4 \times \frac{6}{1 + 8\alpha}\right] - \beta^2 \left[\alpha^4 \times 9 - \alpha^6 \times \frac{300}{1 + 16\alpha}\right]\right\} \\
n(\alpha, \nu) &\leq \beta \mathbb{E}\left[(1 - \tanh^2(\alpha X))\{1 + \beta^2 \sinh^2(\alpha X)\}\right] = \beta \mathbb{E}[1 - (1 - \beta^2)\tanh^2(\alpha X)] \\
&\leq \beta \left\{1 - (1 - \beta^2)\mathbb{E}\left[(\alpha X)^2 - \frac{2}{3}(\alpha X)^4\right]\right\} = \beta \left\{1 - (1 - \beta^2)\left[\alpha^2 - \alpha^4 \times 6\right]\right\}
\end{aligned}
$$

$\square$

**Lemma A.8.** Let $m(\alpha, \nu) := \mathbb{E}[\tanh(\alpha X + \nu)X]$ be the expecatation with respect to a random variable with density $X \sim K_0(|x|)/\pi$, suppose $\alpha \in [0, 0.31)$ and $\beta := \tanh(\nu) \geq 0$, then

$$
\begin{aligned}
m(\alpha, \nu) &\geq \alpha(1 - \beta^2) \cdot \left\{1 - \alpha^2 \left[3 - \beta^2 \times 9\right] - \alpha^4 \beta^2 \times 225\right\} \\
m(\alpha, \nu) &\leq \alpha(1 - \beta^2) \cdot \left\{1 - \alpha^2 \left[\frac{3}{1 + 8\alpha} - \beta^2 \times \frac{9}{1 - \frac{25}{3}\alpha^2}\right] - \alpha^4 \beta^2 \times \frac{\frac{225}{3}}{1 + 16\alpha}\right\}
\end{aligned}
$$

*Proof.* We obtain the following lower/upper bounds for $m(\alpha, \nu)$ by applying Lemmas of indentities in Subsection A.1, invoking Lemmas of inequalities in Subsection A.2 and noting that $\sinh^2(t) = \frac{\cosh(2t) - 1}{2} \geq t^2$.

$$
\begin{aligned}
m(\alpha, \nu) &= (1 - \beta^2)\mathbb{E}\left[\frac{\tanh(\alpha X)X}{1 - \beta^2 \tanh^2(\alpha X)}\right] \\
&\geq (1 - \beta^2)\mathbb{E}\left[\tanh(\alpha X)X\{1 + \beta^2 \tanh^2(\alpha X)\}\right] = (1 - \beta^2)\mathbb{E}\left[|X|\tanh(\alpha|X|) + \beta^2|X|\tanh^3(\alpha|X|)\right] \\
&\geq (1 - \beta^2)\left\{\mathbb{E}\left[|X|\left((\alpha|X|) - \frac{1}{3}(\alpha|X|)^3\right)\right] + \beta^2 \mathbb{E}\left[|X|\left((\alpha|X|)^3 - (\alpha|X|)^5\right)\right]\right\} \\
&= (1 - \beta^2)\left\{\alpha - \alpha^3 \times 3 + \beta^2(\alpha^3 \times 9 - \alpha^5 \times 225)\right\} \\
m(\alpha, \nu) &\leq (1 - \beta^2)\mathbb{E}\left[\tanh(\alpha|X|)|X|\{1 + \beta^2 \sinh^2(\alpha X)\}\right] \\
&\leq (1 - \beta^2)\mathbb{E}\left[\left((\alpha|X|) - \frac{1}{3}(\alpha|X|)^3 \exp(-\alpha|X|)\right)|X|(1 + \beta^2 \sinh^2(\alpha X))\right] \\
&\leq (1 - \beta^2)\mathbb{E}\left[|X|\left((\alpha|X|) - \frac{1}{3}(\alpha|X|)^3 \exp(-\alpha|X|)\right)\right] \\
&+ (1 - \beta^2)\beta^2 \mathbb{E}\left[|X|(\alpha|X|)\frac{(\cosh(2\alpha|X|) - 1)}{2} - \frac{1}{3}|X|(\alpha|X|)^5 \exp(-\alpha|X|)\right] \\
&= (1 - \beta^2)\left\{\alpha - \alpha^3 \times \frac{3}{1 + 8\alpha} + \beta^2\left(\alpha^3 \times \frac{9}{1 - \frac{25}{3}\alpha^2} - \alpha^5 \times \frac{\frac{225}{3}}{1 + 16\alpha}\right)\right\}
\end{aligned}
$$

$\square$

**Lemma A.9.** Let $m_0(\alpha) := \mathbb{E}[\tanh(\alpha X)X]$ be the expectation with respect to a random variable with density $X \sim K_0(|x|)/\pi$, suppose $\alpha \in [0, 0.31)$, then

$$
\alpha - 3\alpha^3 \leq m_0(\alpha) \leq \alpha - \frac{3\alpha^3}{1 + 8\alpha}
$$

*Proof.* By letting $\beta := \tanh(\nu) = 0$ in the previous Lemma, the lower/upper bounds for $m_0(\alpha)$ are obtained.

$\square$

# B   Proofs for Results of Population EM Updates

## B.1   Proof for EM Update Rules

**Theorem B.1.** (Identity 4.1 in Section 4: EM Updates for Overspecified 2MLR) Suppose a 2MLR model is fitted to the overspecified model with no separation, where $\theta^* = \vec{0}$. The EM update rules at the population level for $\bar{\theta}^t := \theta^t/\sigma = M(\theta^{t-1}, \nu^{t-1})/\sigma$ and $\tanh(\nu^t) := \pi^t(1) - \pi^t(2) = N(\theta^{t-1}, \nu^{t-1})$ are then as follows.

$$\bar{\theta}^t = \frac{\bar{\theta}^0}{\|\bar{\theta}^0\|} \cdot \frac{1}{\pi} \int_{\mathbb{R}} \tanh(\|\bar{\theta}^{t-1}\| x - \nu^{t-1}) x K_0(|x|) \mathrm{d}x,$$

$$\tanh(\nu^t) = \frac{1}{\pi} \int_{\mathbb{R}} \tanh(\nu^{t-1} - \|\bar{\theta}^{t-1}\| x) K_0(|x|) \mathrm{d}x,$$

where $\bar{\theta}^0 := \theta^0/\sigma$.

*Proof.* In the overspecified setting, namely $\theta^* = \vec{0}$, the expectation in the EM update rules (equation 3, equation 4) becomes

$$\mathbb{E}_{s \sim p(s|\theta^*, \pi^*)} := \mathbb{E}_{x \sim \mathcal{N}(0, I_d)} \mathbb{E}_{y|x \sim \pi^*(1)\mathcal{N}(\langle x, \theta^*\rangle, \sigma^2) + \pi^*(2)\mathcal{N}(-\langle x, \theta^*\rangle, \sigma^2)} = \mathbb{E}_{x \sim \mathcal{N}(0, I_d)} \mathbb{E}_{y \sim \mathcal{N}(0, \sigma^2)}.$$

We decompose $x = \bar{x} \frac{\theta^{t-1}}{\|\theta^{t-1}\|} + x^\perp$, and $\bar{x} \sim \mathcal{N}(0, 1), x^\perp \in \mathrm{span}(\theta^{t-1})^\perp$ with $\mathbb{E}[x^\perp] = 0$. Since $\bar{x}, x^\perp, \bar{y} := y/\sigma = \varepsilon/\sigma$ are independent of each other, we obtain

$$\theta^t = \mathbb{E}_{x \sim \mathcal{N}(0, I_d)} \mathbb{E}_{y \sim \mathcal{N}(0, \sigma^2)} \tanh\left(\frac{y\langle x, \theta^{t-1}\rangle}{\sigma^2} + \nu^{t-1}\right) yx$$

$$= \sigma \frac{\theta^{t-1}}{\|\theta^{t-1}\|} \mathbb{E}_{\bar{x} \sim \mathcal{N}(0, 1)} \mathbb{E}_{\bar{y} \sim \mathcal{N}(0, 1)} \tanh\left(\bar{y}\bar{x} \cdot \frac{\|\theta^{t-1}\|}{\sigma} + \nu^{t-1}\right) \bar{y}\bar{x}$$

$$\tanh(\nu^t) = \mathbb{E}_{\bar{x} \sim \mathcal{N}(0, 1)} \mathbb{E}_{\bar{y} \sim \mathcal{N}(0, 1)} \tanh\left(\bar{y}\bar{x} \cdot \frac{\|\theta^{t-1}\|}{\sigma} + \nu^{t-1}\right)$$

It is well known that the Normal Product Distribution is $z := \bar{y}\bar{x} \sim \frac{K_0(|z|)}{\pi}$, see also Page 50, Section 4.4 Bessel Function Distributions, Chapter 12 Continuous Distributions (General) of Johnson et al. (1970) for more information.

$$\frac{\theta^t}{\sigma} = \frac{\theta^t}{\|\theta^t\|} \mathbb{E}_{z \sim \frac{K(|z|)}{\pi}} \tanh\left(z \cdot \frac{\|\theta^{t-1}\|}{\sigma} + \nu^{t-1}\right) z$$

$$\tanh(\nu^t) = \mathbb{E}_{z \sim \frac{K(|z|)}{\pi}} \tanh\left(z \cdot \frac{\|\theta^{t-1}\|}{\sigma} + \nu^{t-1}\right)$$

The above two equations immediately give the EM update rules and equation 10 is proved. Let $x := -z \sim \frac{K(|x|)}{\pi}$, consider the direction $\frac{\bar{\theta}^t}{\|\bar{\theta}^t\|}$, note that $\tanh\left(\nu^{t-1} + \|\bar{\theta}^{t-1}\| x\right) - \tanh\left(\nu^{t-1} - \|\bar{\theta}^{t-1}\| x\right) > 0$ for $\|\bar{\theta}^{t-1}\| \neq 0, x > 0$.

$$\int_{\mathbb{R}} \tanh\left(\|\bar{\theta}^{t-1}\| x - \nu^{t-1}\right) x \cdot K_0(|x|) \mathrm{d}x = \left[\int_0^\infty + \int_{-\infty}^0\right] \tanh\left(\|\bar{\theta}^{t-1}\| x - \nu^{t-1}\right) x \cdot K_0(|x|) \mathrm{d}x$$

$$= \int_0^\infty \left(\tanh\left(\nu^{t-1} + \|\bar{\theta}^{t-1}\| x\right) - \tanh\left(\nu^{t-1} - \|\bar{\theta}^{t-1}\| x\right)\right) x \cdot K_0(|x|) \mathrm{d}x > 0$$

Hence, we conclude that $\frac{\bar{\theta}^t}{\|\bar{\theta}^t\|} = \frac{\bar{\theta}^{t-1}}{\|\bar{\theta}^{t-1}\|} = \cdots = \frac{\bar{\theta}^0}{\|\bar{\theta}^0\|}$, thereby the proof is complete. $\square$

## B.2 Proof for Nonincreasing Property

**Theorem B.2.** (Fact 4.2 in Section 4: Monotonicity of Expectations) Let $m(\alpha, \nu) := \mathbb{E}[\tanh(\alpha X + \nu)X]$ and $n(\alpha, \nu) := \mathbb{E}[\tanh(\alpha X + \nu)]$ be the expectations with respect to $X \sim \frac{K_0(|x|)}{\pi}$. Then they satisfy the monotonicity properties:

(monotonicity of $m(\alpha, \nu)$): $m(\alpha, \nu)$ is a nonincreasing function of $\nu \geq 0$ for fixed $\alpha \geq 0$, and a nondecreasing function of $\alpha \geq 0$ for fixed $\nu \geq 0$, namely:

$$0 = m(\alpha, \infty) \leq m(\alpha, \nu') \leq m(\alpha, \nu) \leq m(\alpha, 0) \leq \alpha \text{ for } 0 \leq \nu \leq \nu', \alpha \geq 0,$$

$$0 = m(0, \nu) \leq m(\alpha, \nu) \leq m(\alpha', \nu) \leq m(\infty, \nu) = \frac{2}{\pi} \text{ for } 0 \leq \alpha \leq \alpha', \nu \geq 0.$$

(monotonicity of $n(\alpha, \nu)$): $n(\alpha, \nu)$ is a nonincreasing function of $\alpha \geq 0$ for fixed $\nu \geq 0$, and a nondecreasing function of $\nu \geq 0$ for fixed $\alpha \geq 0$, namely:

$$0 = n(\infty, \nu) \leq n(\alpha', \nu) \leq n(\alpha, \nu) \leq n(0, \nu) = \tanh(\nu) \text{ for } 0 \leq \alpha \leq \alpha', \nu \geq 0,$$

$$0 = n(\alpha, 0) \leq n(\alpha, \nu) \leq n(\alpha, \nu') \leq n(\alpha, \infty) = 1 \text{ for } 0 \leq \nu \leq \nu', \alpha \geq 0.$$

*Proof.* By using the identities of hyperbolic functions and expectations in Subsection A.1, and noting that $\frac{1-t^2}{1-r^2t^2}, \frac{t}{1-r^2t^2}$ are nonincreasing and nondecreasing respectively in $t \in [0, 1]$ for $r \in [0, 1]$ and $rt \neq 1$, we have

$$m(\alpha, \nu) \leq m(\alpha', \nu), \quad m(\alpha, \nu) \geq m(\alpha, \nu') \qquad \text{for } 0 \leq \alpha \leq \alpha', 0 \leq \nu \leq \nu'$$

$$n(\alpha, \nu) \leq n(\alpha', \nu), \quad n(\alpha, \nu) \geq n(\alpha, \nu') \qquad \text{for } 0 \leq \alpha \leq \alpha', 0 \leq \nu \leq \nu'$$

Moreover, we can invoke the dominated convergence theorem to exchange the order of taking limit and taking the expectations (see Theorem 1.5.8, page 24 of Durrett (2019)), since $\frac{1-x^2}{1-r^2x^2} \leq 1, \frac{x}{1-r^2x^2} \leq \frac{1}{1-r^2}$ are bounded in $x \in [0, 1]$ and therefore integrable. Note that $\mathbb{E}[|X|] = \frac{2}{\pi}, X \sim \frac{K_0(|x|)}{\pi}$ in Subsection A.1, we have

$$
\begin{aligned}
m(\infty, \nu) &= \lim_{\alpha \to \infty} \mathbb{E}[\tanh(\alpha X + \nu)X] = \mathbb{E}[\lim_{\alpha \to \infty} \tanh(\alpha X + \nu)X] = \mathbb{E}[|X|] = \frac{2}{\pi} \\
n(\alpha, \infty) &= \lim_{\nu \to \infty} \mathbb{E}[\tanh(\alpha X + \nu)] = \mathbb{E}[\lim_{\nu \to \infty} \tanh(\alpha X + \nu)] = \mathbb{E}[1] = 1
\end{aligned}
$$

Similarly, we have

$$
\begin{aligned}
m(\alpha, \infty) &= \lim_{\nu \to \infty} \mathbb{E}[\tanh(\alpha X + \nu)X] = \mathbb{E}[\lim_{\nu \to \infty} \tanh(\alpha X + \nu)X] = \mathbb{E}[X] = 0 \\
n(\infty, \nu) &= \lim_{\alpha \to \infty} \mathbb{E}[\tanh(\alpha X + \nu)] = \mathbb{E}[\lim_{\alpha \to \infty} \tanh(\alpha X + \nu)] = \mathbb{E}[\text{sgn}(X)] = 0
\end{aligned}
$$

Also, we have $m(0, \nu) = \tanh(\nu)\mathbb{E}[X] = 0, n(0, \nu) = \mathbb{E}[\tanh(\nu)] = \tanh(\nu), n(\alpha, 0) = \mathbb{E}[\tanh(\alpha X)] = 0$ and

$$m(\alpha, 0) = \mathbb{E}[\tanh(\alpha X)X] \leq \mathbb{E}[\alpha X^2] = \alpha \mathbb{E}[X^2] = \alpha, \quad \forall \alpha \geq 0.$$

Therefore, we have completed the proof by combining the above results. □

**Theorem B.3.** (Fact 4.3 in Section 4: Nonincreasing and Bounded)

Let $\alpha^t := \|\theta^t\|/\sigma = \|M(\theta^{t-1}, \nu^{t-1})\|/\sigma$ and $\beta^t := \tanh(\nu^t) = N(\theta^{t-1}, \nu^{t-1})$ for all $t \in \mathbb{Z}_+$ be the $t$-th iteration of the EM update rules $\|M(\theta, \nu)\|/\sigma$ and $N(\theta, \nu)$ at the population level, then $\beta^t \cdot \beta^0 \geq 0, \alpha^t \leq 2/\pi$, and $\{\alpha^t\}_{t=0}^\infty$ and $\{|\beta^t|\}_{t=0}^\infty$ are non-increasing.

*Proof.* Note that $X \sim \frac{K_0(|x|)}{\pi}$ is a symmetric random variable, we have that $m(\alpha, \nu) = m(\alpha, -\nu) = m(\alpha, |\nu|)$ is an even function of $\nu$ and $n(\alpha, \nu) = -n(\alpha, -\nu) = \text{sgn}(\nu)n(\alpha, |\nu|)$ is an odd function of $\nu$. By the recurrence relation of EM updates equation 10, we have $\alpha^{t+1} = m(\alpha^t, \nu^t), \beta^{t+1} = n(\alpha^t, \nu^t)$ with $\beta^t = \tanh(\nu^t)$.

By applying the previous proved monotonicity of $m(\alpha, \nu), n(\alpha, \nu)$ (Fact 4.2 in Section 4), we have

$$\beta^{t+1} \cdot \beta^t = [\text{sgn}(\nu^t)]^2 n(\alpha^t, |\nu^t|) \tanh|\nu^t| \geq 0, \quad |\beta^{t+1}| = n(\alpha^t, |\nu^t|) \leq n(0, |\nu^t|) = \tanh|\nu^t| = |\beta^t|$$

Therefore, by induction, we have $\beta^t \cdot \beta^0 \geq 0$ and $\{|\beta^t|\}_{t=0}^{\infty}$ is non-increasing. By the monotonicity of $m(\alpha, \nu)$,

$$0 \leq \alpha^{t+1} = m(\alpha^t, \nu^t) = m(\alpha^t, |\nu^t|) \leq m(\alpha^t, 0) \leq \alpha^t, \quad \alpha^{t+1} = m(\alpha^t, |\nu^t|) \leq m(\alpha^t, \infty) = \frac{2}{\pi}$$

Therefore, we have $\alpha^t \leq 2/\pi$ for all $t \in \mathbb{Z}_+$ and $\{\alpha^t\}_{t=0}^{\infty}$ is non-increasing. $\square$

## B.3 Proof for Approximate Dynamic Equations

**Theorem B.4.** (Proposition 4.4 in Section 4: Approximate Dynamic Equations)

Let $\alpha^t := \|\theta^t\|/\sigma = \|M(\theta^{t-1}, \nu^{t-1})\|/\sigma$ and $\beta^t := \tanh(\nu^t) = N(\theta^{t-1}, \nu^{t-1})$ for all $t \in \mathbb{Z}_+$ be the $t$-th iteration of the EM update rules $\|M(\theta, \nu)\|/\sigma$ and $N(\theta, \nu)$ at the population level, then the series approximations for EM update rules are

$$\begin{aligned} \alpha^{t+1} &= \alpha^t(1 - [\beta^t]^2) + \mathcal{O}([\alpha^t]^3), \\ \beta^{t+1} &= \beta^t(1 - \alpha^t\alpha^{t+1}) + \mathcal{O}([\alpha^t]^4). \end{aligned}$$

*Proof.* By applying the recurrence relation equation 10, namely $\alpha^{t+1} = m(\alpha^t, \nu^t), \beta^{t+1} = n(\alpha^t, \nu^t)$, and invoking the lower/upper bounds for $m(\alpha, \nu)$ and $n(\alpha, \nu)$ in Subsection A.3, we show that

$$\begin{aligned} \alpha^{t+1} &= \alpha^t(1 - [\beta^t]^2) + \mathcal{O}([\alpha^t]^3) \\ \beta^{t+1} &= \beta^t\{1 - [\alpha^t]^2(1 - [\beta^t]^2)\} + \mathcal{O}([\alpha^t]^4) \end{aligned}$$

Therefore, $\alpha^t(\alpha^{t+1} - \alpha^t(1 - [\beta^t]^2)) = \alpha^t\mathcal{O}([\alpha^t]^3) = \mathcal{O}([\alpha^t]^4)$, and

$$\beta^{t+1} = \beta^t(1 - \alpha^t\alpha^{t+1}) + \mathcal{O}([\alpha^t]^4)$$

$\square$

**Remark.** When $\alpha^t$ is sufficiently small, by the proven Lemma of series approximations of $m(\alpha, \nu)$ and $n(\alpha, \nu)$ in Subsection H.1, we have

$$\begin{aligned} \alpha^{t+1} &= m(\alpha^t, \nu^t) = \mathbb{E}[\tanh(\alpha^t X + \nu^t)X] = \alpha^t(1 - [\beta^t]^2) + (1 - [\beta^t]^2)\mathcal{O}([\alpha^t]^3), \\ \beta^{t+1} &= n(\alpha^t, \nu^t) = \mathbb{E}[\tanh(\alpha^t X + \nu^t)] = \beta^t\{1 - [\alpha^t]^2(1 - [\beta^t]^2)\} + \beta^t(1 - [\beta^t]^2)\mathcal{O}([\alpha^t]^4) \\ &= \beta^t(1 - \alpha^t\alpha^{t+1}) + \beta^t(1 - [\beta^t]^2)\mathcal{O}([\alpha^t]^4) \end{aligned}$$

Hence, we have these approximations when $\alpha^t \neq 0$ and $\beta^t \neq 0$ respectively (see Lemmas in Appendix C):

$$\begin{aligned} \frac{\alpha^{t+1}}{\alpha^t(1 - [\beta^t]^2)} &= 1 + \mathcal{O}([\alpha^t]^2), \quad \frac{\alpha^{t+1} - \alpha^t}{\alpha^t} = -[\beta^t]^2 + (1 - [\beta^t]^2)\mathcal{O}([\alpha^t]^2) \\ \frac{\beta^{t+1} - \beta^t}{\beta^t} &= -\alpha^t\alpha^{t+1} + (1 - [\beta^t]^2)\mathcal{O}([\alpha^t]^4) = -(1 - [\beta^t]^2)([\alpha^t]^2 + \mathcal{O}([\alpha^t]^4)) \end{aligned}$$

In the special case of balanced mixing weights, namely $\beta^t = \tanh\nu^t = 0$, then by using the bounds for $\alpha^{t+1} = m(\alpha^t, \nu^t) = m(\alpha^t, 0) = m_0(\alpha^t)$ in Subsubsection A.3, for sufficiently small $\alpha^t$, we have

$$\alpha^t - 3[\alpha^t]^3 \leq \alpha^{t+1} \leq \alpha^t - \frac{3[\alpha^t]^3}{1 + 8[\alpha^t]}$$

The above upper bound still holds when $\beta^t \neq 0$, since $\alpha^{t+1} = m(\alpha^t, \nu^t) = m(\alpha^t, |\nu^t|) \leq m(\alpha^t, 0) = m_0(\alpha^t) \leq \alpha^t - 3[\alpha^t]^3/(1 + 8[\alpha^t])$ by using the monotonicity of $m(\alpha, \nu)$ (Fact 4.2 in Section 4).

## C   Lemmas in Proofs for Results of Population Level Analysis

**Lemma C.1.** Let $\alpha^t := \|\theta^t\|/\sigma = \|M(\theta^{t-1}, \nu^{t-1})\|/\sigma \in (0, 0.1], \beta^t := \tanh(\nu^t) = N(\theta^{t-1}, \nu^{t-1}), \forall t \in \mathbb{Z}_+$ be the $t$-th iteration of the EM update rules $\|M(\theta, \nu)\|/\sigma, N(\theta, \nu)$ at population level, then

$$0.97 \leq \frac{\alpha^{t+1}}{\alpha^t(1 - [\beta^t]^2)} \leq 1 - \frac{5}{3}[\alpha^t]^2 + 9.53[\alpha^t]^2[\beta^t]^2, \quad -3[\alpha^t]^2 - [\beta^t]^2 \leq \frac{\alpha^{t+1} - \alpha^t}{\alpha^t} \leq -\frac{5}{3}[\alpha^t]^2 - \frac{4}{5}[\beta^t]^2.$$

*Proof.* By applying equation 10 $\alpha^{t+1} = m(\alpha^t, \nu^t), \beta^{t+1} = n(\alpha^t, \nu^t)$, invoking the lower/upper bounds for the expectation of $m, n$ in Subsection A.3, and noting that $\max_{0 \leq \alpha^t \leq 0.1, 0 \leq \beta^t \leq 1} 9[\alpha^t]^2 \left[\frac{1}{3} - [\beta^t]^2\right] + 225[\alpha^t]^4[\beta^t]^2 = 0.03$,

$\frac{1200(1 + \frac{25}{48}[\alpha^t])[\alpha^t]^3}{(1 + 16[\alpha^t])(1 - \frac{25}{3}[\alpha^t]^2)} \leq \frac{1200(1 + \frac{25}{48} \cdot 0.1)0.1^3}{(1 + 16 \cdot 0.1)(1 - \frac{25}{3}0.1^2)} \approx 0.52972 < 0.53$, we obtain the following lower/upper bounds.

$$\frac{\alpha^{t+1}}{\alpha^t(1 - [\beta^t]^2)} \geq 1 - 9[\alpha^t]^2 \left[\frac{1}{3} - [\beta^t]^2\right] - 225[\alpha^t]^4[\beta^t]^2 \geq 1 - 0.03 = 0.97$$

$$\frac{\alpha^{t+1}}{\alpha^t(1 - [\beta^t]^2)} \leq 1 - 9[\alpha^t]^2 \left[\frac{\frac{1}{3}}{1 + 8[\alpha^t]} - [\beta^t]^2\right] + [\alpha^t]^5[\beta^t]^2 \frac{1200(1 + \frac{25}{48}[\alpha^t])}{(1 + 16[\alpha^t])(1 - \frac{25}{3}[\alpha^t]^2)}$$

$$\leq 1 - 9[\alpha^t]^2 \left[\frac{\frac{1}{3}}{1 + 8 \cdot 0.1} - [\beta^t]^2\right] + 0.53[\alpha^t]^2[\beta^t]^2 = 1 - \frac{5}{3}[\alpha^t]^2 + 9.53[\alpha^t]^2[\beta^t]^2$$

By dropping out the term of $[\alpha^t]^2[\beta^t]^4$ and using $\alpha^t \leq 0.1$, the upper bound for $\alpha^{t+1}/\alpha^t$ is established.

$$\frac{\alpha^{t+1}}{\alpha^t} \leq (1 - [\beta^t]^2)\left(1 - \frac{5}{3}[\alpha^t]^2 + 9.53[\alpha^t]^2[\beta^t]^2\right) \leq 1 - \frac{5}{3}[\alpha^t]^2 - 0.888[\beta^t]^2 \leq 1 - \frac{5}{3}[\alpha^t]^2 - \frac{4}{5}[\beta^t]^2$$

By applying the lower bound for $\alpha^{t+1} = m(\alpha^t, \nu^t)$ in Subsection A.3, when $\alpha^t \in (0, 0.1]$,

$$\frac{\alpha^{t+1}}{\alpha^t} \geq (1 - [\beta^t]^2)\left(1 - 3[\alpha^t]^2 + 9[\alpha^t]^2[\beta^t]^2(1 - 25[\alpha^t]^2)\right) \geq 1 - 3[\alpha^t]^2 - [\beta^t]^2$$

$\square$

**Lemma C.2.** Let $\alpha^t := \|\theta^t\|/\sigma = \|M(\theta^{t-1}, \nu^{t-1})\|/\sigma \in (0, 0.1], \beta^t := \tanh(\nu^t) = N(\theta^{t-1}, \nu^{t-1}) \in (0, \sqrt{\frac{2}{5}}], \forall t \in \mathbb{Z}_+$ be the $t$-th iteration of EM update rules $\|M(\theta, \nu)\|/\sigma, N(\theta, \nu)$ at population level, then

$$-[\alpha^t]^2 \leq -[\alpha^t]^2(1 - [\beta^t]^2) \leq \frac{\beta^{t+1} - \beta^t}{\beta^t} \leq -0.94[\alpha^t]^2(1 - [\beta^t]^2) \leq -\frac{1}{2}[\alpha^t]^2.$$

*Proof.* By applying equation 10 $\alpha^{t+1} = m(\alpha^t, \nu^t), \beta^{t+1} = n(\alpha^t, \nu^t)$, invoking the lower/upper bounds for the expectation of $n$ in Subsection A.3, and noting that $\frac{6(1 - [\beta^t]^2)}{1 + 8[\alpha^t]} - 9[\beta^t]^2 + [\alpha^t]^2[\beta^t]^2 \frac{300}{1 + 16[\alpha^t]} \geq 0$ holds by using $[\beta^t]^2 \leq \frac{2}{5} \leq \frac{128[\alpha^t]^2 + 104\alpha^t + 6}{428[\alpha^t]^2 + 248\alpha^t + 15}$ for $\alpha^t \geq 0$, we have

$$\frac{\beta^{t+1}}{\beta^t} \geq 1 - [\alpha^t]^2(1 - [\beta^t]^2) + [\alpha^t]^4\left[\frac{6(1 - [\beta^t]^2)}{1 + 8[\alpha^t]} - 9[\beta^t]^2\right] + [\alpha^t]^6[\beta^t]^2 \frac{300}{1 + 16[\alpha^t]} \geq 1 - [\alpha^t]^2(1 - [\beta^t]^2)$$

$$\frac{\beta^{t+1}}{\beta^t} \leq 1 - [\alpha^t]^2(1 - 6[\alpha^t]^2)(1 - [\beta^t]^2) \leq 1 - 0.94[\alpha^t]^2(1 - [\beta^t]^2)$$

By applying $(1 - [\beta^t]^2) \leq 1$ and $0.94(1 - [\beta^t]^2) \geq \frac{1}{2}$ for $\beta^t \in (0, \sqrt{\frac{2}{5}}]$, we obtain the following bounds.

$$-[\alpha^t]^2 \leq -[\alpha^t]^2(1 - [\beta^t]^2) \leq \frac{\beta^{t+1} - \beta^t}{\beta^t} \leq -0.94[\alpha^t]^2(1 - [\beta^t]^2) \leq -\frac{1}{2}[\alpha^t]^2$$

$\square$

# D  Proofs for Results of Population Level Analysis

## D.1  Proof for Sublinear Convergence Rate

**Theorem D.1.** (Proposition 5.3 in Section 5: Sublinear Convergence Rate)

Let $\alpha^t := \|\theta^t\|/\sigma = \|M(\theta^{t-1}, \nu^{t-1})\|/\sigma, \beta^t := \tanh(\nu^t) = N(\theta^{t-1}, \nu^{t-1}), \forall t \in \mathbb{Z}_+$ be the $t$-th iteration of the EM update rules $\|M(\theta, \nu)\|/\sigma, N(\theta, \nu)$ at population level. Suppose that $\alpha^0 \in (0, 0.31)$, then

$$\alpha^t \leq \frac{1}{\sqrt{6t + \{8 + \frac{1}{\alpha^0}\}^2 - 8}} \quad \forall t \in \mathbb{Z}_{\geq 0};$$

when the initial guess of mixing weights is balanced $\pi^0 = (\frac{1}{2}, \frac{1}{2})$, namely $\beta^0 = \tanh \nu^0 = \|\pi^0 - \frac{1}{2}\|_1 = 0$, then

$$\alpha^t \geq \frac{1}{\sqrt{6t + 22\ln(1.2t + 1) + (\frac{1}{\alpha^0})^2}} \quad \forall t \in \mathbb{Z}_{\geq 0}.$$

*Proof.* By invoking the bound for expectation $m_0(\alpha) = m(\alpha, 0) = \mathbb{E}[\tanh(\alpha X)X]$ under the density function $X \sim K_0(|x|)/\pi$ in Subsection A.3, the EM update rule $\alpha^{t+1} = m(\alpha^t, \nu^t) = \mathbb{E}[\tanh(\alpha^t X + \nu^t)X]$ in equation 10, and the monotonicity of $m(\alpha, \nu)$ in Fact 4.2 in Section 4, we have

$$\alpha^{t+1} = m(\alpha^t, \nu^t) = m(\alpha^t, |\nu^t|) \leq m(\alpha^t, 0) \leq \alpha^t - \frac{3[\alpha^t]^3}{1 + 8[\alpha^t]} \leq \alpha^t < 0.31$$

Regarding the upper bound on the right side, we view it as the discretized version of a differential inequality $d\alpha \leq -3[\alpha]^3 dt$. By the method of "variable separation", we conclude the following upper bound. Hence, by using $0 < \alpha^{t+1} \leq \alpha^t$, we obtain $1 \leq \frac{1}{2}\left(\frac{\alpha^t}{\alpha^{t+1}}\right)^2 + \frac{1}{2}\left(\frac{\alpha^t}{\alpha^{t+1}}\right)$ and $1 \leq \left(\frac{\alpha^t}{\alpha^{t+1}}\right)$

$$
\begin{aligned}
T = \sum_{t=0}^{T-1} 1 &\leq \sum_{t=0}^{T-1} \frac{\alpha^t - \alpha^{t+1}}{\frac{3[\alpha^t]^3}{1+8[\alpha^t]}} = \sum_{t=0}^{T-1} \left\{\frac{1}{3}[\alpha^t]^{-3} + \frac{8}{3}[\alpha^t]^{-2}\right\}\{\alpha^t - \alpha^{t+1}\} \\
&\leq \sum_{t=0}^{T-1} \left\{\frac{1}{3}[\alpha^t]^{-3}\left[\frac{1}{2}\left(\frac{\alpha^t}{\alpha^{t+1}}\right)^2 + \frac{1}{2}\left(\frac{\alpha^t}{\alpha^{t+1}}\right)\right] + \frac{8}{3}[\alpha^t]^{-2}\left(\frac{\alpha^t}{\alpha^{t+1}}\right)\right\}\{\alpha^t - \alpha^{t+1}\} \\
&= \frac{1}{6}\sum_{t=0}^{T-1}\{[\alpha^{t+1}]^{-2} - [\alpha^t]^{-2}\} + \frac{8}{3}\sum_{t=0}^{T-1}\{[\alpha^{t+1}]^{-1} - [\alpha^t]^{-1}\} \\
&= \left\{\frac{1}{6}[\alpha^T]^{-2} + \frac{8}{3}[\alpha^T]^{-1}\right\} - \left\{\frac{1}{6}[\alpha^0]^{-2} + \frac{8}{3}[\alpha^0]^{-1}\right\}
\end{aligned}
$$

By substituting $t \to T$, namely $[\alpha^t]^{-2} + 16[\alpha^t]^{-1} \geq 6t + [\alpha^0]^{-2} + 16[\alpha^0]^{-1}$, and solving for $\alpha^t$, the upper bound for $\alpha^t$ is established.

Let's provide a matching lower bound to justify the sublinear convergence rate for the worst case of a balanced initial guess $\pi^0 = (\frac{1}{2}, \frac{1}{2})$, namely $\beta^0 = \tanh \nu^0 = \pi^0(1) - \pi^0(2) = 0$. By applying Fact 4.3 in Section 4, $\{|\beta^t|\}_{t=0}^{\infty} = \{\tanh|\nu^t|\}_{t=0}^{\infty}$ is nonincreasing, therefore, $\nu^t = 0, \forall t \in \mathbb{Z}_{\geq 0}$ given the initial guess is balanced.

Again, by invoking the bounds for expectations in Subsection A.3, and the EM update rule $\alpha^{t+1} = m(\alpha^t, \nu^t) = \mathbb{E}[\tanh(\alpha^t X + \nu^t)X]$ in equation 10, we obtain the following lower bounds for $\alpha^t := \|\theta^t\|/\sigma \in (0, 0.31)$.

$$\alpha^t - 3[\alpha^t]^3 \leq m_0(\alpha^t) = m(\alpha^t, 0) = m(\alpha^t, \nu^t) = \alpha^{t+1}$$

Regarding the lower bound on the left-hand side, multiplying by $12\alpha^t$ on both sides and defining $A^t := 6[\alpha^t]^2 \leq 6[\alpha^t]^2 < \frac{2}{3}$ for ease of analysis, and applying the AM-GM inequality $\sqrt{A^t A^{t+1}} \leq (A^t + A^{t+1})/2$, then

$$2A^t - [A^t]^2 \leq 2\sqrt{A^t A^{t+1}} \leq A^t + A^{t+1}$$

Hence, $A^t - A^{t+1} \leq [A^t]^2$ and dividing by $A^t A^{t+1}$ on both sides,

$$\frac{1}{A^{t+1}} - \frac{1}{A^t} \leq \frac{A^t}{A^t - (A^t - A^{t+1})} \leq \frac{A^t}{A^t - [A^t]^2} = 1 + \frac{1}{\frac{1}{A^t} - 1}$$

To obtain the lower bound for $\frac{1}{A^t} = \frac{1}{6}[\alpha^t]^{-2}$, we use the upper bound for $\alpha^t$ which has been obtained: $[\alpha^t]^{-2} + 16[\alpha^t]^{-1} \geq 6t + [\alpha^0]^{-2} + 16[\alpha^0]^{-1}$; also apply the AM-GM inequality $[\alpha^t]^{-2} + 9 \geq 6[\alpha^t]^{-1}$, and $[\alpha^0]^{-2} + 16[\alpha^0]^{-1} \geq 3^2 + 16 \times 3 = 57$.

$$\frac{11}{3}[\alpha^t]^{-2} + 24 = [\alpha^t]^{-2} + \frac{16}{6} \times ([\alpha^t]^{-2} + 9) \geq 6t + 57$$

Therefore, we have shown the lower bound for $\frac{1}{A^t}$.

$$\frac{1}{A^t} = \frac{1}{6}[\alpha^t]^{-2} \geq \frac{3}{11}t + \frac{3}{2}$$

Hence, the use of telescoping summations and bounding a summation by an integral $\sum_{\tau=0}^{t-1} \frac{1}{\tau + \frac{11}{6}} \leq \int_{\frac{5}{6}}^{t+\frac{5}{6}} \frac{d\tau}{\tau} = \ln(\frac{t+\frac{5}{6}}{\frac{5}{6}}) = \ln(1.2t + 1)$ gives the following upper bound for $\frac{1}{A^t} - \frac{1}{A^0}$.

$$\frac{1}{A^t} - \frac{1}{A^0} = \sum_{\tau=0}^{t-1}\left(\frac{1}{A^{\tau+1}} - \frac{1}{A^\tau}\right) \leq \sum_{\tau=0}^{t-1} 1 + \sum_{\tau=0}^{t-1} \frac{1}{\frac{3}{11}\tau + \frac{1}{2}} \leq t + \frac{11}{3}\ln(1.2t + 1)$$

Consequently, we obtain the following lower bound for $\alpha^t$ by substituting $A^t = 6[\alpha^t]^2$, $A^0 = 6[\alpha^0]^2$.

$$\alpha^t \geq \frac{1}{\sqrt{6t + 22\ln(1.2t + 1) + (\frac{1}{\alpha^0})^2}}$$

By combining the upper/lower bounds for $\alpha^t := \|\theta^t\|/\sigma$ in the worst case of balanced initial guess $\pi^0 = (\frac{1}{2}, \frac{1}{2})$,

$$\frac{1}{\sqrt{6t + 22\ln(1.2t + 1) + (\frac{1}{\alpha^0})^2}} \leq \alpha^t \leq \frac{1}{\sqrt{6t + (8 + \frac{1}{\alpha^0})^2 - 8}}$$

which justifies the sublinear convergence rate, as $t \gtrsim \ln(1.2t + 1)$. $\qquad\square$

**Remark.** For a finer analysis, we can establish a tighter upper bound for $\alpha^{t+1}$ by starting from the series expansion of $\tanh(\alpha^t X)$ and using the fact that $\mathbb{E}[X^{2n}] = [(2n-1)!!]^2$ for $X \sim \frac{K_0(|x|)}{\pi}$:

$$
\begin{aligned}
\alpha^{t+1} &= \mathbb{E}[\tanh(\alpha^t X + \nu^t)X] \leq \mathbb{E}[\tanh(\alpha^t X)X] \\
&\leq \alpha^t \mathbb{E}[X^2] - \frac{1}{3}[\alpha^t]^3 \mathbb{E}[X^4] + \frac{2}{15}[\alpha^t]^5 \mathbb{E}[X^6] - \frac{17}{315}[\alpha^t]^7 \mathbb{E}[X^8] + \frac{62}{2835}[\alpha^t]^9 \mathbb{E}[X^{10}] \\
&= \alpha^t - 3[\alpha^t]^3 + 30[\alpha^t]^5 - (17 \times 35)[\alpha^t]^7 + (62 \times 315)[\alpha^t]^9 \leq \alpha^t - 3[\alpha^t]^3/(1 + 10[\alpha^t]^2)
\end{aligned}
$$

where the last line holds for $\alpha^t \in [0, 0.13]$ is sufficiently small, which is tighter than the upper bound $\alpha^{t+1} \leq \alpha^t - 3[\alpha^t]^3/(1 + 8[\alpha^t])$ (since $8[\alpha^t] \geq 10[\alpha^t]^2$ for $\alpha^t \in [0, 0.13]$). By applying the same method of "variable separation", and noting that $(\alpha^t - \alpha^{t+1})/\alpha^t \leq \ln(\alpha^t/\alpha^{t+1})$ (since $x/(1+x) \leq \ln(1+x), \forall x \geq 0$), we can establish the following relation for $\alpha^t$:

$$6T \leq 6\sum_{t=0}^{T-1} \frac{\alpha^t - \alpha^{t+1}}{3[\alpha^t]^3/(1 + 10[\alpha^t]^2)} \leq \left\{\left(\frac{1}{\alpha^T}\right)^2 + 20\ln\left(\frac{1}{\alpha^T}\right)\right\} - \left\{\left(\frac{1}{\alpha^0}\right)^2 + 20\ln\left(\frac{1}{\alpha^0}\right)\right\}$$

By solving the above inequality and introducing Lambert $W$ function (see Section 4.13 Lambert W-Function of Olver et al. (2010)) and noting $\ln x - \ln\ln x \leq W(x), \forall x \geq e$, we obtain the following upper bound for $\alpha^t$ when $\alpha^0 \in (0, 0.13]$:

$$\alpha^t \leq \frac{1}{\sqrt{10W\left(\frac{1}{10[\alpha^0]^2} e^{\frac{1}{10[\alpha^0]^2}} \exp(0.6t)\right)}} \leq \frac{1}{\sqrt{6t + (\frac{1}{\alpha^0})^2 - 10\ln(6[\alpha^0]^2 t - 10[\alpha^0]^2 \ln(10[\alpha^0]^2) + 1)}}$$

## D.2 Proof for Initialization at Population Level

**Theorem D.2.** (Fact 5.2 in Section 5: Initialization at Population Level)

Let $\alpha^t := \|\theta^t\|/\sigma = \|M(\theta^{t-1}, \nu^{t-1})\|/\sigma$ and $\beta^t := \tanh(\nu^t) = N(\theta^{t-1}, \nu^{t-1})$ for all $t \in \mathbb{Z}_+$ be the $t$-th iteration of the EM update rules $\|M(\theta, \nu)\|/\sigma$ and $N(\theta, \nu)$ at the population level. If we run EM at the population level for at most $T_0 = 36$ iterations, then $\alpha^{T_0} < 0.1$.

*Proof.* By the recurrence relation of EM updates equation 10, we have $\alpha^{t+1} = m(\alpha^t, \nu^t)$. Futhermore, by applying the monotonicity of $m(\alpha, \nu)$ (Fact 4.2 in Section 4), we have

$$\alpha^{t+1} = m(\alpha^t, \nu^t) = m(\alpha^t, |\nu^t|) \leq m(\alpha^t, 0) = m_0(\alpha^t)$$

In three iterations of EM updates, by numerical evaluations of $m_0(\cdot)$ and noting that $m_0(\infty) = \frac{2}{\pi}$, applying the monotonicity of $m_0(\cdot)$ (Fact 4.2 in Section 4), we have

$$\alpha^3 \leq m_0(\alpha^2) \leq m_0(m_0(\alpha^1)) \leq m_0(m_0(m_0(\alpha^0))) \leq m_0(m_0(m_0(\infty))) = m_0\left(m_0\left(\frac{2}{\pi}\right)\right) \approx 0.305 < 0.31$$

Note that $\alpha^{t+1} \leq \alpha^t$ and $\alpha^t$ is nonincreasing (Fact 4.3 in Section 4), we justify that $\alpha^t \leq \alpha^3 < 0.31, \forall t \geq 3$. By applying the sublinear convergence rate guarantee in Proposition 5.3, selecting $T_0 = 36$ and using $\alpha^3 < 0.31$,

$$\alpha^{T_0} \leq \frac{1}{\sqrt{6 \times (T_0 - 3) + \left\{8 + \frac{1}{\alpha^3}\right\}^2 - 8}} < 0.1$$

Therefore, we have completed the proof. $\square$

**Remark.** In the worst case of balanced initial guess $\pi^0 = (\frac{1}{2}, \frac{1}{2})$, and $\alpha^0 \to \infty$, then by applying $\alpha^3 = m_0(m_0(m_0(\infty))) \approx 0.305 > 0.3$, the matching lower bound in Proposition 5.3, then for any $t \in \mathbb{Z}_{\geq 0}, 3 \leq t \leq 9$,

$$\alpha^t \geq \frac{1}{\sqrt{6(t-3) + 22\ln(1.2(t-3)+1) + (\frac{1}{\alpha^3})^2}} \geq \frac{1}{\sqrt{6 \times 6 + 22\ln(1.2 \times 6 + 1) + (\frac{1}{0.3})^2}} \approx 0.103 > 0.1$$

Therefore, in the above worst case, $\alpha^t > 0.1, \forall t \in \mathbb{Z}_{\geq 0}, t \leq 9$ is shown. By numerical evaluations in the worst case, we have $\alpha^{20} = \underbrace{m_0(m_0(m_0(\cdots m_0(\infty))))}_{20 \text{ times}} \approx 0.1$ as shown in Fig. 1b. As $9 < 20 < 36$, our theoretical conclusions based on the matching lower bound and the upper bound for $\alpha^t$ in Proposition 5.3 are verified.

## D.3 Proof for Contraction Factor of Linear Convergence

**Theorem D.3.** (Proposition 5.4 in Section 5: Contraction Factor for Linear Convergence)

Let $\alpha^t := \|\theta^t\|/\sigma = \|M(\theta^{t-1}, \nu^{t-1})\|/\sigma$ and $\beta^t := \tanh(\nu^t) = N(\theta^{t-1}, \nu^{t-1})$ for all $t \in \mathbb{Z}_+$ be the $t$-th iteration of the EM update rules $\|M(\theta, \nu)\|/\sigma$ and $N(\theta, \nu)$ at the population level. Then $\beta^\infty := \lim_{t \to \infty} \beta^t$ exists and

(i) if $\beta^0 = 0$, then $\beta^\infty = 0$,

(ii) if $\beta^0 > 0$, then $0 < \beta^\infty \leq \beta^t \leq \beta^0$,

(iii) if $\beta^0 < 0$, then $0 > \beta^\infty \geq \beta^t \geq \beta^0$.

Suppose that $\alpha^t \in (0, 0.1)$, then

$$\frac{\alpha^{t+1}}{\alpha^t} \leq 1 - \frac{4}{5}[\beta^\infty]^2$$

*Proof.* Using Fact 4.3, we have shown that $\beta^t \cdot \beta^0 \geq 0$ and $\{|\beta^t|\}_{t=0}^\infty$ is non-increasing, namely $|\beta^t| \leq |\beta^{t-1}|, \forall t \in \mathbb{Z}_+$. By the monotone convergence theorem of a sequence in Rudin (1976), $\beta^\infty := \lim_{t \to \infty} \beta^t$ exists. Hence, if $\beta^0 > 0$, then $0 \leq \beta^\infty \leq \cdots \leq \beta^t \leq \beta^{t-1} \leq \cdots \leq \beta^0$ for any $\forall t \in \mathbb{Z}_+$. Conversely, if

$\beta^0 < 0$, then $0 \geq \beta^\infty \geq \cdots \geq \beta^t \geq \beta^{t-1} \geq \cdots \geq \beta^0$ for any $\forall t \in \mathbb{Z}_+$. And if $\beta^0 = 0$, then $\beta^\infty = \cdots = \beta^t = \beta^{t-1} = \cdots = \beta^0 = 0$ for any $\forall t \in \mathbb{Z}_+$. Furthermore, we will show $\beta^\infty \neq 0$ when $\beta^0 \neq 0$ in the proof of this proposition 5.4, thereby the results in Proposition 5.4 (i) - (iii) hold for all $t \in \mathbb{Z}_+$.

For brevity, we show that $\beta^\infty > 0$ when $\beta^0 > 0$ in the following steps; similarly, we can validate that $\beta^\infty < 0$ when $\beta^0 < 0$ by following the same procedure. If all $\beta^t \geq \sqrt{\frac{2}{5}}$ for $\forall t \geq 0$, we must have $\beta^\infty \geq \sqrt{\frac{2}{5}}$. Otherwise, we have $\beta^t < \sqrt{\frac{2}{5}}$, starting from $t \geq t_0$ for some $t_0$. Without loss of generality, we may assume $t_0 = 0, 0 \leq \beta^0 < \sqrt{\frac{2}{5}}, 0 \leq \alpha^0 < 0.1$ and continue our discussion below. We begin with the proven Lemmas in Appendix C, which impplies the following inequalities:

$$\ln \alpha^{t+1} - \ln \alpha^t \leq \frac{\alpha^{t+1} - \alpha^t}{\alpha^t} \leq -\frac{5}{3}[\alpha^t]^2 - \frac{4}{5}[\beta^t]^2 \leq -\frac{4}{5}[\beta^t]^2$$

$$\ln \beta^{t+1} - \ln \beta^t \geq 1.006 \frac{\beta^{t+1} - \beta^t}{\beta^t} \geq -1.006[\alpha^t]^2$$

where the first inequality is from $\ln(1+x) \leq x, \forall x \geq 0$, $\ln(1-x) \geq -1.006x, \forall x \in [0, 0.01]$ and the second inequality is from Lemma C. We first give a rough upper bound for $[\alpha^t]^2, t \in \mathbb{Z}_+$ by applying the upper bound of Proposition 5.3 in Subsection D.1, namely $\frac{[\alpha^t]^2}{1 + 16\alpha^t} \leq \frac{1}{6t + \{8 + \frac{1}{\alpha^0}\}^2 - 8^2}$.

$$[\alpha^t]^2 \leq \frac{1 + 16\alpha^t}{6t + \{8 + \frac{1}{\alpha^0}\}^2 - 8^2} \leq \frac{1 + 16 \cdot 0.1}{6(t+1)} = \frac{2.6}{6}(t+2)^{-1}$$

Based on that rough upper bound, we have the following inequality to establish the lower bound for $\beta^t$:

$$\ln \beta^{t+1} - \ln \beta^t \geq -1.006[\alpha^t]^2 \geq -1.006 \cdot \frac{2.6}{6}(t+1)^{-1} \geq -\frac{9}{20}(t+2)^{-1}, \quad \forall t \in \mathbb{Z}_{\geq 0}$$

By taking the telescoping sum of the above inequality and taking the exponential, we have

$$\beta^t \geq \beta^0 \exp\left(-\frac{9}{20}\sum_{\tau=0}^{t-1}(\tau+2)^{-1}\right) \geq \beta^0 \exp\left(-\frac{9}{20}\int_1^{t+1}\tau^{-1}d\tau\right) = \beta^0 \exp\left(-\frac{9}{20}\ln(t+1)\right) = \beta^0(t+1)^{-\frac{9}{20}}$$

where the first inequality is from the telescoping sum of the above inequality and the second inequality is from the fact that $\sum_{\tau=0}^{t-1}(\tau+2)^{-1} \leq \int_1^{t+1}\tau^{-1}d\tau = \ln(t+1)$.

By the above rough lower bound for $\beta^t$, we can establish the other upper bound for $\alpha^t$ via using the proven inequality $\ln \alpha^{t+1} - \ln \alpha^t \leq -\frac{4}{5}[\beta^t]^2$.

$$\alpha^t \leq \alpha^0 \exp\left(-\frac{4}{5}[\beta^0]^2\sum_{\tau=0}^{t-1}(\tau+1)^{-\frac{9}{10}}\right) \leq \alpha^0 \exp\left(-\frac{4}{5}[\beta^0]^2\int_1^{t+1}\tau^{-\frac{9}{10}}d\tau\right) = \alpha^0 \exp\left(-8[\beta^0]^2\left((t+1)^{\frac{1}{10}} - 1\right)\right)$$

where the first inequality is from the telescoping sum of the above inequality and the second inequality is from the fact that $\sum_{\tau=0}^{t-1}(\tau+1)^{-\frac{9}{10}} \leq \int_1^{t+1}\tau^{-\frac{9}{10}}d\tau = 10[(t+1)^{\frac{1}{10}} - 1]$.

By applying the above upper bound for $\alpha^t$, telescoping the inequality of $\ln \beta^{t+1} - \ln \beta^t \geq -1.006[\alpha^t]^2$, taking the limit of $t \to \infty$ and noting that $\sum_{t=0}^{\infty}\exp(-16[\beta^0]^2(t+1)^{\frac{1}{10}}) \leq \int_0^{\infty}\exp(-16[\beta^0]^2t^{\frac{1}{10}})dt = \frac{10!}{(16[\beta^0]^2)^{10}}$ and $1.006 \cdot \exp(16[\beta^0]^2) \leq 1.006 \cdot \exp(16 \cdot \frac{2}{5}) \leq 10^3$, we have

$$\ln \beta^\infty - \ln \beta^0 \geq -1.006\sum_{t=0}^{\infty}[\alpha^t]^2 \geq -[\alpha^0]^2\frac{10^3 \times (10!)}{(16[\beta^0]^2)^{10}} \geq -\frac{1}{300}\frac{[\alpha^0]^2}{[\beta^0]^{20}} > -\infty.$$

Therefore, we have shown that $\beta^\infty > 0$ when $\beta^0 > 0$. By the same procedure, we have $\beta^\infty < 0$ when $\beta^0 < 0$. Consequently, when $\beta^0 \neq 0$, the contraction factor for linear convergence of $\alpha^t$ is bounded by

$$\frac{\alpha^{t+1}}{\alpha^t} \leq 1 - \frac{4}{5}[\beta^t]^2 \leq 1 - \frac{4}{5}[\beta^\infty]^2$$

where the first inequality is from the above upper bound for $(\alpha^{t+1} - \alpha^t)/\alpha^t$ and the second inequality is from the fact of $|\beta^\infty| \leq |\beta^t|$ which is shown earlier. $\qquad\square$

**Remark.** By applying the upper bound for the contraction factor, we have $\alpha^t \leq \alpha^0 \left(1 - \frac{4}{5}[\beta^\infty]^2\right)^t$. To estimate the upper bound for $\beta^\infty$, we start with the following proven inequalities from Appendix C:

$$\ln \alpha^{t+1} - \ln \alpha^t \geq 1.006 \frac{\alpha^{t+1} - \alpha^t}{\alpha^t} \geq -1.006 \left(3[\alpha^t]^2 + [\beta^t]^2\right) \geq -\frac{1}{20}(27(t+2)^{-1} + 21[\beta^0]^2)$$

$$\ln \beta^{t+1} - \ln \beta^t \leq \frac{\beta^{t+1} - \beta^t}{\beta^t} \leq -\frac{1}{2}[\alpha^t]^2 \leq -\frac{1}{2}[\alpha^0]^2(t+1)^{-\frac{27}{10}} \exp\left(-\frac{21}{10}[\beta^0]^2 t\right)$$

where bounds for $(\alpha^{t+1} - \alpha^t)/\alpha^t, (\beta^{t+1} - \beta^t)/\beta^t$ are from lemmas in Appendix C, and the first inequality leads to lower bound for $\alpha^t$, aka $\alpha^t \geq \alpha^0 \exp\left(-\sum_{\tau=0}^{t-1} \frac{1}{20}(27(\tau+2)^{-1} + 21[\beta^0]^2)\right) \geq \alpha^0(t+1)^{-\frac{27}{20}} \exp\left(-\frac{21}{20}[\beta^0]^2 t\right)$.

By applying the above lower bound for $\alpha^t$, then for given $\beta^0 < \sqrt{\frac{2}{5}}$, we have

$$\ln \beta^\infty - \ln \beta^0 \leq -\frac{1}{2}[\alpha^0]^2 \int_0^\infty (t+1)^{-\frac{27}{10}} \exp\left(-\frac{21}{10}[\beta^0]^2 t\right) \mathrm{d}t \leq -\frac{1}{2}[\alpha^0]^2 \int_0^\infty (t+1)^{-\frac{27}{10}} \mathrm{e}^{-\frac{21}{25}t} \mathrm{d}t \leq -\frac{[\alpha^0]^2}{4}$$

where the first inequality is from the telescoping sum of the above inequality.

### D.4 Proof for Convergence Rate at Population Level

**Theorem D.4.** (Theorem 5.1 in Section 5: Convergence Rate at Population Level)

Suppose a 2MLR model is fitted to the overspecified model with no separation $\theta^* = \vec{0}$, then for any $\epsilon \in (0, \frac{2}{\pi}]$:

(unbalanced) if $\pi^0 \neq \left(\frac{1}{2}, \frac{1}{2}\right)$, population EM takes at most $T = \mathcal{O}\left(\log \frac{1}{\epsilon}\right)$ iterations to achieve $\|\theta^T\|/\sigma \leq \epsilon$,

(balanced) if $\pi^0 = \left(\frac{1}{2}, \frac{1}{2}\right)$, population EM takes at most $T = \mathcal{O}(\epsilon^{-2})$ iterations to achieve $\|\theta^T\|/\sigma \leq \epsilon$.

*Proof.* If $\theta^0 = 0$, then $\theta^t = 0 \leq \epsilon, \forall t \in \mathbb{Z}_{\geq 0}$; otherwise, by invoking the fact for initialization at population level in Subsection D.2, then $\alpha^{T_0} = \|\theta^{T_0}\|/\sigma < 0.1$ after running population EM for at most $T_0 = \mathcal{O}(1)$ iterations.

Without loss of generality, we may assume $\alpha^0 = \|\theta^0\|/\sigma \in (0, 0.1)$ in the following discussions.

Let's prove the cases of (i) $\pi^0 = \left(\frac{1}{2}, \frac{1}{2}\right)$ and (ii) $\pi^0 \neq \left(\frac{1}{2}, \frac{1}{2}\right)$, separately.

**Proof for Unbalanced Case** $\pi^0 \neq \left(\frac{1}{2}, \frac{1}{2}\right)$

Note that $|\beta^0| = \left\|\pi^0 - \frac{1}{2}\right\|_1 \neq 0$, then by using the proposition for contraction factor in Subsection D.3, the limit $\beta^\infty := \lim_{t\to\infty} \beta^t \neq 0$, and for $\alpha^0 \in (0, 0.1), 0 < \alpha^{t+1} < \alpha^t$

$$\frac{\alpha^{t+1}}{\alpha^t} \leq 1 - \frac{4}{5}[\beta^\infty]^2$$

Therefore, by taking $T = \lceil \frac{\log \frac{1}{\epsilon} - \log 10}{-\log(1 - \frac{4}{5}[\beta^\infty]^2)} \rceil_+ = \Theta\left(\log \frac{1}{\epsilon}\right)$, then for $\alpha^T := \|\theta^T\|/\sigma$, we have

$$\alpha^T \leq (1 - \frac{4}{5}[\beta^\infty]^2)^T \alpha^0 \leq 0.1$$

**Proof for Balanced Case** $\pi^0 = \left(\frac{1}{2}, \frac{1}{2}\right)$

By applying the proposition for sublinear convergence in Subsection D.1, for $\alpha^0 \in (0, 0.1)$ and $t \in \mathbb{Z}_{\geq 0}$

$$\alpha^t \leq \frac{1}{\sqrt{6t + \{8 + 1/\alpha^0\}^2} - 8}$$

Hence, by taking $T = \lceil (\epsilon^{-2} + 16\epsilon^{-1} - [\alpha^0]^{-2} - 16[\alpha^0]^{-1})/6 \rceil_+ = \Theta(\epsilon^{-2})$, we have $\alpha^T := \|\theta^T\|/\sigma \leq \epsilon$. $\qquad\square$

# E    Lemmas in Proofs for Results of Finite-Sample Level Analysis

## E.1    Modified Log-Sobolev Inequality

The following definition and logarithmic Sobolev inequality are on Page 91, Chapter 5 Entropy and Concentration, (Ledoux, 2001) by Michel Ledoux.

**Definition E.1.** Entropy of non-negative measurable function $f$ given a probability measure $\mu$ is defined as

$$\mathrm{Ent}_\mu[f] \;=\; \begin{cases} \int f \log f \mathrm{d}\mu - \left(\int f \mathrm{d}\mu\right) \log\left(\int f \mathrm{d}\mu\right) = \int f \log \frac{f}{\int f \mathrm{d}\mu} \mathrm{d}\mu & \text{if } \int f \log(1+f)\mathrm{d}\mu < \infty \\ \infty & \text{otherwise} \end{cases}$$

**Remark.** Since $f \log f \le f \log(1+f) \le 1 + f \log f$ and $0 \le f \log(1+f), \int \mathrm{d}\mu = 1$, then

$$-1 \le -1 + \int f \log(1+f)\mathrm{d}\mu \le \int f \log f \mathrm{d}\mu \le \int f \log(1+f)\mathrm{d}\mu$$

If $\int f \log(1+f)\mathrm{d}\mu < \infty$, then $-1 \le \int f \log f \mathrm{d}\mu < \infty$; note $f\left(2 - \frac{4}{f}\right) \le f\left(2 - \frac{2}{1+\frac{f}{2}}\right) = f\left(\frac{f}{1+\frac{f}{2}}\right) \le f \log(1+f)$

$$0 \le \int f \mathrm{d}\mu \le \frac{1}{2}\left\{4 + \int f \log(1+f)\mathrm{d}\mu\right\} < \infty$$

Hence, $\mathrm{Ent}_\mu[f]$ is bounded if $\int f \log(1+f)\mathrm{d}\mu < \infty$.

**Definition E.2** (Equation (5.1) in Ledoux (2001): Logarithmic Sobolev Inequality)**.** A probability measure $\mu$ on $\mathbb{R}^d$ is said to satisfy a logarithmic Sobolev inequality if for some constant $c > 0$ and all smooth enough functions $f$ on $\mathbb{R}^d$,

$$\mathrm{Ent}_\mu[f^2] \;\le\; c\mathbb{E}_\mu[|\nabla f|^2].$$

**Remark.** Let $f^2 = \exp(\psi)$ and $\psi : \mathbb{R} \to \mathbb{R}$ be a Lipschitz function such that $|\psi(x) - \psi(x')| \le \lambda|x - x'|$, then the "modified logarithmic Sobolev inequality" becomes

$$\mathrm{Ent}_\mu[\exp(\psi)] \le c\mathbb{E}_\mu[[\psi']^2 \exp(\psi)] \le c\lambda^2\mathbb{E}_\mu[\exp(\psi)].$$

**Lemma E.3.** (Bounds for Bessel Function, see Chapter 10 Bessel Function of Olver et al. (2010))

Let $K_0$ be the modified Bessel function with parameter 0, then for $x > 0$

$$\sqrt{\frac{1}{2\pi}}\frac{\exp(-x)}{\sqrt{x+1}} \;\le\; \frac{K_0(x)}{\pi} \;\le\; \sqrt{\frac{1}{2\pi}}\frac{\exp(-x)}{\sqrt{x}}$$

*Proof.* By invoking the monotonicity property of $K_\nu$, namely $|K_\nu(x)| \le |K_{\nu'}(x)|$ for $0 \le \nu \le \nu', \forall x > 0$, in Section 10.37 Inequalities; Monotonicity, and $K_{\frac{1}{2}}(x) = \sqrt{\frac{\pi}{2}}\frac{\exp(-x)}{\sqrt{x}}$ in Section 10.39 Relation to Other Functions, Chapter 10 Bessel Function of Olver et al. (2010).

$$\frac{K_0(x)}{\pi} \le \frac{K_{\frac{1}{2}}(x)}{\pi} = \sqrt{\frac{1}{2\pi}}\frac{\exp(-x)}{\sqrt{x}}$$

For the lower bound of $\frac{K_0(x)}{\pi}$, noting that $K_0(x)$ and $\sqrt{\frac{\pi}{2}}\frac{\exp(-x)}{\sqrt{x+1}}$ are monotonically decreasing, it can be confirmed through numerical validation that

$$\sqrt{\frac{\pi}{2}}\frac{\exp(-x)}{\sqrt{x+1}} < \sqrt{\frac{\pi}{2}} < 1.26 < 1.54 < K_0\left(\frac{1}{4}\right) < K_0(x)$$

for $x \in (0, \frac{1}{4})$. Hence, we focus on the case of $x \ge \frac{1}{4}$ in the following discussion.

By applying $1 < (1 - \frac{1}{8} \times \frac{1}{x})^2(1 + \frac{1}{x})$ for $\frac{1}{x} \in (0, 4]$, hence $\frac{1}{\sqrt{x+1}} < (1 - \frac{1}{4} \times \frac{1}{2x})x^{-\frac{1}{2}}$, invoking $\sqrt{\pi}x^{-\frac{1}{2}} = 2x \int_{\mathbb{R}_+} \exp(-xy)\sqrt{y}\mathrm{d}y = \int_{\mathbb{R}_+} \exp(-xy)/\sqrt{y}\mathrm{d}y$, and using $1 - y/4 < 1/\sqrt{1 + y/2}$ for $y > 0$

$$\sqrt{\frac{\pi}{2}} \frac{\exp(-x)}{\sqrt{x+1}} < \frac{\exp(-x)}{\sqrt{2}}\left(1 - \frac{1}{4} \times \frac{1}{2x}\right)\sqrt{\pi}x^{-\frac{1}{2}} = \frac{\exp(-x)}{\sqrt{2}} \int_{\mathbb{R}_+}\left(1 - \frac{y}{4}\right)\frac{\exp(-xy)}{\sqrt{y}}\mathrm{d}t < \int_{\mathbb{R}_+} \frac{\exp(-x[1+y])}{\sqrt{y(2+y)}}\mathrm{d}y$$

Namely, noting that $\int_{\mathbb{R}_+} \frac{\exp(-x[1+y])}{\sqrt{y(2+y)}}\mathrm{d}y = \int_{\mathbb{R}_+} \exp(-x\cosh t)\mathrm{d}t$ by substituting $y = \cosh t - 1$, and invoking $\int_{\mathbb{R}_+} \exp(-x\cosh t)\mathrm{d}t = K_0(x)$ (Equation 10.32.8) in Section 10.32 Integral Representations, Chapter 10 Bessel Function of Olver et al. (2010).

$$\sqrt{\frac{\pi}{2}} \frac{\exp(-x)}{\sqrt{x+1}} \leq K_0(x)$$

$\square$

**Lemma E.4.** (Modified Logarithmic Sobolev Inequality) Let $\psi : \mathbb{R} \to \mathbb{R}$ be a function such that $|\psi(x)| \leq \lambda|x|$ for $\forall x \in \mathbb{R}$, and $0 \leq \lambda \leq \frac{1}{2} < 1$; the probability measusre $\mu$ is induced by a density function $\frac{\mathrm{d}\mu}{\mathrm{d}x} = \frac{K_0(|x|)}{\pi}$, then for any $c \geq 7$,

$$\mathrm{Ent}_\mu[\exp(\psi)] \leq c\lambda^2 \mathbb{E}_\mu[\exp(\psi)] = c\lambda^2 \int \exp(\psi(x))\mathrm{d}\mu.$$

*Proof.* **Step 1**. Upper-bound for $\mathrm{Ent}_\mu[\exp(\psi)]$

Note that $t\log t - t + 1 \geq 0$, and let $t \leftarrow \mathbb{E}_\mu[\exp(\psi)] = \int \exp(\psi)\mathrm{d}\mu$.

Then, we apply $\psi - 1 + \exp(-\psi) \leq \frac{1}{2}\psi^2$, $\mathrm{d}\mu = \frac{K_0(|x|)}{\pi}\mathrm{d}x$ and $\frac{K_0(x)}{\pi} \leq \sqrt{\frac{1}{2\pi}}\frac{\exp(-x)}{\sqrt{x}}$ proved in previous Lemma.

Finally, we adopt the assumpton $0 \leq \lambda \leq \frac{1}{2} < 1$,

$$\begin{aligned}
\mathrm{Ent}_\mu[\exp(\psi)] &\leq \mathbb{E}_\mu[\psi\exp(\psi) - \exp(\psi) + 1] = \mathbb{E}_\mu[\{\psi - 1 + \exp(-\psi)\}\exp(\psi)] \\
&\leq \frac{1}{2}\mathbb{E}_\mu[\psi^2\exp(\psi)] \leq \frac{\lambda^2}{2}\mathbb{E}_\mu[x^2\exp(\lambda|x|)] \\
&\leq \sqrt{\frac{1}{2\pi}}\lambda^2 \int_{\mathbb{R}_+} x^{\frac{3}{2}}\exp(-[1-\lambda]x)\mathrm{d}x = \frac{3}{4\sqrt{2}}\frac{\lambda^2}{(1-\lambda)^{\frac{5}{2}}} \leq 3\lambda^2
\end{aligned}$$

**Step 2**. Lower-bound for $\mathbb{E}_\mu[\exp(\psi)]$

We apply $\frac{K_0(x)}{\pi} \geq \sqrt{\frac{1}{2\pi}}\frac{\exp(-x)}{\sqrt{x+1}}$ in previous Lemma, and use the assumption $0 \leq \lambda \leq \frac{1}{2} < 1$.

$$\begin{aligned}
\mathbb{E}_\mu[\exp(\psi)] &\geq \mathbb{E}_\mu[\exp(-\lambda|x|)] \geq \sqrt{\frac{2}{\pi}} \int_{\mathbb{R}_+} \frac{\exp(-[1+\lambda]x)}{\sqrt{x+1}}\mathrm{d}x \\
&\geq \sqrt{\frac{2}{\pi}} \int_{\mathbb{R}_+} \frac{\exp\left(-\frac{3}{2}x\right)}{\sqrt{x+1}}\mathrm{d}x = \frac{2e^{\frac{3}{2}}}{\sqrt{3}}\mathrm{erfc}\left(\sqrt{\frac{3}{2}}\right)
\end{aligned}$$

**Step 3**. Estimate for $c$

$$\frac{\mathrm{Ent}_\mu[\exp(\psi)]}{\lambda^2 \mathbb{E}_\mu[\exp(\psi)]} \leq \frac{3\lambda^2}{\frac{2e^{\frac{3}{2}}}{\sqrt{3}}\mathrm{erfc}\left(\sqrt{\frac{3}{2}}\right)\lambda^2} = \frac{3\sqrt{3}e^{-\frac{3}{2}}}{2\mathrm{erfc}\left(\sqrt{\frac{3}{2}}\right)} \approx 6.9622594 < 7$$

By choosing $c \geq 7$, the proof is complete.

$$\mathrm{Ent}_\mu[\exp(\psi)] \leq c\lambda^2 \mathbb{E}_\mu[\exp(\psi)] = c\lambda^2 \int \exp(\psi(x))\mathrm{d}\mu$$

$\square$

We could derive the concentration inequality based on the previous Lemma (modified log-Sobolev inequality), see also section 5.3 modified logarithmic Sobolev inequalities in Ledoux (2001).

**Theorem E.5.** (Concentration Inequality with Exponential Tail)

Let $F : \mathbb{R} \to \mathbb{R}$ be a function such that $|F(x)| \le |x|$ for $\forall x \in \mathbb{R}$;

the probability measusre $\mu$ is induced by a density function $\{x_i\}_{i=1}^n \overset{\text{iid}}{\sim} \frac{d\mu}{dx} = \frac{K_0(|x|)}{\pi}$, let $F_i := F(x_i), F = F(x)$, then for any $t \ge 0$ and $c \ge 7$,

$$\mathbb{P}\left(\left|\sum_{i=1}^n (F_i - \mathbb{E}_\mu[F_i])\right| \ge nt\right) \le 2\exp\left\{-\frac{n}{4}\min\left(t, \frac{t^2}{c}\right)\right\}.$$

*Proof.* **Step 1.** Estimate the Chernoff bound for $F$ for $t \ge 0$, and by the independence of $F_i$

$$
\begin{aligned}
\mathbb{P}\left(\sum_{i=1}^n (F_i - \mathbb{E}_\mu[F_i]) \ge nt\right) &= \mathbb{P}\left(\exp(\lambda[\sum_{i=1}^n F_i - \mathbb{E}_\mu[F_i]]) \ge \exp(n\lambda t)\right) \\
&\le \inf_{\lambda>0} \frac{\mathbb{E}_\mu\left[\exp(\lambda[\sum_{i=1}^n F_i - \mathbb{E}_\mu[F_i]])\right]}{\exp(n\lambda t)} \\
&= \exp\left\{n\inf_{\lambda>0}\lambda\left[\frac{\log\mathbb{E}_\mu[\exp(\lambda F)]}{\lambda} - \mathbb{E}_\mu[F] - t\right]\right\}
\end{aligned}
$$

**Step 2.** Upper-bound for $\zeta(\lambda) := \frac{\log\mathbb{E}_\mu[\exp(\lambda F)]}{\lambda}$ for $0 < \lambda \le \frac{1}{2} < 1$

We study the evolution of $\zeta(\lambda)$ by taking the derivative of $\zeta(\lambda)$.

$$\frac{d}{d\lambda}\zeta(\lambda) = \frac{\frac{\lambda\mathbb{E}_\mu[F\exp(\lambda F)]}{\mathbb{E}_\mu[\exp(\lambda F)]} - \log\mathbb{E}_\mu[\exp(\lambda F)]}{\lambda^2} = \frac{\text{Ent}_\mu[\exp(\lambda F)]}{\lambda^2\mathbb{E}_\mu[\exp(\lambda F)]}$$

By using the previous Lemma, $\frac{\text{Ent}_\mu[\exp(\lambda F)]}{\lambda^2\mathbb{E}_\mu[\exp(\lambda F)]} \le c$ for $0 \le \lambda \le \frac{1}{2} < 1$ and $c \ge 7$.

$$\frac{d}{d\lambda}\zeta(\lambda) \le c \qquad \zeta(0) = \lim_{\lambda\to 0}\frac{\log\mathbb{E}_\mu[\exp(\lambda F)]}{\lambda} = \lim_{\lambda\to 0}\frac{\frac{\mathbb{E}_\mu[F\exp(\lambda F)]}{\mathbb{E}_\mu[\exp(\lambda F)]}}{1} = \mathbb{E}_\mu[F]$$

Hence, we show the following upper-bound for $\zeta(\lambda), \forall \lambda \in (0, \frac{1}{2}]$.

$$\zeta(\lambda) = \zeta(0) + \int_0^\lambda \zeta(\lambda')d\lambda' \le \mathbb{E}_\mu[F] + c\lambda$$

**Step 3.** Obtain the concentration inequality for $F$ when $t \ge 0$

Note that $\frac{\log\mathbb{E}_\mu[\exp(\lambda F)]}{\lambda} - \mathbb{E}_\mu[F] - t = \zeta(\lambda) - \mathbb{E}_\mu[F] - t \le c\lambda - t$, and $-\frac{t}{2} + \frac{c}{4} \le -\frac{t}{4}, \forall t \ge c$

$$
\begin{aligned}
\mathbb{P}\left(\sum_{i=1}^n (F_i - \mathbb{E}_\mu[F_i]) \ge nt\right) &\le \exp\left\{\inf_{\lambda\in(0,\frac{1}{2}]}n\lambda\left[\frac{\log\mathbb{E}_\mu[\exp(\lambda F)]}{\lambda} - \mathbb{E}_\mu[F] + t\right]\right\} \\
&\le \exp\left\{n\inf_{\lambda\in(0,\frac{1}{2}]}\lambda(c\lambda - t)\right\} = \exp\left(n\lambda(c\lambda - t)_{\lambda=\min\left(\frac{1}{2}, \frac{t}{2c}\right)}\right) \\
&\le \exp\left\{n\left\{-\frac{t}{4}\right\}\mathbb{1}_{t\ge c} + n\left\{-\frac{t^2}{4c}\right\}\mathbb{1}_{0\le t<c}\right\} = \exp\left\{-\frac{n}{4}\min\left(t, \frac{t^2}{c}\right)\right\}
\end{aligned}
$$

Let $F \leftarrow -F$ in the above expression, we obtain the upper-bound for the probablity measure

$$\mathbb{P}\left(\sum_{i=1}^n (F_i - \mathbb{E}_\mu[F_i]) \le -nt\right) \le \exp\left\{-\frac{n}{4}\min\left(t, \frac{t^2}{c}\right)\right\}$$

Combine the above two cases, we conclude the following for $\forall t \geq 0$ and $c \geq 7$,

$$\mathbb{P}\left(\left|\sum_{i=1}^{n}(F_i - \mathbb{E}_{\mu}[F_i])\right| \geq nt\right) \leq 2\exp\left\{-\frac{n}{4}\min\left(t, \frac{t^2}{c}\right)\right\}.$$

$\square$

### E.2 Azuma-Hoeffding Inequality and Elementary Inequality of $\tanh$

**Lemma E.6.** (Azuma-Hoeffding Inequality, Corollary 2.20 on page 36 in Wainwright (2019))

Let $(\{(D_k, \mathcal{F}_k)\}_{k=1}^{\infty})$ be a martingale difference sequence for which there are constants $\{(a_k, b_k)\}_{k=1}^{n}$ such that $D_k \in [a_k, b_k]$ almost surely for all $k = 1, \ldots, n$. Then, for all $t \geq 0$,

$$\mathbb{P}\left[\left|\sum_{k=1}^{n} D_k\right| \geq t\right] \leq 2\exp\left(-\frac{2t^2}{\sum_{k=1}^{n}(b_k - a_k)^2}\right).$$

**Lemma E.7.** (Upper Bound for tanh) For $\forall t, \nu \in \mathbb{R}$, we have

$$|\tanh(t + \nu) - \tanh\nu| \leq 1 + \tanh|\nu| \leq 2$$

and

$$|\tanh(t + \nu) - \tanh\nu| \leq \frac{1 + \tanh|\nu|}{1 + |\nu|} \times |t| \leq \frac{2|t|}{1 + |\nu|}.$$

*Proof.* The first inequality is proved by applying $|a + b| \leq |a| + |b|$ and $\tanh|\nu| \leq 1$.

Let's prove the second inequality by discussing these two cases $t \leq 0$ and $t > 0$. Without loss of generality, we assume $\nu \geq 0$ in following discussions, since we can always let $-\nu \to \nu, -t \to t$ for $\nu < 0$.

(i) If $t \leq 0$, then by noting that $g(t) := \min\{\tanh(t+\nu), t+\nu\}$ is a concave function of $t$, the straight line which connects these two points $g(t)|_{t=-(1+\nu)}$ and $g(t)|_{t=0}$ must be not greater than $g(t)$ for $\forall t \in [-(1+\nu), 0]$.

$$g(t) \geq g(t)|_{t=0} + \frac{g(t)|_{t=0} - g(t)|_{t=-(1+\nu)}}{1 + \nu} \times t \quad \forall t \in [-(1+\nu), 0]$$

Namely,

$$\tanh(t + \nu) \geq g(t) := \min\{\tanh(t + \nu), t + \nu\} \geq \tanh\nu + \frac{1 + \tanh\nu}{1 + \nu}t \quad \forall t \in [-(1+\nu), 0]$$

Note that

$$\tanh(t + \nu) \geq -1 > \tanh\nu + \frac{1 + \tanh\nu}{1 + \nu}t \quad \forall t < -(1+\nu)$$

By applying $\tanh(t + \nu) - \tanh\nu \leq 0$ and $1 + \tanh\nu \leq 2$, we validate the second inequality for the case of $t \leq 0$.

(ii) If $t > 0$, by applying the identity $\tanh(t) = (\tanh(t + \nu) - \tanh\nu)/(1 - \tanh(t + \nu)\tanh\nu)$ and $\tanh t < t, 0 \leq \tanh\nu < \tanh(t + \nu)$

$$\tanh(t + \nu) - \tanh\nu \leq (1 - \tanh^2\nu) \times t \quad \forall t > 0$$

Then, by invoking $\frac{\nu}{1+\nu} \leq \tanh\nu \leq 1$, we have

$$\tanh(t + \nu) - \tanh\nu \leq \frac{1 + \tanh\nu}{1 + \nu}t \leq \frac{2t}{1 + \nu} \quad \forall t > 0$$

$\square$

### E.3 Bounds for Statstics

**Lemma E.8.** (Operator Norm Bounds for Gaussian Ensemble, see Example 6.2 of Wainwright (2019))

Let $\{\vec{x}_i\}_{i=1}^n \overset{\text{iid}}{\sim} \mathcal{N}(0, I_d)$ and $d \leq n$, then for the minimal eigenvalue $\gamma_{\min}$ and maximal eigenvalue $\gamma_{\max}$

$$\left\| \left[ \frac{1}{n} \sum_{i=1}^n \vec{x}_i \vec{x}_i^\top \right]^{-1} \right\|_2^{-\frac{1}{2}} = \left[ \gamma_{\max} \left( \left[ \frac{1}{n} \sum_{i=1}^n \vec{x}_i \vec{x}_i^\top \right]^{-1} \right) \right]^{-\frac{1}{2}} = \sqrt{\gamma_{\min} \left( \frac{1}{n} \sum_{i=1}^n \vec{x}_i \vec{x}_i^\top \right)} \geq 1 - t - \sqrt{\frac{d}{n}} \quad \forall t \geq 0$$

and the $\ell_2$-operator norm $\| \cdot \|_2$

$$\left\| \frac{1}{n} \sum_{i=1}^n \vec{x}_i \vec{x}_i^\top - I_d \right\|_2 \leq 2 \left( t + \sqrt{\frac{d}{n}} \right) + \left( t + \sqrt{\frac{d}{n}} \right)^2$$

with probability greater than $1 - 2 \exp(-nt^2/2)$.

**Lemma E.9.** (Upper Bound for Chi-square r.v., see Lemma 1, page 1325 in Laurent & Massart (2000))

Let $Z \sim \chi^2(n)$, then for $t \geq 0$

$$\mathbb{P}\left( Z \geq n + 2\sqrt{nt} + 2t \right) \leq \exp(-t)$$

**Lemma E.10.** (Upper-Bound for Weighted Sum of Gaussian Vectors)

Let $\{x_i\}_{i=1}^n \overset{\text{iid}}{\sim} \mathcal{N}(0,1)$ and $\{\vec{x}_i\}_{i=1}^n \overset{\text{iid}}{\sim} \mathcal{N}(0, I_d)$, then

$$\left\| \sum_{i=1}^n x_i \vec{x}_i \right\|_2 \leq 2\sqrt{nd} + 2 \log \frac{2}{\delta} + 2(\sqrt{n} + \sqrt{d}) \sqrt{\log \frac{2}{\delta}}$$

with probability at least $1 - \delta$, namely, if $n \geq d$

$$\left\| \frac{1}{n} \sum_{i=1}^n x_i \vec{x}_i \right\|_2 = \mathcal{O}\left( \sqrt{\frac{d}{n}} \vee \frac{\log \frac{1}{\delta}}{n} \vee \sqrt{\frac{\log \frac{1}{\delta}}{n}} \right)$$

*Proof.* By using the rotational invariance of Gaussians, we can rewrite the $\ell_2$ norm as the geometrical mean of two Chi-square random variables $Z_1 \sim \chi^2(n), Z_2 \sim \chi^2(d)$.

$$\left\| \sum_{i=1}^n x_i \vec{x}_i \right\|_2 = \sqrt{Z_1 Z_2}$$

By using the concentration inequality for Chi-square distribution (see Lemma 1, page 1325 in Laurent & Massart (2000)), then with at least probability at least $1 - \delta$,

$$Z_1 \leq \left( \sqrt{n} + \sqrt{\log \frac{2}{\delta}} \right)^2 + \log \frac{2}{\delta}, \quad Z_2 \leq \left( \sqrt{d} + \sqrt{\log \frac{2}{\delta}} \right)^2 + \log \frac{2}{\delta}$$

Therefore

$$\sqrt{Z_1 Z_2} \leq 2 \left( \sqrt{n} + \sqrt{\log \frac{2}{\delta}} \right) \left( \sqrt{d} + \sqrt{\log \frac{2}{\delta}} \right) = 2\sqrt{nd} + 2 \log \frac{2}{\delta} + 2(\sqrt{n} + \sqrt{d}) \sqrt{\log \frac{2}{\delta}}$$

If $n \geq d$, we have

$$\left\| \frac{1}{n} \sum_{i=1}^n x_i \vec{x}_i \right\|_2 = \mathcal{O}\left( \sqrt{\frac{d}{n}} \vee \frac{\log \frac{1}{\delta}}{n} \vee \sqrt{\frac{\log \frac{1}{\delta}}{n}} \right)$$

$\square$

**Lemma E.11.** (Upper Bound for Norm of Sum of r.v., Corollary 38 on page 55 of (Ahle et al., 2020)) Let $p \geq 2, C > 0$ and $\alpha \geq 1$. Let $(X_i)_{i \in [n]}$ be iid. mean 0 random variables such that $\|X_i\|_p \sim (Cp)^\alpha$, then $\left\|\sum_{i=1}^n X_i\right\|_p \sim C^\alpha \max\left\{2^\alpha \sqrt{pn}, (n/p)^{1/p} p^\alpha\right\}$.

**Lemma E.12.** (Upper-Bound for Cubic of Exp r.v.)Suppose $\{Z_i\}_{i=1}^n \overset{\text{iid}}{\sim} \exp(-z)\mathbb{I}_{z \geq 0}$, then we have

$$\frac{1}{n}\sum_{i=1}^n Z_i^3 = \mathcal{O}\left(\sqrt{\frac{\log\frac{1}{\delta}}{n}} \vee \frac{\log^3\frac{1}{\delta'}}{n} \vee 1\right)$$

with probability at least $1 - (\delta + \delta')$.

*Proof.* Noting Khintchine's inequality $\|Z_i^3\|_p = \Gamma(3p+1)^{1/p} \lesssim p^3$ and $\mathbb{E}[Z_i^3] = 6$, $\|Z_i^3 - \mathbb{E}[Z_i^3]\|_p \lesssim 6 + p^3 \lesssim p^3$ for $p \geq 2$, applying Corollary 38 on page 55 of Ahle et al. (2020) in the second step.

$$\begin{aligned}
\mathbb{P}\left\{\frac{1}{n}\sum_{i=1}^n Z_i^3 \geq \mathbb{E}[Z_i^3] + \varepsilon\right\} &\leq \inf_{p>0}(n\varepsilon)^{-p}\mathbb{E}\left[\sum_{i=1}^n(Z_i^3 - \mathbb{E}[Z_i^3])\right]^p = \inf_{p>0}(n\varepsilon)^{-p}\left|\sum_{i=1}^n Z_i^3 - \mathbb{E}[Z_i^3]\right|_p^p \\
&\lesssim \inf_{p\geq 2}(n\varepsilon)^{-p}\max\left\{\sqrt{pn}, (n/p)^{1/p}p^3\right\}^p \\
&\leq \inf_{p\geq 2}\left(\frac{\sqrt{n}\varepsilon}{\sqrt{p}}\right)^{-p} + \inf_{p\geq 2}\frac{n}{p}\left(\frac{n\varepsilon}{p^3}\right)^{-p} \\
&\leq \exp\left(-\frac{n\varepsilon^2}{2e}\right) + \frac{1}{\varepsilon}\left((n\varepsilon)^{\frac{1}{3}}\right)^2 \exp\left(1 - 3\frac{(n\varepsilon)^{\frac{1}{3}}}{e}\right)
\end{aligned}$$

The last step is achieved by taking $p = \frac{n\varepsilon^2}{e}, p = \frac{(n\varepsilon)^{\frac{1}{3}}}{e}$ for these two above terms respectively, and $\varepsilon \geq \max\left\{\sqrt{\frac{2e}{n}}, \frac{(2e)^3}{n}\right\}$.

Note that inequality $\varepsilon \geq \sqrt{\frac{2e}{n}}$, then $\frac{1}{\varepsilon} \leq (2e)^{-\frac{1}{2}} \times \sqrt{n}$ and $n\varepsilon \geq (2en)^{\frac{1}{2}}$, therefore $\frac{5}{e}(n\varepsilon)^{\frac{1}{3}} \geq \frac{5}{e}(2en)^{\frac{1}{6}} \geq \log n$, namely $\sqrt{n}\exp(-\frac{5}{2e}(n\varepsilon)^{\frac{1}{3}}) \leq 1$ for $\forall n \geq 1$

$$\mathbb{P}\left\{\frac{1}{n}\sum_{i=1}^n Z_i^3 \geq \mathbb{E}[Z_i^3] + \varepsilon\right\} \lesssim \exp\left(-\frac{n\varepsilon^2}{2e}\right) + \left(\frac{e}{2}\right)^{\frac{1}{2}}\left((n\varepsilon)^{\frac{1}{3}}\right)^2\exp\left(-\frac{(n\varepsilon)^{\frac{1}{3}}}{2e}\right) \lesssim \exp\left(-\frac{n\varepsilon^2}{2e}\right) + \exp\left(-\frac{(n\varepsilon)^{\frac{1}{3}}}{4e}\right)$$

By letting $\varepsilon = \Theta\left(\sqrt{\frac{\log\frac{1}{\delta}}{n}} \vee \frac{\log^3\frac{1}{\delta'}}{n}\right)$, then with probability at least $1 - (\delta + \delta')$

$$\frac{1}{n}\sum_{i=1}^n Z_i^3 \leq \mathbb{E}[Z_i^3] + \varepsilon = \Theta\left(\sqrt{\frac{\log\frac{1}{\delta}}{n}} \vee \frac{\log^3\frac{1}{\delta'}}{n} \vee 1\right)$$

$\square$

### E.4 Bounds for Sum of Functions with $\tanh$

**Lemma E.13.** (Upper-Bound for $[\tanh(\alpha x + \nu) - \tanh \nu] x$) Suppose $\{x_i\}_{i=1}^n \overset{\text{iid}}{\sim} K_0(|x|)/\pi$, $\alpha \geq 0$, then we have

$$\left| \left( \frac{1}{n} \sum_{i=1}^n - \mathbb{E} \right) [\tanh(\alpha x_i + \nu) - \tanh \nu] x_i \right| = \frac{\alpha}{1 + |\nu|} \Theta \left( \sqrt{\frac{\log \frac{1}{\delta}}{n}} \vee \frac{\log^2 \frac{1}{\delta'}}{n} \right)$$

with probability at least $1 - (\delta + \delta')$.

*Proof.* By invoking the upper bound for tanh in proven Lemma in Subsection E.2,

$$|[\tanh(\alpha x_i + \nu) - \tanh \nu] x_i| \leq \frac{2\alpha}{1 + |\nu|} x_i^2$$

If $\alpha = 0$, then Lemma is valid, let's assume $\alpha > 0$ and define $Z_i := [\tanh(\alpha x_i + \nu) - \tanh \nu] x_i / \left( \frac{2\alpha}{1 + |\nu|} \right)$, with $|Z_i| \leq x_i^2$.

By using the expectation with $x^{2p}$ and $K_0(x)$ in Lemma of Subsection A.1, we obtain Khintchine's inequality $\|Z_i\|_p \leq \|x_i^2\|_p \leq 2^2 \Gamma(2p + \frac{1}{2})^{1/p} \Gamma \left( \frac{1}{2} \right)^{-1/p} \lesssim p^2$ and $|\mathbb{E}[Z_i]| \leq \mathbb{E}|Z_i| \leq \mathbb{E}[x_i^2] = 1$, $\|Z_i - \mathbb{E}[Z_i]\|_p \lesssim p^2 + 1 \lesssim p^2$ for $p \geq 2$, applying Corollary 38 on page 55 of (Ahle et al., 2020) int the second step.

$$\begin{aligned}
\mathbb{P} \left\{ \frac{1}{n} \sum_{i=1}^n Z_i \geq \mathbb{E}[Z_i] + \varepsilon \right\} &\leq \inf_{p > 0} (n\varepsilon)^{-p} \mathbb{E} \left[ \sum_{i=1}^n (Z_i - \mathbb{E}[Z_i]) \right]^p = \inf_{p > 0} (n\varepsilon)^{-p} \left| \sum_{i=1}^n Z_i - \mathbb{E}[Z_i] \right|_p^p \\
&\lesssim \inf_{p \geq 2} (n\varepsilon)^{-p} \max \left\{ \sqrt{pn}, (n/p)^{1/p} p^2 \right\}^p \\
&\leq \inf_{p \geq 2} \left( \frac{\sqrt{n}\varepsilon}{\sqrt{p}} \right)^{-p} + \inf_{p \geq 2} \frac{n}{p} \left( \frac{n\varepsilon}{p^2} \right)^{-p} \\
&\leq \exp \left( -\frac{n\varepsilon^2}{2e} \right) + (n\varepsilon)^{\frac{1}{2}} \exp \left( 1 - 2\frac{(n\varepsilon)^{\frac{1}{2}}}{e} \right)
\end{aligned}$$

The last step is achieved by taking $p = \frac{n\varepsilon^2}{e}, p = \frac{(n\varepsilon)^{\frac{1}{2}}}{e}$ for these two above terms respectively, and $\varepsilon \geq \max \left\{ \sqrt{\frac{2e}{n}}, \frac{(2e)^2}{n} \right\}$.

By letting $\varepsilon = \Theta \left( \sqrt{\frac{\log \frac{1}{\delta}}{n}} \vee \frac{\log^2 \frac{1}{\delta'}}{n} \right)$, taking the bounds for two sides, then with probability at least $1 - (\delta + \delta')$

$$\left| \left( \frac{1}{n} \sum_{i=1}^n - \mathbb{E} \right) [\tanh(\alpha x_i + \nu) - \tanh \nu] x_i \right| = \frac{2\alpha}{1 + |\nu|} \left| \left( \frac{1}{n} \sum_{i=1}^n - \mathbb{E} \right) Z_i \right| \leq \frac{\alpha}{1 + |\nu|} \Theta \left( \sqrt{\frac{\log \frac{1}{\delta}}{n}} \vee \frac{\log^2 \frac{1}{\delta'}}{n} \right)$$

$\square$

**Lemma E.14.** Let $\{x_i\}_{i=1}^n \overset{\text{iid}}{\sim} \mathcal{N}(0, 1)$, $\{x_i'\}_{i=1}^n \overset{\text{iid}}{\sim} \mathcal{N}(0, 1)$, $\{\vec{x}_i\}_{i=1}^n \overset{\text{iid}}{\sim} \mathcal{N}(0, I_d)$, and $\alpha \geq 0$, then for $n \gtrsim d \vee \log \frac{1}{\delta}$, with probability at least $1 - \delta$,

$$\left\| \frac{1}{n} \sum_{i=1}^n \tanh(\alpha x_i x_i' + \nu) x_i \vec{x}_i \right\|_2 = \mathcal{O} \left( \sqrt{\frac{d}{n}} \vee \sqrt{\frac{\log \frac{1}{\delta}}{n}} \right)$$

and for $n \gtrsim d \vee \log \frac{1}{\delta} \vee \log^3 \frac{1}{\delta'}$, with probability at least $1 - (\delta + \delta')$

$$\left\| \frac{1}{n} \sum_{i=1}^n \tanh(\alpha x_i x_i' + \nu) x_i \vec{x}_i \right\|_2 = \left\{ \tanh |\nu| + \frac{\alpha}{1 + |\nu|} \right\} \mathcal{O} \left( \sqrt{\frac{d}{n}} \vee \sqrt{\frac{\log \frac{1}{\delta}}{n}} \right)$$

*Proof.* We decompose the sum of vectors into two parts.

$$\frac{1}{n}\sum_{i=1}^{n}\tanh(\alpha x_i x_i' + \nu)x_i\vec{x}_i \;=\; \tanh(\nu)\left[\frac{1}{n}\sum_{i=1}^{n}x_i\vec{x}_i\right] + \frac{1}{n}\sum_{i=1}^{n}[\tanh(\alpha x_i x_i' + \nu) - \tanh(\nu)]x_i\vec{x}_i$$

By the previous Lemma of bound for weighted Gaussian vectors in Subsection E.3, we upper-bound the $\ell_2$ norm of first term by

$$\tanh|\nu|\cdot\left\|\frac{1}{n}\sum_{i=1}^{n}x_i\vec{x}_i\right\|_2 \;\leq\; \tanh|\nu|\cdot\mathcal{O}\left(\sqrt{\frac{d}{n}}\vee\frac{\log\frac{1}{\delta}}{n}\vee\sqrt{\frac{\log\frac{1}{\delta}}{n}}\right)$$

with probability at least $1 - \delta/2$.

Let $(\vec{x}_1,\cdots,\vec{x}_n) = (\vec{x}_1',\cdots,\vec{x}_{d-1}')^\top$, $\vec{\gamma} := \{[\tanh(\alpha x_i x_i' + \nu) - \tanh(\nu)]x_i\}_{i=1}^n$, then $\left\{\frac{\vec{\gamma}^\top \vec{x}_j'}{\|\vec{\gamma}\|_2}\right\}_{j=1}^{d-1} \sim \mathcal{N}(0, I_{d-1})$

$$\left\|\frac{1}{n}\sum_{i=1}^{n}[\tanh(\alpha x_i x_i' + \nu) - \tanh(\nu)]x_i\vec{x}_i\right\|_2 \;=\; \frac{1}{n}\|\vec{\gamma}\|_2\cdot\left\|\left\{\frac{\vec{\gamma}^\top \vec{x}_j'}{\|\vec{\gamma}\|_2}\right\}_{j=1}^{d-1}\right\|_2$$

Hence, $\left\|\left\{\frac{\vec{\gamma}^\top \vec{x}_j'}{\|\vec{\gamma}\|_2}\right\}_{j=1}^{d-1}\right\|_2^2 \sim \chi^2(d)$, and by applying the upper bound for Chi-square r.v. $\left\|\left\{\frac{\vec{\gamma}^\top \vec{x}_j'}{\|\vec{\gamma}\|_2}\right\}_{j=1}^{d-1}\right\|_2 =$
$\mathcal{O}\left(\sqrt{d}\vee\sqrt{\log\frac{1}{\delta}}\right)$ with probability $1 - \delta/4$, and invoking the bound for tanh in proven Lemma in Subsection E.2.

$$\left\|\frac{1}{n}\sum_{i=1}^{n}[\tanh(\alpha x_i x_i' + \nu) - \tanh(\nu)]x_i\vec{x}_i\right\|_2 = \frac{1}{n}\|\vec{\gamma}\|_2\cdot\mathcal{O}\left(\sqrt{d}\vee\sqrt{\log\frac{1}{\delta}}\right)$$

$$= \sqrt{\frac{1}{n}\sum_{i=1}^{n}|\tanh(\alpha x_i x_i' + \nu) - \tanh(\nu)|^2 x_i^2}\cdot\mathcal{O}\left(\sqrt{\frac{d}{n}}\vee\sqrt{\frac{\log\frac{1}{\delta}}{n}}\right)$$

Note that $\sum_{i=1}^{n}x_i^2 \sim \chi^2(n)$, then by applying the upper bound for Chi-square r.v. again, and using $n \gtrsim \log\frac{1}{\delta}$, we have $\frac{1}{n}\sum_{i=1}^{n}x_i^2 = \mathcal{O}(1)$ with probability at least $1 - \delta/4$, and

$$\left\|\frac{1}{n}\sum_{i=1}^{n}[\tanh(\alpha x_i x_i' + \nu) - \tanh(\nu)]x_i\vec{x}_i\right\|_2 = \mathcal{O}\left(\sqrt{\frac{d}{n}}\vee\sqrt{\frac{\log\frac{1}{\delta}}{n}}\right)$$

Let $Z_i := \frac{x_i^2 + (x_i')^2}{2} \overset{\text{iid}}{\sim} \exp(-z)\mathbb{1}_{\geq 0}$, then $x_i^2 \leq 2Z_i$ and $[x_i x_i']^2 \leq Z_i^2$

$$\frac{1}{n}\sum_{i=1}^{n}x_i^2\cdot\left(\frac{\alpha}{1+|\nu|}\right)^2[x_i x_i']^2 \;\leq\; 2\left(\frac{\alpha}{1+|\nu|}\right)^2\times\frac{1}{n}\sum_{i=1}^{n}Z_i^3$$

By invoking the upper-bound for cubic of Exp r.v. in Lemma of Subsection E.3 and $n \gtrsim d\vee\log\frac{1}{\delta}\vee\log^3\frac{1}{\delta'}$, then $\frac{1}{n}\sum_{i=1}^{n}Z_i^3 = \mathcal{O}(1)$ with probability at least $1 - (\delta/4 + \delta')$ and

$$\left\|\frac{1}{n}\sum_{i=1}^{n}[\tanh(\alpha x_i x_i' + \nu) - \tanh(\nu)]x_i\vec{x}_i\right\|_2 \;\leq\; \frac{\alpha}{1+|\nu|}\cdot\mathcal{O}\left(\sqrt{\frac{d}{n}}\vee\sqrt{\frac{\log\frac{1}{\delta}}{n}}\right)$$

Combine the bound for two terms, we conclude that

$$\left|\frac{1}{n}\sum_{i=1}^{n}\tanh(\alpha x_i x_i' + \nu)x_i\vec{x}_i\right|_2 \;\leq\; \mathcal{O}\left(\sqrt{\frac{d}{n}}\vee\sqrt{\frac{\log\frac{1}{\delta}}{n}}\right)$$

$$\left|\frac{1}{n}\sum_{i=1}^{n}\tanh(\alpha x_i x_i' + \nu)x_i\vec{x}_i\right|_2 \;\leq\; \left\{\tanh|\nu| + \frac{\alpha}{1+|\nu|}\right\}\mathcal{O}\left(\sqrt{\frac{d}{n}}\vee\sqrt{\frac{\log\frac{1}{\delta}}{n}}\right)$$

hold with probability at least $1 - \delta$ and $1 - (\delta + \delta')$ respectively. $\qquad\square$

**Lemma E.15.** Let $\{x_i\}_{i=1}^n \overset{\text{iid}}{\sim} K_0(|x|)/\pi$ and $\alpha \geq 0$, then for $n \gtrsim \log\frac{1}{\delta}$, with probability at least $1 - \delta$,

$$\left|\left(\frac{1}{n}\sum_{i=1}^n -\mathbb{E}\right)\tanh(\alpha x_i + \nu)\right| = \mathcal{O}\left(\sqrt{\frac{\log\frac{1}{\delta}}{n}}\right)$$

and

$$\left|\left(\frac{1}{n}\sum_{i=1}^n -\mathbb{E}\right)\tanh(\alpha x_i + \nu)\right| = \frac{\alpha}{1 + |\nu|}\mathcal{O}\left(\sqrt{\frac{\log\frac{1}{\delta}}{n}}\right)$$

*Proof.* Note that $\tanh(\alpha x_i + \nu) \in [-1, 1]$ is bounded, then by applying the Lemma of Azuma-Hoeffding Inequality in Subsection E.2,

$$\left|\left(\frac{1}{n}\sum_{i=1}^n -\mathbb{E}\right)\tanh(\alpha x_i + \nu)\right| = \mathcal{O}\left(\sqrt{\frac{\log\frac{1}{\delta}}{n}}\right)$$

We can rewite

$$\left(\frac{1}{n}\sum_{i=1}^n -\mathbb{E}\right)\tanh(\alpha x_i + \nu) = \left(\frac{1}{n}\sum_{i=1}^n -\mathbb{E}\right)[\tanh(\alpha x_i + \nu) - \tanh\nu]$$

By invoking the Upper Bound for tanh in proven Lemma in Subsection E.2, $|\tanh(\alpha x_i + \nu) - \tanh\nu| \leq \frac{2\alpha}{1+|\nu|}|x_i|$, applying Concentration Inequality with Exponential Tail in proven Lemma in Subsection E.1 and $n \gtrsim \log\frac{1}{\delta}$.

$$\left|\left(\frac{1}{n}\sum_{i=1}^n -\mathbb{E}\right)[\tanh(\alpha x_i + \nu) - \tanh\nu]\right| = \frac{\alpha}{1 + |\nu|}\mathcal{O}\left(\sqrt{\frac{\log\frac{1}{\delta}}{n}}\right)$$

$\qquad\square$

**Lemma E.16.** Let $\{x_i\}_{i=1}^n \overset{\text{iid}}{\sim} K_0(|x|)/\pi$ and $\alpha \geq 0$,

then for $n \gtrsim \log\frac{1}{\delta}$, with probability at least $1 - \delta$,

$$\left|\left(\frac{1}{n}\sum_{i=1}^n -\mathbb{E}\right)\tanh(\alpha x_i + \nu)x_i\right| = \mathcal{O}\left(\sqrt{\frac{\log\frac{1}{\delta}}{n}}\right)$$

and for $n \gtrsim \log\frac{1}{\delta} \vee \log^2\frac{1}{\delta'}$, with probability at least $1 - (\delta + \delta')$,

$$\left|\left(\frac{1}{n}\sum_{i=1}^n -\mathbb{E}\right)\tanh(\alpha x_i + \nu)x_i\right| = \left\{\tanh|\nu| + \frac{\alpha}{1 + |\nu|}\right\}\mathcal{O}\left(\sqrt{\frac{\log\frac{1}{\delta}}{n}}\right).$$

*Proof.* We decompose the sum of vectors into two parts, and note that $\mathbb{E}[x_i] = 0$

$$\left(\frac{1}{n}\sum_{i=1}^n -\mathbb{E}\right)\tanh(\alpha x_i + \nu)x_i = \tanh(\nu)\frac{1}{n}\sum_{i=1}^n x_i + \left(\frac{1}{n}\sum_{i=1}^n -\mathbb{E}\right)[\tanh(\alpha x_i + \nu) - \tanh(\nu)]x_i$$

The first term is bounded $\left|\frac{1}{n}\sum_{i=1}^n x_i\right| = \mathcal{O}\left(\sqrt{\frac{\log\frac{1}{\delta}}{n}}\right)$ with probability at least $1 - \delta/2$, by invoking Concentration Inequality with Exponential Tail in proven Lemma in Subsection E.1 and $n \gtrsim \log\frac{1}{\delta}$.

$$\left|\tanh(\nu)\frac{1}{n}\sum_{i=1}^n x_i\right| \leq \tanh|\nu| \cdot \mathcal{O}\left(\sqrt{\frac{\log\frac{1}{\delta}}{n}}\right)$$

By $[\tanh(\alpha x_i + \nu) - \tanh(\nu)]x_i \leq 2|x_i|$, the second term is is bounded with probability at least $1 - \delta/2$

$$\left| \left( \frac{1}{n} \sum_{i=1}^{n} -\mathbb{E} \right) [\tanh(\alpha x_i + \nu) - \tanh(\nu)]x_i \right| = \mathcal{O}\left( \sqrt{\frac{\log \frac{1}{\delta}}{n}} \right)$$

By invoking the bound for $[\tanh(\alpha x + \nu) - \tanh \nu]x$ in proven Lemma in this Subsection E.4 and $n \gtrsim \log \frac{1}{\delta} \vee \log^2 \frac{1}{\delta'}$, with probability at least $1 - (\delta/2 + \delta')$

$$\left| \left( \frac{1}{n} \sum_{i=1}^{n} -\mathbb{E} \right) [\tanh(\alpha x_i + \nu) - \tanh \nu] \right| = \frac{\alpha}{1 + |\nu|} \mathcal{O}\left( \sqrt{\frac{\log \frac{1}{\delta}}{n}} \right)$$

Combine the bound for two terms, we conclude that

$$\left| \left( \frac{1}{n} \sum_{i=1}^{n} -\mathbb{E} \right) \tanh(\alpha x_i + \nu)x_i \right| = \mathcal{O}\left( \sqrt{\frac{\log \frac{1}{\delta}}{n}} \right)$$

$$\left| \left( \frac{1}{n} \sum_{i=1}^{n} -\mathbb{E} \right) \tanh(\alpha x_i + \nu)x_i \right| = \left\{ \tanh |\nu| + \frac{\alpha}{1 + |\nu|} \right\} \mathcal{O}\left( \sqrt{\frac{\log \frac{1}{\delta}}{n}} \right)$$

hold with probability at least $1 - \delta$ and $1 - (\delta + \delta')$ respectively. $\qquad\square$

# F  Bounds for Statistical Errors of Finite-Sample Level Analysis

## F.1  Proof for Statistical Error of Mixing Weights

**Theorem F.1.** (Proposition 6.3 in Section 6: Statistical Error of Mixing Weights)

Let $N(\theta, \nu)$ and $N_n(\theta, \nu)$ be the EM update rules for mixing weights $\tanh(\nu) := \pi(1) - \pi(2)$ at the population level and finite-sample level with $n$ samples, and $n \gtrsim \log \frac{1}{\delta}, \delta \in (0, 1)$, then

$$|N_n(\theta, \nu) - N(\theta, \nu)| = \min\left\{\frac{\|\theta\|/\sigma}{1 + |\nu|}, 1\right\} \mathcal{O}\left(\sqrt{\frac{\log \frac{1}{\delta}}{n}}\right)$$

with probability at least $1 - \delta$.

*Proof.* Since $N_n(\theta, \nu) := \frac{1}{n} \sum_{i=1}^{n} \tanh\left(\frac{y_i \langle x_i, \theta \rangle}{\sigma^2} + \nu\right)$ and $N(\theta, \nu) := \mathbb{E}_{s \sim p(s|\theta^*, \pi^*)} \tanh\left(\frac{y\langle x, \theta \rangle}{\sigma^2} + \nu\right)$, then let $\alpha := \|\theta\|/\sigma$, note that $y_i/\sigma \sim \mathcal{N}(0,1), \langle x_i, \theta \rangle/\|\theta\| \sim \mathcal{N}(0,1)$, then Normal Product Distribution is

$$Z_i := \frac{y_i}{\sigma} \times \frac{\langle x_i, \theta \rangle}{\|\theta\|} \sim \frac{K_0(|z|)}{\pi}$$

see also Page 50, Section 4.4 Bessel Function Distributions, Chapter 12 Continuous Distributions (General) of Johnson et al. (1970) for more information. Hence, we can rewrite the error as

$$N_n(\theta, \nu) - N(\theta, \nu) = \left(\frac{1}{n} \sum_{i=1}^{n} -\mathbb{E}\right) \tanh(\alpha Z_i + \nu) \quad \{Z_i\}_{i=1}^{n} \overset{\text{iid}}{\sim} \frac{K_0(|z|)}{\pi}$$

By invoking upper bound for sum of tanh in Subsection E.4, using $\alpha = \|\theta\|/\sigma, n \gtrsim \log \frac{1}{\delta}$, the proof is complete. $\square$

## F.2  Proof for Projected Error of Regression Parameters

**Theorem F.2.** (Projected Error of Regression Parameters)

Let $M(\theta, \nu)$ be the EM update rule for regression parameters $\theta$ at population level, $M_n^{\text{easy}}(\theta, \nu)$ be the easy version of finite-sample EM update rule for $\theta$ with $n$ samples, and if $n \gtrsim \log \frac{1}{\delta}, \delta \in (0, 1)$, then with probability at least $1 - \delta$, the projection on span$\{\theta\}$ for the statistical error satisfies

$$\left|\left\langle \frac{\theta}{\|\theta\|}, M_n^{\text{easy}}(\theta, \nu) - M(\theta, \nu)\right\rangle\right|/\sigma = \mathcal{O}\left(\sqrt{\frac{\log \frac{1}{\delta}}{n}}\right),$$

and if $n \gtrsim \log \frac{1}{\delta} \vee \log^2 \frac{1}{\delta'}, \delta' \in (0, 1)$, then with probability at least $1 - (\delta + \delta')$,

$$\left|\left\langle \frac{\theta}{\|\theta\|}, M_n^{\text{easy}}(\theta, \nu) - M(\theta, \nu)\right\rangle\right|/\sigma = \left\{\tanh |\nu| + \frac{\|\theta\|/\sigma}{1 + |\nu|}\right\} \mathcal{O}\left(\sqrt{\frac{\log \frac{1}{\delta}}{n}}\right).$$

*Proof.* Since $M_n^{\text{easy}}(\theta, \nu) := \frac{1}{n} \sum_{i=1}^{n} \tanh\left(\frac{y_i \langle x_i, \theta \rangle}{\sigma^2} + \nu\right) y_i x_i, M(\theta, \nu) := \mathbb{E}_{s \sim p(s|\theta^*, \pi^*)} \tanh\left(\frac{y\langle x, \theta \rangle}{\sigma^2} + \nu\right) yx$, then let $\alpha := \|\theta\|/\sigma$, note that $y_i/\sigma \sim \mathcal{N}(0,1), \langle x_i, \theta \rangle/\|\theta\| \sim \mathcal{N}(0,1)$, then Normal Product Distribution is

$$Z := \frac{y_i}{\sigma} \times \frac{\langle x_i, \theta \rangle}{\|\theta\|} \sim \frac{K_0(|z|)}{\pi}$$

see also Page 50, Section 4.4 Bessel Function Distributions, Chapter 12 Continuous Distributions (General) of for more information. Hence, we can rewrite the projected statistical error as

$$\left\langle \frac{\theta}{\|\theta\|}, M_n(\theta,\nu) - M(\theta,\nu) \right\rangle / \sigma = \left( \frac{1}{n} \sum_{i=1}^{n} - \mathbb{E} \right) \tanh(\alpha Z_i + \nu) Z_i \quad \{Z_i\}_{i=1}^{n} \overset{\text{iid}}{\sim} \frac{K_0(|z|)}{\pi}$$

Apply the upper bound for sum of $\tanh(\alpha Z_i + \nu) Z_i$ in Subsection E.2, the proof is complete. $\qquad\square$

### F.3 Proof for Statistical Error of Regression Parameters for Easy EM

**Theorem F.3.** (Statistical Error of Regression Parameters for Easy EM)

Let $M(\theta,\nu)$ be the EM update rule for regression parameters $\theta$ at population level, $M_n^{\text{easy}}(\theta,\nu)$ be the easy version of finite-sample EM update rule for $\theta$ with $n$ samples, and if $n \gtrsim d \vee \log \frac{1}{\delta}, \delta \in (0,1)$, then with probability at least $1 - \delta$,

$$\|M_n^{\text{easy}}(\theta,\nu) - M(\theta,\nu)\|/\sigma = \mathcal{O}\left( \sqrt{\frac{d \vee \log \frac{1}{\delta}}{n}} \right),$$

and if $n \gtrsim d \vee \log \frac{1}{\delta} \vee \log^3 \frac{1}{\delta'}, \delta' \in (0,1)$, then with probability at least $1 - (\delta + \delta')$,

$$\|M_n^{\text{easy}}(\theta,\nu) - M(\theta,\nu)\|/\sigma = \left\{ \tanh|\nu| + \frac{\|\theta\|/\sigma}{1 + |\nu|} \right\} \mathcal{O}\left( \sqrt{\frac{d \vee \log \frac{1}{\delta}}{n}} \right).$$

*Proof.* Since $M_n^{\text{easy}}(\theta,\nu) := \frac{1}{n} \sum_{i=1}^{n} \tanh\left( \frac{y_i \langle x_i, \theta \rangle}{\sigma^2} + \nu \right) y_i x_i, M(\theta,\nu) := \mathbb{E}_{s \sim p(s|\theta^*, \pi^*)} \tanh\left( \frac{y \langle x, \theta \rangle}{\sigma^2} + \nu \right) yx,$ then let $\alpha := \|\theta\|/\sigma$, and decompose $x_i = Z_i \theta + P_\theta^\perp \vec{Z_i}$ into two parts in two subspaces $\text{span}\{\theta\}$ and $\text{span}\{\theta\}^\perp$, where $Z_i \sim \mathcal{N}(0,1), \vec{Z_i} \sim \mathcal{N}(0, I_{d-1})$, and the orthogonal projection matrix $P_\theta^\perp$ satisfies $\text{span}(P_\theta^\perp) = \text{span}\{\theta\}^\perp$. The projection matrix has the following properties: $P_\theta^\perp \theta = \vec{0}, (P_\theta^\perp)^2 = P_\theta^\perp = (P_\theta^\perp)^\top$.

$$(M_n^{\text{easy}}(\theta,\nu) - M(\theta,\nu))/\sigma = \left[ \left\langle \frac{\theta}{\|\theta\|}, M_n^{\text{easy}}(\theta,\nu) - M(\theta,\nu) \right\rangle / \sigma \right] \frac{\theta}{\|\theta\|} + P_\theta^\perp (M_n^{\text{easy}}(\theta,\nu) - M(\theta,\nu))/\sigma$$

The $\ell_2$ norm of the first term (projected statistical error) is bounded in Proposition of of the projected statistical error. Let's focus on the second term, and use the notation $\alpha := \|\theta\|/\sigma, y/\sigma = Z_i' \sim \mathcal{N}(0,1)$, then

$$P_\theta^\perp (M_n^{\text{easy}}(\theta,\nu) - M(\theta,\nu))/\sigma = P_\theta^\perp \left( \frac{1}{n} \sum_{i=1}^{n} - \mathbb{E} \right) \tanh(\alpha Z_i Z_i' + \nu) Z_i \vec{Z_i}$$

where $\{Z_i\}_{i=1}^{n}, \{Z_i'\}_{i=1}^{n} \overset{\text{iid}}{\sim} \mathcal{N}(0,1), \vec{Z_i} \sim \mathcal{N}(0, I_{d-1})$ are independent of each other.

By applying $\|P_\theta^\perp\|_2 = 1$ and the upper bound for sum of $\tanh(\alpha Z_i Z_i' + \nu) Z_i \vec{Z_i}$ in proven Lemma in Subsection E.4, we complete the proof. $\qquad\square$

### F.4 Proof for Statistical Error of Regression Parameters

**Theorem F.4.** (Proposition 6.4 in Section 6: Statistical Error of Regression Parameters)

Let $M(\theta, \nu)$ and $M_n(\theta, \nu)$ be the EM update rules for regression parameters $\theta$ at the population level and finite-sample level with $n$ samples, and if $n \gtrsim d \vee \log \frac{1}{\delta}, \delta \in (0, 1)$, then with probability at least $1 - \delta$,

$$\|M_n(\theta, \nu) - M(\theta, \nu)\|/\sigma = \mathcal{O}\left( \sqrt{\frac{d \vee \log \frac{1}{\delta}}{n}} \right),$$

and if $n \gtrsim d \vee \log \frac{1}{\delta} \vee \log^3 \frac{1}{\delta'}, \delta' \in (0, 1)$, then with probability at least $1 - (\delta + \delta')$,

$$\|M_n(\theta, \nu) - M(\theta, \nu)\|/\sigma = \{\tanh |\nu| + \|\theta\|/\sigma\}\mathcal{O}\left( \sqrt{\frac{d \vee \log \frac{1}{\delta}}{n}} \right).$$

*Proof.* By using the connection between $M_n$ and $M_n^{\text{easy}}$, and decomposing the error into two terms,

$$
\begin{aligned}
(M_n(\theta, \nu) - M(\theta, \nu))/\sigma &= \left( \frac{\sum_{i=1}^n x_i x_i^\top}{n} \right)^{-1} (M_n^{\text{easy}}(\theta, \nu) - M(\theta, \nu))/\sigma \\
&\quad - \left( \frac{\sum_{i=1}^n x_i x_i^\top}{n} \right)^{-1} \left( \frac{\sum_{i=1}^n x_i x_i^\top}{n} - I_d \right) M(\theta, \nu)/\sigma
\end{aligned}
$$

By using Operator norm bounds for the standard Gaussian ensemble in Lemma of Subsection E.3, and $n \gtrsim d \vee \log \frac{1}{\delta}$

$$\left\| \left( \frac{\sum_{i=1}^n x_i x_i^\top}{n} \right)^{-1} \right\|_2 = \mathcal{O}(1) \qquad \left\| \frac{\sum_{i=1}^n x_i x_i^\top}{n} - I_d \right\|_2 = \mathcal{O}\left( \sqrt{\frac{d \vee \log \frac{1}{\delta}}{n}} \right)$$

By invoking the proven upper bound of Statistical Error of Regression Parameters for Easy EM, and using the facts that $\|M(\theta, \nu)\| \le \|\theta\|$ is nonincreasing and bounded $\|M(\theta, \nu)\|/\sigma = \mathcal{O}(1)$ in Subsection B.2, the bounds in this proposition are established. $\qquad \square$

# G  Proofs for Results of Finite-Sample Level Analysis

## G.1  Proof for Initialization at Finite-Sample Level

**Theorem G.1.** (Fact 6.2 in Section 6: Initialization at Finite-Sample Level)

Let $\alpha^t := \|\theta^t\|/\sigma = \|M_n(\theta^{t-1}, \nu^{t-1})\|/\sigma$ and $\beta^t := \tanh(\nu^t) = N_n(\theta^{t-1}, \nu^{t-1})$ for all $t \in \mathbb{Z}_+$ be the $t$-th iteration of the EM update rules $\|M_n(\theta, \nu)\|/\sigma$ and $N_n(\theta, \nu)$ at the finite-sample level. If we run EM at the finite-sample level for at most $T_0 = \mathcal{O}(1)$ iterations with $n = \Omega\left(d \vee \log \frac{1}{\delta}\right)$ samples, then $\alpha^{T_0} < 0.1$ with probability at least $1 - \delta$.

*Proof.* For brevity, we write $\bar{\alpha}^t := \|M(\theta^{t-1}, \nu^{t-1})\|/\sigma$ for the EM update rule at population level. By invoking the bound for the statistical error of regression parameters in Subsection F.4, and selecting $n \geq (2000C)^2 \left[d \vee \log \frac{1}{\delta}\right]$ for some constant $C$

$$|\alpha^t - \bar{\alpha}^t| \leq C\sqrt{\frac{d \vee \log \frac{1}{\delta}}{n}} \leq \frac{1}{2} \times 10^{-3}$$

If $\alpha^t < 0.1$, then it satisfies the condition; otherwise, by invoking the population EM update rule $\bar{\alpha}^{t+1} = m(\alpha^t, \nu^t) = \mathbb{E}[\tanh(\alpha^t X + \nu^t)X]$ in equation 10, and the monotonicity of $m(\alpha, \nu)$ in Fact 4.2 of Section 4,

$$\alpha^{t+1} \leq \bar{\alpha}^{t+1} + |\alpha^{t+1} - \bar{\alpha}^{t+1}| = m(\alpha^t, \nu^t) + \frac{1}{2} \times 10^{-3} \leq m(\alpha^t, 0) + \frac{1}{2} \times 10^{-3} = m_0(\alpha^t) + \frac{1}{2} \times 10^{-3}$$

Applying the monotonicity of $m(\alpha, \nu)$ in Fact 4.2 of Section 4, which is also applicable to $m_0(\alpha) = m(\alpha, 0)$

$$\alpha^1 \leq m_0(\alpha^0) + \frac{1}{2} \times 10^{-3} \leq m_0(\infty) + \frac{1}{2} \times 10^{-3} = \frac{2}{\pi} + \frac{1}{2} \times 10^{-3} < 0.64$$

$$\alpha^2 \leq m_0(\alpha^1) + \frac{1}{2} \times 10^{-3} \leq m_0(0.64) + \frac{1}{2} \times 10^{-3} < 0.4$$

$$\alpha^3 \leq m_0(\alpha^2) + \frac{1}{2} \times 10^{-3} \leq m_0(0.4) + \frac{1}{2} \times 10^{-3} < 0.31$$

Furthermore, by applying the upper bound for expectation $m_0(\alpha) = m(\alpha, 0) = \mathbb{E}[\tanh(\alpha X)X]$ under the density function $X \sim K_0(|x|)/\pi$ in Subsection A.3 for $t \geq 3$

$$\alpha^{t+1} \leq m_0(\alpha^t) + \frac{1}{2} \times 10^{-3} \leq \alpha^t - \frac{3[\alpha^t]^3}{1 + 8[\alpha^t]} + \frac{1}{2} \times 10^{-3}$$

By using $\alpha^t \geq 0.1$, then $r^t := \alpha^t - \frac{1}{20} \geq \frac{1}{20}$ and

$$r^{t+1} \leq r^t - \frac{5}{3}\left[r^t + \frac{1}{20}\right]^3 + \frac{1}{2} \times 10^{-3} < r^t - \frac{5}{3}[r^t]^3 < r^t - [r^t]^3$$

Hence, by using $0 < r^{t+1} \leq r^t$, we obtain $2 \leq \left(\frac{r^t}{r^{t+1}}\right)^2 + \left(\frac{r^t}{r^{t+1}}\right)$

$$2(T_0 - 3) \leq \sum_{t=3}^{T_0-1} \frac{r^t - r^{t+1}}{\frac{1}{2}[r^t]^3} \leq \sum_{t=3}^{T_0-1}\left\{[r^t]^{-3}\left[\left(\frac{r^t}{r^{t+1}}\right)^2 + \left(\frac{r^t}{r^{t+1}}\right)\right]\right\}\{r^t - r^{t+1}\} = [r^{T_0}]^{-2} - [r^3]^{-2}$$

By selecting $T_0 = 196 = \mathcal{O}(1)$ and using $r^3 := \alpha^3 - \frac{1}{20} < 0.31 - \frac{1}{20}$, we have

$$\alpha^{T_0} = r^{T_0} + \frac{1}{20} \leq \frac{1}{\sqrt{2(T_0 - 3) + [r^3]^{-2}}} + \frac{1}{20} < \frac{1}{20} + \frac{1}{20} = 0.1$$

By substituting $\delta/T_0 \to \delta$ in the above expressions, then $\alpha^{T_0} < 0.1$ with probability at least $1 - \delta$. $\qquad\square$

## G.2 Proof for Convergence Rate at Finite-Sample Level

**Theorem G.2.** (Theorem 6.1 in Section 6: Convergence Rate at Finite-Sample Level for Fixed Mixing Weights) Suppose a 2MLR model is fitted to the overspecified model with no separation $\theta^* = \vec{0}$, given mixing weights $\pi^t = \pi^0$ and $n = \Omega\left(d \vee \log\frac{1}{\delta} \vee \log^3\frac{1}{\delta'}\right)$ samples:

(sufficiently unbalanced) if $\left\|\pi^0 - \frac{1}{2}\right\|_1 \gtrsim \left[\frac{d\vee\log\frac{1}{\delta}}{n}\right]^{\frac{1}{4}}$, finite-sample EM takes at most $T = \mathcal{O}\left(\left\|\pi^0 - \frac{1}{2}\right\|_1^{-2}\log\frac{n}{d\vee\log\frac{1}{\delta}}\right)$ iterations to achieve $\|\theta^T\|/\sigma = \mathcal{O}\left(\left\|\pi^0 - \frac{1}{2}\right\|_1^{-1}\left[\frac{d\vee\log\frac{1}{\delta}}{n}\right]^{\frac{1}{2}}\right)$,

(sufficiently balanced) if $\left\|\pi^0 - \frac{1}{2}\right\|_1 \lesssim \left[\frac{d\vee\log\frac{1}{\delta}}{n}\right]^{\frac{1}{4}}$, finite-sample EM takes at most $T = \mathcal{O}\left(\left[\frac{n}{d\vee\log\frac{1}{\delta}}\right]^{\frac{1}{2}}\right)$ iterations to achieve $\|\theta^T\|/\sigma = \mathcal{O}\left(\left[\frac{d\vee\log\frac{1}{\delta}}{n}\right]^{\frac{1}{4}}\right)$, with probability at least $1 - T(\delta + \delta')$.

*Proof.* By invoking the fact for initialization at finite-sample level in Subsection G.1, then $\alpha^{T_0} = \|\theta^{T_0}\|/\sigma < 0.1$ after running population EM for at most $T_0 = \mathcal{O}(1)$ iterations. Without loss of generality, we may assume $\alpha^0 = \|\theta^0\|/\sigma \in (0, 0.1)$ in the following discussions.

For brevity, we write the output of the EM update rule for regression parameters at Population level as $\bar{\alpha}^{t+1} = \|M(\theta^t, \nu^t)\|/\sigma = \|M(\theta^t, \nu^0)\|/\sigma$, and output of the EM update rules for regression parameters at Finite-smaple level as $\alpha^{t+1} = \|\theta^{t+1}\|/\sigma = \|M_n(\theta^t, \nu^t)\|/\sigma = \|M_n(\theta^t, \nu^0)\|/\sigma$.

Without the loss of generality, we assume $\beta^t = \pi^t(1) - \pi^t(2) \geq 0$, then by invoking the inequality in Appendix C for $\alpha^t \leq 0.1$

$$\frac{\bar{\alpha}^{t+1}}{\alpha^t(1 - [\beta^t]^2)} \leq 1 - \frac{5}{3}[\alpha^t]^2 + 9.53[\alpha^t]^2[\beta^t]^2$$

and the upper bound for statistical error in Subsection F.4, namely $|\alpha^{t+1} - \bar{\alpha}^{t+1}| \leq c^2(\beta^t + \alpha^t)\sqrt{\frac{d\vee\log\frac{1}{\delta}}{n}}$ for some univeral constant $c$ when $n \gtrsim d \vee \log\frac{1}{\delta} \vee \log^3\frac{1}{\delta'}$, we have

$$
\begin{aligned}
\alpha^{t+1} &\leq |\alpha^{t+1} - \bar{\alpha}^{t+1}| + \bar{\alpha}^{t+1} \\
&\leq \alpha^t(1 - [\beta^t]^2)\left(1 - \frac{5}{3}[\alpha^t]^2 + 9.53[\alpha^t]^2[\beta^t]^2\right) + c^2(\beta^t + \alpha^t)\sqrt{\frac{d\vee\log\frac{1}{\delta}}{n}} \\
&= \alpha^t\left(1 - [\beta^t]^2 + c^2\sqrt{\frac{d\vee\log\frac{1}{\delta}}{n}}\right) + c^2\beta^t\sqrt{\frac{d\vee\log\frac{1}{\delta}}{n}} - [\alpha^t]^3(1 - [\beta^t]^2)\left(\frac{5}{3} - 9.53[\beta^t]^2\right)
\end{aligned}
$$

with the probability at least $1 - (\delta + \delta')$.

**Proof for part a) (sufficiently unbalanced)** $\left\|\pi^0 - \frac{1}{2}\right\|_1 \gtrsim \left[\frac{d \vee \log \frac{1}{\delta}}{n}\right]^{\frac{1}{4}}$

Consider the following two cases.

(i) if $|\beta^t| = |\beta^0| = \left\|\pi^0 - \frac{1}{2}\right\|_1 \geq 2c\left[\frac{d \vee \log \frac{1}{\delta}}{n}\right]^{\frac{1}{4}}$, $|\beta^t| > 0.4$, by invoking $-(1-[\beta^t]^2)\left(\frac{5}{3} - 9.53[\beta^t]^2\right) < \frac{5}{3}, |\beta^t| \leq 1$

$$\alpha^{t+1} \leq \alpha^t\left(1 - 0.4^2 + c^2\sqrt{\frac{d \vee \log \frac{1}{\delta}}{n}} + \frac{5}{3}[\alpha^t]^2\right) + c^2\sqrt{\frac{d \vee \log \frac{1}{\delta}}{n}}$$

By selecting $n \geq (5c)^4\left[d \vee \log \frac{1}{\delta}\right]$, then $1 - 0.4^2 + c^2\sqrt{\frac{d \vee \log \frac{1}{\delta}}{n}} + \frac{5}{3}[\alpha^t]^2 < 0.9$ for $\alpha^t \leq 0.1$, therefore

$$\alpha^{t+1} - 10c^2\sqrt{\frac{d \vee \log \frac{1}{\delta}}{n}} \leq 0.9\left(\alpha^t - 10c^2\sqrt{\frac{d \vee \log \frac{1}{\delta}}{n}}\right)$$

Furthermore, by letting $n \geq (10c)^4\left[d \vee \log \frac{1}{\delta}\right]$, then $10c^2\sqrt{\frac{d \vee \log \frac{1}{\delta}}{n}} \leq 0.1$ and $\alpha^{t+1} \leq 0.1$, hence

$$\alpha^T \leq 10c^2\sqrt{\frac{d \vee \log \frac{1}{\delta}}{n}} + 0.9^T\left(\alpha^0 - 10c^2\sqrt{\frac{d \vee \log \frac{1}{\delta}}{n}}\right) \leq 10c^2\sqrt{\frac{d \vee \log \frac{1}{\delta}}{n}} + 0.1 \times 0.9^T$$

Let's choose $T = \left\lceil\frac{\frac{1}{2}\log\frac{n}{d \vee \log \frac{1}{\delta}} - 2\log c - \log 10}{\log\frac{1}{0.9}}\right\rceil_+ = \Theta\left(\log\frac{n}{d \vee \log \frac{1}{\delta}}\right)$, then

$$\alpha^T \leq 10c^2\sqrt{\frac{d \vee \log \frac{1}{\delta}}{n}} + c^2\sqrt{\frac{d \vee \log \frac{1}{\delta}}{n}} = 11c^2\sqrt{\frac{d \vee \log \frac{1}{\delta}}{n}} = \Theta\left(\sqrt{\frac{d \vee \log \frac{1}{\delta}}{n}}\right)$$

(ii) if $|\beta^t| = |\beta^0| = \left\|\pi^0 - \frac{1}{2}\right\|_1 \geq 2c\left[\frac{d \vee \log \frac{1}{\delta}}{n}\right]^{\frac{1}{4}}$, $|\beta^t| \leq 0.4$, by invoking $-(1 - [\beta^t]^2)\left(\frac{5}{3} - 9.53[\beta^t]^2\right) < -0.1, |\beta^t| \leq 0.4$

$$\alpha^{t+1} \leq \alpha^t\left(1 - \frac{3}{4}[\beta^t]^2\right) + c^2\beta^t\sqrt{\frac{d \vee \log \frac{1}{\delta}}{n}} - 0.1[\alpha^t]^3$$

Hence,

$$\alpha^{t+1} - \frac{4}{3}\frac{c^2}{\beta^0}\sqrt{\frac{d \vee \log \frac{1}{\delta}}{n}} \leq \left(1 - \frac{3}{4}[\beta^0]^2\right)\left(\alpha^t - \frac{4}{3}\frac{c^2}{\beta^0}\sqrt{\frac{d \vee \log \frac{1}{\delta}}{n}}\right)$$

Furthermore, by letting $n \geq \left(\frac{20}{3}c\right)^4\left[d \vee \log \frac{1}{\delta}\right]$, then $\frac{4}{3}\frac{c^2}{\beta^0}\sqrt{\frac{d \vee \log \frac{1}{\delta}}{n}} \leq \frac{2}{3}c\left[\frac{d \vee \log \frac{1}{\delta}}{n}\right]^{\frac{1}{4}} \leq 0.1$

$$\alpha^T \leq \frac{4}{3}\frac{c^2}{\beta^0}\sqrt{\frac{d \vee \log \frac{1}{\delta}}{n}} + 0.1\left(1 - \frac{3}{4}[\beta^0]^2\right)^T$$

Let's choose $T = \left\lceil\frac{\frac{1}{2}\log\frac{n}{d \vee \log \frac{1}{\delta}} - \log\frac{1}{\beta^0} + \log\frac{3}{2c^2}}{-\log\left(1 - \frac{3}{4}[\beta^0]^2\right)}\right\rceil_+ = \Theta\left([\beta^0]^{-2}\log\frac{n}{d \vee \log \frac{1}{\delta}}\right)$ for $|\beta^0| \leq 0.4$, here we use that fact $-\log(1 - \frac{3}{4}[\beta^0]^2) = \Theta([\beta^0]^2)$ for small $\beta^0$, then

$$\alpha^T \leq \frac{4}{3}\frac{c^2}{\beta^0}\sqrt{\frac{d \vee \log \frac{1}{\delta}}{n}} + \frac{2}{3}\frac{c^2}{\beta^0}\sqrt{\frac{d \vee \log \frac{1}{\delta}}{n}} = \frac{2c^2}{\beta^0}\sqrt{\frac{d \vee \log \frac{1}{\delta}}{n}}$$

To sum up, by combining case (i) and case (ii), we show that with $n = \Omega\left(d \vee \log \frac{1}{\delta} \vee \log^3\frac{1}{\delta'}\right)$ and $|\beta^0| = \left\|\pi^0 - \frac{1}{2}\right\|_1 \geq 2c\left[\frac{d \vee \log \frac{1}{\delta}}{n}\right]^{\frac{1}{4}}, \alpha^0 = \|\theta^0\|/\sigma \leq 0.1$, if we run finite-sample EM for at most $T = \mathcal{O}\left([\beta^0]^{-2}\log\frac{n}{d \vee \log \frac{1}{\delta}}\right)$ iterations, then $\alpha^T = \|\theta^T\|/\sigma = \mathcal{O}\left(\frac{1}{\beta^0}\sqrt{\frac{d \vee \log \frac{1}{\delta}}{n}}\right)$ with probability at least $1 - T(\delta + \delta')$.

**Proof for part b) (sufficiently balanced)** $\left\| \pi^0 - \frac{1}{2} \right\|_1 \lesssim \left[ \frac{d \vee \log \frac{1}{\delta}}{n} \right]^{\frac{1}{4}}$

If $|\beta^t| = |\beta^0| = \left\| \pi^0 - \frac{1}{2} \right\|_1 \leq 2c \left[ \frac{d \vee \log \frac{1}{\delta}}{n} \right]^{\frac{1}{4}}$, let $n \geq (10c)^4 \left[ d \vee \log \frac{1}{\delta} \right]$, then $\beta^t \leq \frac{1}{5}, 1 + \frac{1}{4(1-[\beta^t]^2)} < \frac{5}{3} - 9.53[\beta^t]^2$,

$$\alpha^{t+1} \leq \alpha^t \left( c^2 \sqrt{\frac{d \vee \log \frac{1}{\delta}}{n}} - \frac{[\alpha^t]^2}{4} + (1 - [\alpha^t]^2)(1 - [\beta^t]^2) \right) + c^2 \beta^t \sqrt{\frac{d \vee \log \frac{1}{\delta}}{n}}$$

If $\alpha^t \leq 2c \left[ \frac{d \vee \log \frac{1}{\delta}}{n} \right]^{\frac{1}{4}}$, then the goal $\alpha^t = \mathcal{O}\left( \left[ \frac{d \vee \log \frac{1}{\delta}}{n} \right]^{\frac{1}{4}} \right)$ is achieved; otherwise, we have $\alpha^t > 2c \left[ \frac{d \vee \log \frac{1}{\delta}}{n} \right]^{\frac{1}{4}}$,

$$\alpha^{t+1} \leq (1 - [\beta^t]^2)\alpha^t(1 - [\alpha^t]^2) + c^2 \beta^t \sqrt{\frac{d \vee \log \frac{1}{\delta}}{n}}$$

(i) if $\alpha^t > \frac{2c^2}{\beta^0} \sqrt{\frac{d \vee \log \frac{1}{\delta}}{n}}$, note that $\alpha^t(1 - [\alpha^t]^2) \geq \frac{\alpha^t}{2}$ for $\alpha^t \leq 0.1 < \frac{\sqrt{2}}{2}$, then

$$\alpha^{t+1} \leq \alpha^t(1 - [\alpha^t]^2)$$

(ii) if $\alpha^t \leq \frac{2c^2}{\beta^0} \sqrt{\frac{d \vee \log \frac{1}{\delta}}{n}}$, then by invoking $1 - [\beta^t]^2 \leq 1$ and $\alpha^t > 2c \left[ \frac{d \vee \log \frac{1}{\delta}}{n} \right]^{\frac{1}{4}}$, we have

$$\alpha^{t+1} \leq \alpha^t(1 - [\alpha^t]^2) + \frac{2c^4}{\alpha^t} \frac{d \vee \log \frac{1}{\delta}}{n} \leq \alpha^t(1 - [\alpha^t]^2) + 2c^3 \left[ \frac{d \vee \log \frac{1}{\delta}}{n} \right]^{\frac{3}{4}} \leq \alpha^t \left( 1 - \frac{3}{4}[\alpha^t]^2 \right)$$

To sum up, by combining case (i) and case (ii), the following inequality holds for $\alpha^t > 2c \left[ \frac{d \vee \log \frac{1}{\delta}}{n} \right]^{\frac{1}{4}}$

$$\alpha^{t+1} \leq \alpha^t \left( 1 - \frac{3}{4}[\alpha^t]^2 \right)$$

Hence, by using $0 < \alpha^{t+1} \leq \alpha^t$, we obtain $1 \leq \frac{1}{2} \left( \frac{\alpha^t}{\alpha^{t+1}} \right)^2 + \frac{1}{2} \left( \frac{\alpha^t}{\alpha^{t+1}} \right)$ and

$$
\begin{aligned}
T = \sum_{t=0}^{T-1} 1 &\leq \sum_{t=0}^{T-1} \frac{\alpha^t - \alpha^{t+1}}{\frac{3}{4}[\alpha^t]^3} \leq \frac{4}{3} \times \frac{1}{2} \sum_{t=0}^{T-1} \frac{\alpha^t + \alpha^{t+1}}{[\alpha^t]^2[\alpha^{t+1}]^2}(\alpha^t - \alpha^{t+1}) \\
&= \frac{2}{3} \sum_{t=0}^{T-1} ([\alpha^{t+1}]^{-2} - [\alpha^t]^{-2}) = \frac{2}{3}([\alpha^T]^{-2} - [\alpha^0]^{-2})
\end{aligned}
$$

Namely,

$$\alpha^T \leq \frac{1}{\sqrt{\frac{3}{2}T + [\alpha^0]^{-2}}} \leq \frac{1}{\sqrt{\frac{3}{2}T}}$$

Let's choose $T = \lceil \frac{1}{6c^2} \left[ \frac{n}{d \vee \log \frac{1}{\delta}} \right]^{\frac{1}{2}} \rceil_+ = \Theta\left( \left[ \frac{n}{d \vee \log \frac{1}{\delta}} \right]^{\frac{1}{2}} \right)$, then

$$\alpha^T \leq \frac{1}{\sqrt{\frac{3}{2}T}} \leq 2c \left[ \frac{d \vee \log \frac{1}{\delta}}{n} \right]^{\frac{1}{4}}$$

Therefore, we show that with $n = \Omega\left( d \vee \log \frac{1}{\delta} \vee \log^3 \frac{1}{\delta'} \right)$ and $|\beta^0| = \left\| \pi^0 - \frac{1}{2} \right\|_1 \leq 2c \left[ \frac{d \vee \log \frac{1}{\delta}}{n} \right]^{\frac{1}{4}}, \alpha^0 = \|\theta^0\|/\sigma \leq 0.1$, if we run finite-sample EM for at most $T = \mathcal{O}\left( \left[ \frac{n}{d \vee \log \frac{1}{\delta}} \right]^{\frac{1}{2}} \right)$ iterations, then $\alpha^T = \|\theta^T\|/\sigma = \mathcal{O}\left( \left[ \frac{d \vee \log \frac{1}{\delta}}{n} \right]^{\frac{1}{4}} \right)$ with probability at least $1 - T(\delta + \delta')$.

$\square$

# H Extension to Finite Low SNR Regime

## H.1 Series Expansions for Expectations

In this section, we provide series expansions for several expectations involving the hyperbolic tangent function. These expansions are crucial for analyzing the behavior of EM updates in the low SNR regime.

**Lemma H.1** (Derivative Polynomials for tanh). Let $P_n(t) = \frac{\mathrm{d}^n}{\mathrm{d}Z^n} \tanh(Z)$ for any $n \in \mathbb{Z}_{\geq 0}$ and $t := \tanh(Z)$, then

$$P_0(t) = \tanh(Z) = t, \quad P_{n+1}(t) = (1 - t^2)\frac{\mathrm{d}}{\mathrm{d}t}P_n(t),$$

and $P_n(t) \in \mathbb{Z}[t]$ are polynomials in $t$ for any $n \in \mathbb{Z}_{\geq 0}$, and we have:

$$\deg(P_n(t)) = n + 1, \quad \max_{t \in [-1,1]} |P_n(t)| \leq n!.$$

*Proof.* Let $P_n(t) = \frac{\mathrm{d}^n}{\mathrm{d}Z^n} \tanh(Z)$ for any $n \in \mathbb{Z}_{\geq 0}$ and $t := \tanh(Z)$, then by the chain rule:

$$P_0(t) = \tanh(Z) = t \in \mathbb{Z}[t], \quad P_{n+1}(t) = \frac{\mathrm{d}t}{\mathrm{d}Z} \cdot \frac{\mathrm{d}}{\mathrm{d}t}P_n(t) = (1 - t^2)\frac{\mathrm{d}}{\mathrm{d}t}P_n(t).$$

Thus, by the induction, we have $P_n(t) \in \mathbb{Z}[t]$ are polynomials in $t$ for any $n \in \mathbb{Z}_{\geq 0}$, and we have:

$$\deg(P_0(t)) = 1, \quad \deg(P_{n+1}(t)) = 2 + [\deg(P_n(t)) - 1] \implies \deg(P_n(t)) = n + 1.$$

By Bernstein's inequality (see section 7, page 91 of Cheney (1966) and Shadrin (2004)), we have $\max_{t \in [-1,1]} |\sqrt{1 - t^2}[\frac{\mathrm{d}}{\mathrm{d}t}p(t)]| \leq \deg(p(t)) \max_{t \in [-1,1]} |p(t)|$ for any polynomial $p(t)$:

$$\max_{t \in [-1,1]} |P_{n+1}(t)| \leq \max_{t \in [-1,1]} \left| \sqrt{1 - t^2} \left[ \frac{\mathrm{d}}{\mathrm{d}t}P_n(t) \right] \right| \leq \deg(P_n(t)) \max_{t \in [-1,1]} |P_n(t)| = (n + 1) \max_{t \in [-1,1]} |P_n(t)|.$$

Therefore, by the above recurrence relation, we have:

$$\max_{t \in [-1,1]} |P_n(t)| \leq (n!) \max_{t \in [-1,1]} |P_0(t)| = (n!) \max_{t \in [-1,1]} |t| = n!.$$

$\square$

**Lemma H.2** (Reexpressions of Expectations). Let $X \sim f_X(x) = \frac{K_0(|x|)}{\pi}$ be a random variable with probability density involving the Bessel function $K_0$, and $\beta = \tanh(\nu)$. Then for any $n \in \mathbb{Z}_{\geq 0}$: If $0 \leq \alpha < 1/2$, then:

$$\begin{aligned}
\mathbb{E}[\tanh(\alpha X + \nu)X^{2n}] &= \beta\mathbb{E}[X^{2n}] - \beta(1 - \beta^2)\mathbb{E}\left[\tanh^2(\alpha X)X^{2n}\right] \\
&\quad + (1 - \beta^2)\beta^3\mathcal{O}(\alpha^4), \\
\mathbb{E}[\tanh(\alpha X + \nu)X^{2n+1}] &= (1 - \beta^2)\mathbb{E}[\tanh(\alpha X)X^{2n+1}] \\
&\quad + (1 - \beta^2)\beta^2\mathbb{E}[\tanh^3(\alpha X)X^{2n+1}] \\
&\quad + (1 - \beta^2)\beta^4\mathcal{O}(\alpha^5).
\end{aligned}$$

If $0 \leq \alpha < 1/4$, then:

$$\begin{aligned}
\mathbb{E}[\tanh^2(\alpha X + \nu)X^{2n}] &= \beta^2\mathbb{E}[X^{2n}] + (1 - \beta^2)(1 - 3\beta^2)\mathbb{E}\left[\tanh^2(\alpha X)X^{2n}\right] \\
&\quad + (1 - \beta^2)\beta^2\mathcal{O}(\alpha^4), \\
\mathbb{E}[\tanh^2(\alpha X + \nu)X^{2n+1}] &= 2(1 - \beta^2)\beta\mathbb{E}[\tanh(\alpha X)X^{2n+1}] \\
&\quad + 2(1 - \beta^2)(-1 + 2\beta^2)\beta\mathbb{E}[\tanh^3(\alpha X)X^{2n+1}] \\
&\quad + (1 - \beta^2)\beta^3\mathcal{O}(\alpha^5).
\end{aligned}$$

*Proof.* We have the following facts for $\tanh(\alpha X + \nu)$ and $\beta = \tanh(\nu)$:

$$
\begin{aligned}
\tanh(\alpha X + \nu) + \tanh(-\alpha X + \nu) &= 2\beta - 2\beta(1-\beta^2) \cdot \frac{\tanh^2(\alpha X)}{1 - \beta^2 \tanh^2(\alpha X)} \\
&= 2\beta - 2\beta(1-\beta^2) \tanh^2(\alpha X) \\
&\quad - 2\beta(1-\beta^2) \cdot \beta^2 \frac{\tanh^4(\alpha X)}{1 - \beta^2 \tanh^2(\alpha X)}, \\
\tanh(\alpha X + \nu) - \tanh(-\alpha X + \nu) &= 2(1-\beta^2) \cdot \frac{\tanh(\alpha X)}{1 - \beta^2 \tanh^2(\alpha X)} \\
&= 2(1-\beta^2) \cdot \tanh(\alpha X) \\
&\quad + 2(1-\beta^2) \cdot \beta^2 \tanh^3(\alpha X) \\
&\quad + 2(1-\beta^2) \cdot \beta^4 \cdot \frac{\tanh^5(\alpha X)}{1 - \beta^2 \tanh^2(\alpha X)}, \\
\tanh^2(\alpha X + \nu) + \tanh^2(-\alpha X + \nu) &= 2\beta^2 + 2(1-\beta^2) \cdot (1 - 3\beta^2) \tanh^2(\alpha X) \\
&\quad + 2(1-\beta^2) \cdot \beta^2 [(1 - 3\beta^2)(2 - \beta^2 \tanh^2(\alpha X)) \\
&\quad + (1 + \beta^2)] \frac{\tanh^4(\alpha X)}{[1 - \beta^2 \tanh^2(\alpha X)]^2}, \\
\tanh^2(\alpha X + \nu) - \tanh^2(-\alpha X + \nu) &= 4\beta(1-\beta^2) \cdot \frac{\tanh(\alpha X)(1 - \tanh^2(\alpha X))}{[1 - \beta^2 \tanh^2(\alpha X)]^2} \\
&= 4\beta(1-\beta^2) \cdot \tanh(\alpha X) \\
&\quad + 4\beta(1-\beta^2) \cdot (-1 + 2\beta^2) \tanh^3(\alpha X) \\
&\quad + 4\beta(1-\beta^2) \cdot \beta^2 [(-1 + 2\beta^2)(2 - \beta^2 \tanh^2(\alpha X)) \\
&\quad - \beta^2] \frac{\tanh^5(\alpha X)}{[1 - \beta^2 \tanh^2(\alpha X)]^2}.
\end{aligned}
$$

Since $X \sim f_X(x) = \frac{K_0(|x|)}{\pi}$ is a symmetric random variable, we have:

$$
\mathbb{E}[f(X)] = \mathbb{E}[f(-X)] = \mathbb{E}[[f(X) + f(-X)]\mathbb{1}_{X \geq 0}] = \frac{1}{2}\mathbb{E}[[f(X) + f(-X)]].
$$

By substituting $f(X)$ with $\tanh(\alpha X + \nu)X^{2n}, \tanh(\alpha X + \nu)X^{2n+1}, \tanh^2(\alpha X + \nu)X^{2n}, \tanh^2(\alpha X + \nu)X^{2n+1}, n \in \mathbb{Z}_{\geq 0}$, we only have to consider the upper bounds for the absolute values of the remainder terms in the above equalities.

For the remainder terms of $\mathbb{E}[\tanh(\alpha X + \nu)X^{2n}]$ and $\mathbb{E}[\tanh(\alpha X + \nu)X^{2n+1}]$, suppose that for a fixed $\alpha_0$ and $0 \leq \alpha \leq \alpha_0 < 1/2$, we have:

$$
\mathbb{E}\left[\frac{\tanh^4(\alpha X)}{1 - \beta^2 \tanh^2(\alpha X)}X^{2n}\right] \leq \alpha^2 \mathbb{E}[\sinh^2(\alpha X)X^{2n+2}] \leq \alpha^4(\mathbb{E}[\sinh^2(\alpha_0 X)X^{2n+2}]/\alpha_0^2) = \mathcal{O}(\alpha^4),
$$

$$
\mathbb{E}\left[\frac{\tanh^5(\alpha X)}{1 - \beta^2 \tanh^2(\alpha X)}X^{2n+1}\right] \leq \alpha^3 \mathbb{E}[\sinh^2(\alpha X)X^{2n+4}] \leq \alpha^5(\mathbb{E}[\sinh^2(\alpha_0 X)X^{2n+4}]/\alpha_0^2) = \mathcal{O}(\alpha^5).
$$

Note that the coefficients of the remainder for $\mathbb{E}[\tanh^2(\alpha X + \nu)X^{2n}]$ and $\mathbb{E}[\tanh^2(\alpha X + \nu)X^{2n+1}]$ are bounded by:

$$
\begin{aligned}
|(1 - 3\beta^2)(2 - \beta^2 \tanh^2(\alpha X)) + (1 + \beta^2)| &\leq 3, \\
|(-1 + 2\beta^2)(2 - \beta^2 \tanh^2(\alpha X)) - \beta^2| &\leq 2.
\end{aligned}
$$

Suppose that for a fixed $\alpha_0$ and $0 \leq \alpha \leq \alpha_0 < 1/4$, we have:

$$\mathbb{E}\left[\frac{\tanh^4(\alpha X)}{[1 - \beta^2 \tanh^2(\alpha X)]^2} X^{2n}\right] \leq \mathbb{E}[\sinh^4(\alpha X)X^{2n}] \leq \alpha^4(\mathbb{E}[\sinh^4(\alpha_0 X)X^{2n}]/\alpha_0^4) = \mathcal{O}(\alpha^4),$$

$$\mathbb{E}\left[\frac{\tanh^5(\alpha X)}{[1 - \beta^2 \tanh^2(\alpha X)]^2} X^{2n+1}\right] \leq \alpha\mathbb{E}[\sinh^4(\alpha X)X^{2n+2}] \leq \alpha^5(\mathbb{E}[\sinh^4(\alpha_0 X)X^{2n+2}]/\alpha_0^4) = \mathcal{O}(\alpha^5).$$

These remainder terms are bounded by $\mathcal{O}(\alpha^4)$ and $\mathcal{O}(\alpha^5)$ respectively, we have proved the lemma. $\qquad\square$

**Lemma H.3** (Series Expansions for Expectations of tanh). Let $X \sim f_X(x) = \frac{K_0(|x|)}{\pi}$ be a random variable with probability density involving the Bessel function $K_0$, and $\beta = \tanh(\nu)$. Then for $0 \leq \alpha < 1/2$:

$$\begin{aligned}
\mathbb{E}[\tanh(\alpha X + \nu)X^2] &= \beta - 9\alpha^2\beta(1 - \beta^2) + (1 - \beta^2)\beta\mathcal{O}(\alpha^4), \\
\mathbb{E}[\tanh(\alpha X + \nu)X] &= \alpha(1 - \beta^2) - 3\alpha^3(1 - \beta^2)(1 - 3\beta^2) + (1 - \beta^2)\mathcal{O}(\alpha^5), \\
\mathbb{E}[\tanh(\alpha X + \nu)] &= \beta - \alpha^2\beta(1 - \beta^2) + (1 - \beta^2)\beta\mathcal{O}(\alpha^4).
\end{aligned}$$

*Proof.* We start by defining these expectations w.r.t the distribution with a density $f_X(x) = \frac{K_0(|x|)}{\pi}$:

$$l(\alpha, \nu) = \mathbb{E}[\tanh(\alpha X + \nu)X^2], \quad m(\alpha, \nu) = \mathbb{E}[\tanh(\alpha X + \nu)X], \quad n(\alpha, \nu) = \mathbb{E}[\tanh(\alpha X + \nu)].$$

By Lemma H.1 and chain rule, and let $Z := \alpha X + \nu, t = \tanh(Z) = \tanh(\alpha X + \nu)$, then:

$$\left|\frac{\partial^n}{\partial\alpha^n}[\tanh^2(\alpha X + \nu)]\right| = \left|\left[\frac{\partial^n}{\partial Z^n}\tanh^2(Z)\right]X^n\right| = |P_n(t)| \cdot |X^n| \leq n!\,|X^n|.$$

Since $\mathbb{E}|X|^n < \infty, \forall n \in \mathbb{Z}_{\geq 0}$, we can invoke the dominated convergence theorem to exchange the order of taking limit and taking the expectations (see Theorem 1.5.8, page 24 of Durrett (2019)):

$$\frac{\partial^n l(\alpha, \nu)}{\partial\alpha^n} = \mathbb{E}[P_n(t)X^{n+2}], \quad \frac{\partial^n m(\alpha, \nu)}{\partial\alpha^n} = \mathbb{E}[P_n(t)X^{n+1}], \quad \frac{\partial^n n(\alpha, \nu)}{\partial\alpha^n} = \mathbb{E}[P_n(t)X^n].$$

By using the fact that $P_n(t)\mid_{\alpha=0} = P_n(\beta)$ with $\beta = \tanh(\nu)$, we have:

$$\left[\frac{\partial^n l(\alpha, \nu)}{\partial\alpha^n}\right]_{\alpha=0} = P_n(\beta)\mathbb{E}[X^{n+2}], \quad \left[\frac{\partial^n m(\alpha, \nu)}{\partial\alpha^n}\right]_{\alpha=0} = P_n(\beta)\mathbb{E}[X^{n+1}], \quad \left[\frac{\partial^n n(\alpha, \nu)}{\partial\alpha^n}\right]_{\alpha=0} = P_n(\beta)\mathbb{E}[X^n].$$

Since $m(\alpha, \nu)$ and $n(\alpha, \nu)$ have infinitely many derivatives, we can use the Taylor expansion at $\alpha = 0$ (see 5.15 Theorem, pages 110-111 of Rudin (1976)) to approximate $l(\alpha, \nu), m(\alpha, \nu), n(\alpha, \nu)$:

$$\begin{aligned}
l(\alpha, \nu) &= \sum_{n=0}^{3} P_n(\beta)\mathbb{E}[X^{n+2}]\frac{\alpha^n}{n!} + \mathbb{E}[P_4(t)\mid_{\alpha=\alpha'} X^{4+2}]\frac{\alpha^4}{4!} = \sum_{n=0}^{3} P_n(\beta)\mathbb{E}[X^{n+2}]\frac{\alpha^n}{n!} + \mathcal{O}(\alpha^4), \\
m(\alpha, \nu) &= \sum_{n=0}^{4} P_n(\beta)\mathbb{E}[X^{n+1}]\frac{\alpha^n}{n!} + \mathbb{E}[P_5(t)\mid_{\alpha=\alpha''} X^{5+1}]\frac{\alpha^5}{5!} = \sum_{n=0}^{4} P_n(\beta)\mathbb{E}[X^{n+1}]\frac{\alpha^n}{n!} + \mathcal{O}(\alpha^5), \\
n(\alpha, \nu) &= \sum_{n=0}^{3} P_n(\beta)\mathbb{E}[X^n]\frac{\alpha^n}{n!} + \mathbb{E}[P_4(t)\mid_{\alpha=\alpha'''} X^4]\frac{\alpha^4}{4!} = \sum_{n=0}^{3} P_n(\beta)\mathbb{E}[X^n]\frac{\alpha^n}{n!} + \mathcal{O}(\alpha^4),
\end{aligned}$$

where $\alpha', \alpha'', \alpha'''$ are some values between 0 and $\alpha$, and coefficients of remainders are bounded by:

$$\begin{aligned}
\left|\mathbb{E}[P_4(t)\mid_{\alpha=\alpha'} X^{4+2}]/4!\right| &\leq \left|\mathbb{E}[X^{4+2}]\right| \cdot 4!/4! = \mathbb{E}[X^6] = (5!!)^2 = 225, \\
\left|\mathbb{E}[P_5(t)\mid_{\alpha=\alpha''} X^{5+1}]/5!\right| &\leq \left|\mathbb{E}[X^{5+1}]\right| \cdot 5!/5! = \mathbb{E}[X^6] = (5!!)^2 = 225, \\
\left|\mathbb{E}[P_4(t)\mid_{\alpha=\alpha'''} X^4]/4!\right| &\leq \left|\mathbb{E}[X^4]\right| \cdot 4!/4! = \mathbb{E}[X^4] = (3!!)^2 = 9.
\end{aligned}$$

By the recurrence relation $P_{n+1}(t) = (1 - t^2)\frac{\mathrm{d}}{\mathrm{d}t}P_n(t), P_0(t) = t$ in Lemma H.1, we have

$$P_1(\beta) = 1 - \beta^2, P_2(\beta) = (1 - \beta^2) \cdot (-2\beta), P_3(\beta) = (1 - \beta^2) \cdot 2(-1 + 3\beta^2), P_4(\beta) = (1 - \beta^2) \cdot 8\beta(2 - 3\beta^2).$$

Note that $\mathbb{E}[X^{2k}] = [(2k-1)!!]^2, \mathbb{E}[X^{2k-1}] = 0$ for any $k \in \mathbb{Z}_+$, then:

$$
\begin{aligned}
l(\alpha, \nu) &= P_0(\beta) + \frac{9}{2}P_2(\beta)\alpha^2 + \mathcal{O}(\alpha^4) = \beta - 9\alpha^2\beta(1-\beta^2) + \mathcal{O}(\alpha^4) \\
m(\alpha, \nu) &= P_1(\beta)\alpha + \frac{3}{2}P_3(\beta)\alpha^3 + \mathcal{O}(\alpha^5) = \alpha(1-\beta^2) - 3\alpha^3(1-\beta^2)(1-3\beta^2) + \mathcal{O}(\alpha^5) \\
n(\alpha, \nu) &= P_0(\beta) + \frac{1}{2}P_2(\beta)\alpha^2 + \mathcal{O}(\alpha^4) = \beta - \alpha^2\beta(1-\beta^2) + \mathcal{O}(\alpha^4)
\end{aligned}
$$

By invoking Lemma H.2 for $0 \le \alpha < 1/2$ and the fact that $\mathbb{E}[X^{2k}] = [(2k-1)!!]^2$ for any $k \in \mathbb{Z}_+$

$$
\begin{aligned}
\mathbb{E}[P_4(t)\mid_{\alpha=\alpha'} X^{4+2}]\frac{\alpha^4}{4!} &= -\beta(1-\beta^2)\mathbb{E}[(\tanh^2(\alpha X) - (\alpha X)^2)X^2] + (1-\beta^2)\beta^3\mathcal{O}(\alpha^4), \\
\mathbb{E}[P_5(t)\mid_{\alpha=\alpha''} X^{5+1}]\frac{\alpha^5}{5!} &= (1-\beta^2)\mathbb{E}[(\tanh(\alpha X) - [(\alpha X) - (\alpha X)^3/3])X] \\
&\quad + (1-\beta^2)\beta^2\mathbb{E}[(\tanh^3(\alpha X) - (\alpha X)^3)X] + (1-\beta^2)\beta^4\mathcal{O}(\alpha^5), \\
\mathbb{E}[P_4(t)\mid_{\alpha=\alpha'''} X^4]\frac{\alpha^4}{4!} &= -\beta(1-\beta^2)\mathbb{E}[(\tanh^2(\alpha X) - (\alpha X)^2)] + (1-\beta^2)\beta^3\mathcal{O}(\alpha^4).
\end{aligned}
$$

By $t - \frac{t^3}{3} \le \tanh(t) \le t - \frac{t^3}{3} + \frac{2t^5}{15}, t^2 - \frac{2t^4}{3} \le \tanh^2(t) \le t^2, t^3 - t^5 \le \tanh^3(t) \le t^3, \forall t \ge 0$, we have:

$$
\begin{aligned}
\left| \mathbb{E}[P_4(t)\mid_{\alpha=\alpha'} X^{4+2}]\frac{\alpha^4}{4!} \right| &\le (1-\beta^2)|\beta|\left( \left| \alpha^4 \cdot \frac{2}{3}\mathbb{E}[X^{4+2}] \right| + \beta^2|\mathcal{O}(\alpha^4)| \right) = (1-\beta^2)|\beta|\mathcal{O}(\alpha^4), \\
\left| \mathbb{E}[P_5(t)\mid_{\alpha=\alpha''} X^{5+1}]\frac{\alpha^5}{5!} \right| &\le (1-\beta^2)\left( \left| \alpha^5 \cdot \frac{2}{15}\mathbb{E}[X^{5+1}] \right| + \beta^2\left| \alpha^5\mathbb{E}[X^{5+1}] \right| + \beta^4|\mathcal{O}(\alpha^5)| \right) \\
&= (1-\beta^2)\mathcal{O}(\alpha^5), \\
\left| \mathbb{E}[P_4(t)\mid_{\alpha=\alpha'''} X^4]\frac{\alpha^4}{4!} \right| &\le (1-\beta^2)|\beta|\left( \left| \alpha^4 \cdot \frac{2}{3}\mathbb{E}[X^4] \right| + \beta^2|\mathcal{O}(\alpha^4)| \right) = (1-\beta^2)|\beta|\mathcal{O}(\alpha^4).
\end{aligned}
$$

$\square$

**Lemma H.4** (Series Expansions for Expectations of tanh Squared). Let $X \sim f_X(x) = \frac{K_0(|x|)}{\pi}$ be a random variable with probability density involving the Bessel function $K_0$, and $\beta = \tanh(\nu)$. Then for $0 \le \alpha < 1/4$:

$$
\begin{aligned}
\mathbb{E}[\tanh^2(\alpha X + \nu)X^2] &= \beta^2 + 9\alpha^2(1-\beta^2)(1-3\beta^2) + (1-\beta^2)\mathcal{O}(\alpha^4), \\
\mathbb{E}[\tanh^2(\alpha X + \nu)X] &= 2\alpha\beta(1-\beta^2) - 12\alpha^3\beta(1-\beta^2)(2-3\beta^2) + (1-\beta^2)\beta\mathcal{O}(\alpha^5).
\end{aligned}
$$

*Proof.* We start by defining these expectations w.r.t the distribution with a density $f_X(x) = \frac{K_0(|x|)}{\pi}$:

$$
m(\alpha, \nu) = \mathbb{E}[\tanh(\alpha X + \nu)X], \quad n(\alpha, \nu) = \mathbb{E}[\tanh(\alpha X + \nu)].
$$

As shown in the proof of Lemma H.3, we can invoke the dominated convergence theorem to exchange the order of taking limit and taking the expectations (see Theorem 1.5.8, page 24 of Durrett (2019)):

$$
\frac{\partial^n}{\partial \alpha^n}m(\alpha, \nu) = \mathbb{E}[P_n(t)X^{n+1}], \quad \frac{\partial^n}{\partial \alpha^n}n(\alpha, \nu) = \mathbb{E}[P_n(t)X^n].
$$

When $n = 1$, by $P_1(t) = 1 - t^2$ from Lemma H.1 and $\mathbb{E}[X^2] = 1, \mathbb{E}[X] = 0$, we have:

$$
\begin{aligned}
\mathbb{E}[\tanh^2(\alpha X + \nu)X^2] &= \mathbb{E}[(1 - P_1(t))X^2] = \mathbb{E}[X^2] - \mathbb{E}[P_1(t)X^2] = 1 - \frac{\partial m(\alpha, \nu)}{\partial \alpha}, \\
\mathbb{E}[\tanh^2(\alpha X + \nu)X] &= \mathbb{E}[(1 - P_1(t))X] = \mathbb{E}[X] - \mathbb{E}[P_1(t)X] = -\frac{\partial n(\alpha, \nu)}{\partial \alpha}.
\end{aligned}
$$

Note that $\mathbb{E}[X^{2k}] = [(2k-1)!!]^2, \mathbb{E}[X^{2k-1}] = 0$ for any $k \in \mathbb{Z}_+$, and since $m(\alpha, \nu)$ and $n(\alpha, \nu)$ have infinitely many derivatives, we can use the Taylor expansion at $\alpha = 0$ (see 5.15 Theorem, pages 110-111 of Rudin (1976)) to approximate $\frac{\partial}{\partial \alpha} m(\alpha, \nu)$ and $\frac{\partial}{\partial \alpha} n(\alpha, \nu)$:

$$\frac{\partial m(\alpha, \nu)}{\partial \alpha} = \sum_{n=1}^{4} P_n(\beta) \frac{\mathbb{E}[X^{n+1}]}{(n-1)!} \cdot \alpha^{n-1} + \mathbb{E}[P_5(t) \mid_{\alpha=\alpha'} X^{5+1}] \frac{\alpha^4}{4!} = P_1(\beta) + \frac{9}{2} P_3(\beta)\alpha^2 + \mathcal{O}(\alpha^4),$$

$$\frac{\partial n(\alpha, \nu)}{\partial \alpha} = \sum_{n=1}^{5} P_n(\beta) \frac{\mathbb{E}[X^n]}{(n-1)!} \cdot \alpha^{n-1} + \mathbb{E}[P_6(t) \mid_{\alpha=\alpha''} X^6] \frac{\alpha^5}{5!} = P_2(\beta)\alpha + \frac{3}{2} P_4(\beta)\alpha^3 + \mathcal{O}(\alpha^5).$$

where $\alpha', \alpha''$ are some values between 0 and $\alpha$, and the coefficients of remainders are bounded by:

$$\left| \mathbb{E}[P_5(t) \mid_{\alpha=\alpha'} \cdot X^{5+1}]/4! \right| \leq \mathbb{E}[X^6] \cdot 5!/4! = 5 \cdot (5!!)^2,$$

$$\left| \mathbb{E}[P_6(t) \mid_{\alpha=\alpha''} \cdot X^6]/5! \right| \leq \mathbb{E}[X^6] \cdot 6!/5! = 6 \cdot (5!!)^2.$$

By the recurrence relation $P_{n+1}(t) = (1 - t^2) \frac{\mathrm{d}}{\mathrm{d}t} P_n(t)$ with $P_0(t) = t$ from Lemma H.1, we have

$$P_1(\beta) = 1 - \beta^2, \ P_2(\beta) = (1 - \beta^2) \cdot (-2\beta), \ P_3(\beta) = (1 - \beta^2) \cdot 2(-1 + 3\beta^2), \ P_4(\beta) = (1 - \beta^2) \cdot 8\beta(2 - 3\beta^2).$$

Hence, by substituting the values of $P_n(\beta)$, we have:

$$\mathbb{E}[\tanh^2(\alpha X + \nu)X^2] = \beta^2 + 9\alpha^2(1 - \beta^2)(1 - 3\beta^2) + \mathcal{O}(\alpha^4)$$

$$\mathbb{E}[\tanh^2(\alpha X + \nu)X] = 2\alpha\beta(1 - \beta^2) - 12\alpha^3\beta(1 - \beta^2)(2 - 3\beta^2) + \mathcal{O}(\alpha^5).$$

By invoking Lemma H.2 for $0 \leq \alpha < 1/4$ and the fact that $\mathbb{E}[X^{2k}] = [(2k-1)!!]^2$ for any $k \in \mathbb{Z}_+$

$$-\mathbb{E}[P_5(t) \mid_{\alpha=\alpha'} X^{5+1}] \frac{\alpha^4}{4!} = (1 - \beta^2)(1 - 3\beta^2)\mathbb{E}[(\tanh^2(\alpha X) - (\alpha X)^2)X^2] + (1 - \beta^2)\beta^2 \mathcal{O}(\alpha^4),$$

$$-\mathbb{E}[P_6(t) \mid_{\alpha=\alpha''} X^6] \frac{\alpha^5}{5!} = 2(1 - \beta^2)\beta \mathbb{E}[(\tanh(\alpha X) - ((\alpha X) - (\alpha X)^3/3))X]$$

$$+ 2(1 - \beta^2)(-1 + 2\beta^2)\beta \mathbb{E}[(\tanh^3(\alpha X) - (\alpha X)^3)X] + (1 - \beta^2)\beta^2 \mathcal{O}(\alpha^5).$$

By $t - \frac{t^3}{3} \leq \tanh(t) \leq t - \frac{t^3}{3} + \frac{2t^5}{15}, t^2 - \frac{2t^4}{3} \leq \tanh^2(t) \leq t^2, t^3 - t^5 \leq \tanh^3(t) \leq t^3, \forall t \geq 0$, we have:

$$\left| \mathbb{E}[P_5(t) \mid_{\alpha=\alpha'} X^{5+1}] \frac{\alpha^4}{4!} \right| \leq (1 - \beta^2) \left( \left| \alpha^4 \cdot 2 \cdot \frac{2}{3} \mathbb{E}[X^{4+2}] \right| + \beta^2 |\mathcal{O}(\alpha^4)| \right) = (1 - \beta^2)\mathcal{O}(\alpha^4),$$

$$\left| \mathbb{E}[P_6(t) \mid_{\alpha=\alpha''} X^6] \frac{\alpha^5}{5!} \right| \leq (1 - \beta^2)|\beta| \left( \left| \alpha^5 \cdot 2 \cdot \frac{2}{15} \mathbb{E}[X^{5+1}] \right| + |\alpha^5 \cdot 2 \cdot \mathbb{E}[X^{5+1}]| + \beta^2 |\mathcal{O}(\alpha^5)| \right)$$

$$= (1 - \beta^2)|\beta|\mathcal{O}(\alpha^5).$$

$\square$

**Lemma H.5** (Expectations w.r.t Gaussian Distribution). Let $X \sim f_X(x) = \frac{K_0(|x|)}{\pi}$ be a random variable with probability density involving the Bessel function $K_0$, and $Z_1, Z_2 \overset{\text{i.i.d.}}{\sim} \mathcal{N}(0, 1)$. Then:

$$\mathbb{E}[\tanh(\alpha X + \nu)X^2] = \mathbb{E}[\tanh(\alpha Z_1 Z_2 + \nu)Z_2^2] + \mathbb{E}[\tanh'(\alpha Z_1 Z_2 + \nu)\alpha Z_1 Z_2^3],$$

$$\mathbb{E}[\tanh(\alpha X + \nu)X] = \mathbb{E}[\tanh(\alpha Z_1 Z_2 + \nu)Z_1 Z_2] = \mathbb{E}[\tanh'(\alpha Z_1 Z_2 + \nu)\alpha Z_2^2],$$

$$\mathbb{E}[\tanh(\alpha X + \nu)] = \mathbb{E}[\tanh(\alpha Z_1 Z_2 + \nu)],$$

$$-\alpha\mathbb{E}[\tanh^2(\alpha X + \nu)X] = \mathbb{E}[\tanh'(\alpha Z_1 Z_2 + \nu)\alpha Z_1 Z_2],$$

where $\tanh'(\cdot)$ is the derivative of $\tanh(\cdot)$.

*Proof.* For $\alpha = 0$, We can justify the above identities by $\mathbb{E}[X^2] = \mathbb{E}[Z_2^2] = 1, \mathbb{E}[X] = \mathbb{E}[Z_1 Z_2] = 0$, therefore, we only need to prove the identities for $\alpha \neq 0$. By applying Stein's lemma (see Lemma 2.1 of Ross (2011))

with respect to $Z_1$, and noting the fact that $X := Z_1 Z_2 \sim f_X(x) = \frac{K_0(|x|)}{\pi}$ for any $Z_1, Z_2 \overset{\text{i.i.d.}}{\sim} \mathcal{N}(0,1)$ (see page 50, section 4.4 of chapter 12 in Johnson et al. (1970)), we have:

$$
\begin{aligned}
\mathbb{E}[\tanh(\alpha Z_1 Z_2 + \nu) Z_2^2] &= \frac{1}{\alpha} \mathbb{E}\left[ \frac{\partial \ln \cosh(\alpha Z_1 Z_2 + \nu) Z_2}{\partial Z_1} \right] = \frac{1}{\alpha} \mathbb{E}[\ln \cosh(\alpha Z_1 Z_2 + \nu) Z_2 \times Z_1], \\
&= \frac{1}{\alpha} \mathbb{E}[\ln \cosh(\alpha X + \nu) X], \\
\mathbb{E}[\tanh'(\alpha Z_1 Z_2 + \nu) \alpha Z_2^2] &= \mathbb{E}\left[ \frac{\partial \tanh(\alpha Z_1 Z_2 + \nu) Z_2}{\partial Z_1} \right] = \mathbb{E}[\tanh(\alpha Z_1 Z_2 + \nu) Z_2 \times Z_1], \\
&= \mathbb{E}[\tanh(\alpha X + \nu) X], \\
\mathbb{E}[\tanh'(\alpha Z_1 Z_2 + \nu) \alpha Z_1 Z_2^3] &= \mathbb{E}\left[ \frac{\partial \tanh(\alpha Z_1 Z_2 + \nu) Z_1 Z_2^2}{\partial Z_1} - \tanh(\alpha Z_1 Z_2 + \nu) Z_2^2 \right], \\
&= \mathbb{E}[\tanh(\alpha X + \nu) X^2] - \frac{1}{\alpha} \mathbb{E}[\ln \cosh(\alpha X + \nu) X].
\end{aligned}
$$

By combining the above equations, the first two identities are proved. For the last two identities, we note that $\mathbb{E}[X] = 0$, $\tanh'(\cdot) = 1 - \tanh^2(\cdot)$, then:

$$
\begin{aligned}
\mathbb{E}[\tanh'(\alpha Z_1 Z_2 + \nu) \alpha Z_1 Z_2] &= -\alpha \mathbb{E}[\tanh^2(\alpha X + \nu) X], \\
\mathbb{E}[\tanh(\alpha Z_1 Z_2 + \nu)] &= \mathbb{E}[\tanh(\alpha X + \nu)].
\end{aligned}
$$

$\square$

## H.2 Approximations of EM Update Rules in Finite Low SNR Regime

**Theorem H.6** (Approximations of EM Update Rules in Finite Low SNR Regime). *In the finite low SNR regime where $\eta = \|\theta^*\|/\sigma \lesssim 1$, the EM update rules are:*

$$
\begin{aligned}
M(\theta, \nu)/\sigma &= \vec{e}_1 \left[ \mathbb{E}[\tanh(\alpha X + \nu)X] + \eta \beta^* \rho \mathbb{E}[\tanh(\alpha X + \nu)X^2] + \mathcal{O}(\eta^2 \alpha) \right] \\
&+ \vec{e}_2 \left[ \eta \beta^* \sqrt{1 - \rho^2} \left( \mathbb{E}[\tanh(\alpha X + \nu)] - \alpha \mathbb{E}[\tanh^2(\alpha X + \nu)X] \right) + \mathcal{O}(\eta^2 \alpha) \right], \\
N(\theta, \nu) &= \mathbb{E}[\tanh(\alpha X + \nu)] + \eta \beta^* \rho \mathbb{E}[\tanh(\alpha X + \nu)X] + \mathcal{O}(\eta^2 \alpha^2),
\end{aligned}
$$

*where $X \sim f_X(x) = \frac{K_0(|x|)}{\pi}$ is a random variable with probability density involving the Bessel function $K_0$, $\beta^* = \tanh \nu^* = \pi^*(1) - \pi^*(2)$, $\rho = \frac{\langle \theta, \theta^* \rangle}{\|\theta\| \cdot \|\theta^*\|}$ is the cosine of angle between $\theta$ and $\theta^*$, $\vec{e}_1 = \theta/\|\theta\|$ and $\vec{e}_2 = \frac{\theta^* - \langle \theta^*, \vec{e}_1 \rangle \vec{e}_1}{\|\theta^* - \langle \theta^*, \vec{e}_1 \rangle \vec{e}_1\|}$ form an orthonormal basis, and $\alpha = \|\theta\|/\sigma, \beta = \tanh \nu = \pi(1) - \pi(2)$.*

*Proof.* We start by viewing the response variable $y = \varepsilon + \Delta\varepsilon$ as pure noise $\varepsilon \sim \mathcal{N}(0, \sigma^2)$ with some small perturbation $\Delta\varepsilon := (-1)^{z+1}\langle \theta^*, x \rangle$, where $x \sim \mathcal{N}(0, I_d)$. To simplify our analysis, we introduce several notations. First, let $\rho := \frac{\langle \theta, \theta^* \rangle}{\|\theta\| \cdot \|\theta^*\|}$ denote the cosine of angle between $\theta$ and $\theta^*$. Next, we define a pair of orthonormal vectors: $\vec{e}_1 := \theta/\|\theta\|$ and $\vec{e}_2 := \frac{\theta^* - \langle \theta^*, \vec{e}_1 \rangle \vec{e}_1}{\|\theta^* - \langle \theta^*, \vec{e}_1 \rangle \vec{e}_1\|}$. For the noise terms, we introduce three independent standard Gaussians: $Z_1 := \varepsilon/\sigma \sim \mathcal{N}(0,1)$ and $Z_2 := \langle x, \vec{e}_1 \rangle = \langle x, \theta \rangle/\|\theta\| \sim \mathcal{N}(0,1), Z_3 := \langle x, \vec{e}_2 \rangle \sim \mathcal{N}(0,1)$. Finally, for the low SNR regime where $\eta := \|\theta^*\|/\sigma \lesssim 1$, we define $\alpha := \|\theta\|/\sigma$ and $\beta := \tanh \nu = \pi(1) - \pi(2)$. Then, we can express $\Delta Z_1 := \Delta\varepsilon/\sigma$ as:

$$
\Delta Z_1 = \Delta\varepsilon/\sigma = (-1)^{z+1}\langle \theta^*, x \rangle/\sigma = (-1)^{z+1}\eta(\rho Z_2 + \sqrt{1 - \rho^2} Z_3) \sim \mathcal{N}(0, \eta^2)
$$

Then, we can define $F, G$ as the following functions:

$$
F(Z) := \tanh(\alpha Z_2 Z + \nu)Z, \quad G(Z) := \tanh(\alpha Z_2 Z + \nu).
$$

Consequently, we can express the EM update rules $M(\theta, \nu), N(\theta, \nu)$ at population level as:

$$
\begin{aligned}
M(\theta, \nu)/\sigma &= \mathbb{E}\left[ \tanh\left( \frac{y\langle x, \theta \rangle}{\sigma^2} + \nu \right) yx \right]/\sigma = \mathbb{E}[F(Z_1 + \Delta Z_1)(\vec{e}_1 Z_2 + \vec{e}_2 Z_3)], \\
N(\theta, \nu) &= \mathbb{E}\left[ \tanh\left( \frac{y\langle x, \theta \rangle}{\sigma^2} + \nu \right) \right] = \mathbb{E}[G(Z_1 + \Delta Z_1)].
\end{aligned}
$$

By introducing the derivative of tanh in Lemma H.1 and invoking Taylor's theorem with remainder (see 5.15 Theorem, pages 110-111 of Rudin (1976)), we have:

$$
\begin{aligned}
F(Z_1 + \Delta Z_1) &= F(Z_1) + F'(Z_1)\Delta Z_1 + \frac{1}{2}F''(Z_1 + \xi \Delta Z_1)(\Delta Z_1)^2, \\
G(Z_1 + \Delta Z_1) &= G(Z_1) + G'(Z_1)\Delta Z_1 + \frac{1}{2}G''(Z_1 + \zeta \Delta Z_1)(\Delta Z_1)^2,
\end{aligned}
$$

where $\xi, \zeta \in (0, 1)$, remainders are bounded by (use $\max_{t \in [-1,1]} |P_2(t)| \le 2, \max_{t \in [-1,1]} |P_1(t)| \le 1$):

$$
\begin{aligned}
|F''(Z_1 + \xi \Delta Z_1)| &= |(\alpha Z_2)^2 (Z_1 + \xi \Delta Z_1) P_2(G(Z_1 + \xi \Delta Z_1)) + 2(\alpha Z_2) P_1(G(Z_1 + \xi \Delta Z_1))| \\
&\le 2\alpha^2 Z_2^2(|Z_1| + |\Delta Z_1|) + 2\alpha |Z_2|, \\
|G''(Z_1 + \zeta \Delta Z_1)| &= |(\alpha Z_2)^2 P_2(G(Z_1 + \zeta \Delta Z_1))| \le 2\alpha^2 Z_2^2.
\end{aligned}
$$

and by the orthogonality of $\vec{e}_1, \vec{e}_2$ and $|\Delta Z_1|^3 \leq 4\eta^3(|Z_2|^3 + |Z_3|^3), (\Delta Z_1)^2 \leq 2\eta^2(|Z_2|^2 + |Z_3|^2)$:

$$\left| \mathbb{E}\left\langle \frac{1}{2}F''(Z_1 + \xi\Delta Z_1)(\Delta Z_1)^2(\vec{e}_1 Z_2 + \vec{e}_2 Z_3), \vec{e}_1 \right\rangle \right| \leq \frac{1}{2}\mathbb{E}\left| F''(Z_1 + \xi\Delta Z_1)(\Delta Z_1)^2 Z_2 \right|$$

$$\leq \quad \mathbb{E}[[\alpha^2 Z_2^2(|Z_1| + |\Delta Z_1|) + \alpha|Z_2|](\Delta Z_1)^2|Z_2|]$$

$$\leq \quad 2\eta^2\mathbb{E}[(\alpha^2|Z_1||Z_2|^3 + \alpha|Z_2|^2)(|Z_2|^2 + |Z_3|^2)] + 4\eta^3\alpha^2\mathbb{E}[|Z_2|^3(|Z_2|^3 + |Z_3|^3)]$$

$$= \quad 8\eta^2\alpha(1 + \frac{5}{\pi}\alpha) + 4 \cdot (15 + \frac{8}{\pi})\eta^3\alpha^2 = \mathcal{O}(\eta^2\alpha),$$

$$\left| \mathbb{E}\left\langle \frac{1}{2}F''(Z_1 + \zeta\Delta Z_1)(\Delta Z_1)^2(\vec{e}_1 Z_2 + \vec{e}_2 Z_3), \vec{e}_2 \right\rangle \right| \leq \frac{1}{2}\mathbb{E}\left| F''(Z_1 + \zeta\Delta Z_1)(\Delta Z_1)^2 Z_3 \right|$$

$$\leq \quad \mathbb{E}[[\alpha^2 Z_2^2(|Z_1| + |\Delta Z_1|) + \alpha|Z_2|](\Delta Z_1)^2|Z_3|]$$

$$\leq \quad 2\eta^2\mathbb{E}[(\alpha^2|Z_1||Z_2|^2|Z_3| + \alpha|Z_2||Z_3|)(|Z_2|^2 + |Z_3|^2)] + 4\eta^3\alpha^2\mathbb{E}[|Z_2|^2|Z_3|(|Z_2|^3 + |Z_3|^3)]$$

$$= \quad \frac{16}{\pi}\eta^2\alpha(1 + \frac{5}{4}\alpha) + 4 \cdot (3 + \frac{16}{\pi})\eta^3\alpha^2 = \mathcal{O}(\eta^2\alpha),$$

$$\left| \mathbb{E}\left[ \frac{1}{2}G''(Z_1 + \zeta\Delta Z_1)(\Delta Z_1)^2 \right] \right| \leq \frac{1}{2}\mathbb{E}\left| G''(Z_1 + \zeta\Delta Z_1)(\Delta Z_1)^2 \right| \leq \mathbb{E}\left[ \alpha^2 Z_2^2(\Delta Z_1)^2 \right]$$

$$\leq \quad 2\eta^2\alpha^2\mathbb{E}[|Z_2|^2(|Z_2|^2 + |Z_3|^2)] = 8\eta^2\alpha^2 = \mathcal{O}(\eta^2\alpha^2).$$

Note $\mathbb{E}[Z_3] = 0, \mathbb{E}[Z_3^2] = 1$ and $Z_1, Z_2, Z_3$ are independent and $\mathbb{E}[(-1)^{z+1}] = \beta^* = \tanh\nu^*$, then:

$$M(\theta, \nu)/\sigma = \mathbb{E}[(F(Z_1) + F(Z_1)\Delta Z_1)(\vec{e}_1 Z_2 + \vec{e}_2 Z_3)] + \vec{e}_1\mathcal{O}(\eta^2\alpha) + \vec{e}_2\mathcal{O}(\eta^2\alpha)$$

$$= \vec{e}_1\left[ \mathbb{E}[F(Z_1)Z_2] + \eta\beta^*\rho\mathbb{E}[F'(Z_1)Z_2^2] + \mathcal{O}(\eta^2\alpha) \right]$$

$$+ \vec{e}_2\left[ \eta\beta^*\sqrt{1 - \rho^2}\mathbb{E}[F'(Z_1)] + \mathcal{O}(\eta^2\alpha) \right],$$

$$N(\theta, \nu) = \mathbb{E}[G(Z_1) + G'(Z_1)\Delta Z_1] + \mathcal{O}(\eta^2\alpha)$$

$$= \mathbb{E}[G(Z_1)] + \eta\beta^*\rho\mathbb{E}[G'(Z_1)Z_2] + \mathcal{O}(\eta^2\alpha^2).$$

Let $X := Z_1 Z_2 \sim f_X(x) = \frac{K_0(|x|)}{\pi}$, by invoking Lemma H.5 for expectations w.r.t Gaussians, then:

$$\mathbb{E}[F(Z_1)Z_2] = \mathbb{E}[\tanh(\alpha Z_1 Z_2 + \nu)Z_1 Z_2] = \mathbb{E}[\tanh(\alpha X + \nu)X],$$

$$\mathbb{E}[F'(Z_1)Z_2^2] = \mathbb{E}[\tanh(\alpha Z_1 Z_2 + \nu)Z_2^2] + \mathbb{E}[\tanh'(\alpha Z_1 Z_2 + \nu)\alpha Z_1 Z_2^3] = \mathbb{E}[\tanh(\alpha X + \nu)X^2],$$

$$\mathbb{E}[F'(Z_1)] = \mathbb{E}[\tanh(\alpha Z_1 Z_2 + \nu)] + \mathbb{E}[\tanh'(\alpha Z_1 Z_2 + \nu)\alpha Z_1 Z_2]$$

$$= \mathbb{E}[\tanh(\alpha X + \nu)] - \alpha\mathbb{E}[\tanh^2(\alpha X + \nu)X],$$

$$\mathbb{E}[G(Z_1)] = \mathbb{E}[\tanh(\alpha Z_1 Z_2 + \nu)] = \mathbb{E}[\tanh(\alpha X + \nu)],$$

$$\mathbb{E}[G'(Z_1)Z_2] = \mathbb{E}[\tanh'(\alpha Z_1 Z_2 + \nu)\alpha Z_2^2] = \mathbb{E}[\tanh(\alpha X + \nu)X].$$

Substituting the above results into the EM update rules, we have:

$$M(\theta, \nu)/\sigma = \vec{e}_1\left[ \mathbb{E}[\tanh(\alpha X + \nu)X] + \eta\beta^*\rho\mathbb{E}[\tanh(\alpha X + \nu)X^2] + \mathcal{O}(\eta^2\alpha) \right]$$

$$+ \vec{e}_2\left[ \eta\beta^*\sqrt{1 - \rho^2}\left( \mathbb{E}[\tanh(\alpha X + \nu)] - \alpha\mathbb{E}[\tanh^2(\alpha X + \nu)X] \right) + \mathcal{O}(\eta^2\alpha) \right],$$

$$N(\theta, \nu) = \mathbb{E}[\tanh(\alpha X + \nu)] + \eta\beta^*\rho\mathbb{E}[\tanh(\alpha X + \nu)X] + \mathcal{O}(\eta^2\alpha^2).$$

Therefore, we have established exact expressions of the EM update rules $M(\theta, \nu)$ and $N(\theta, \nu)$ for the regression parameters $\theta$ and the mixing weight imbalance $\tanh\nu = \pi(1) - \pi(2)$ in the low SNR regime where $\eta = \|\theta^*\|/\sigma \lesssim 1$. These expressions involve expectations with respect to the random variable $X \sim f_X(x) = \frac{K_0(|x|)}{\pi}$, where $K_0$ is the modified Bessel function of the second kind. $\qquad\square$

**Remark.** When $|\rho| = 1$, $(\Delta Z_1)_{|\rho|=1} = (-1)^{z+1}\eta Z_2$ is independent of $Z_3$, the remainder satisfies:

$$\mathbb{E}\left\langle \frac{1}{2}F''(Z_1 + \xi\Delta Z_1)(\Delta Z_1)^2(\vec{e}_1 Z_2 + \vec{e}_2 Z_3), \vec{e}_2 \right\rangle_{|\rho|=1} = \mathbb{E}\left[ \frac{1}{2}F''(Z_1 + \Delta Z_1)(\Delta Z_1)^2 \right]_{|\rho|=1} \underbrace{\mathbb{E}[Z_3]}_{=0} = 0.$$

## H.3 Approximate Dynamic Equations in Finite Low SNR Regime

**Theorem H.7.** For the EM iterations of $\alpha^t := \|\theta^t\|/\sigma, \beta^t := \tanh \nu^t, \rho^t := \langle \theta^t, \theta^* \rangle / \|\theta^t\| \|\theta^*\|$, given $\alpha^t < 1/4$ and $\eta = \|\theta^*\|/\sigma \lesssim 1$ in low SNR regime, we have the approximate equations for $\alpha^{t+1}, \beta^{t+1}$:

$$\alpha^{t+1} = \mathbb{E}[\tanh(\alpha^t X + \nu^t)X] + \eta\beta^*\rho^t\mathbb{E}[\tanh(\alpha^t X + \nu^t)X^2] + \eta^2\mathcal{O}\left(\frac{[\beta^*]^2[\beta^t]^2/(1-[\beta^t]^2) \vee [\alpha^t]^2}{\alpha^t}\right),$$

$$\beta^{t+1} = \mathbb{E}[\tanh(\alpha^t X + \nu^t)] + \eta\beta^*\rho^t\mathbb{E}[\tanh(\alpha^t X + \nu^t)X] + \eta^2\mathcal{O}([\alpha^t]^2).$$

Furthermore, with the notations $C_\eta = \alpha^t\frac{(1-[\beta^t]^2)}{|\beta^t \cdot \beta^*|}, C'_\eta = \sqrt{1-[\beta^t]^2}$, when $\eta \lesssim C_\eta \wedge C'_\eta$, we have:

$$\rho^{t+1} = \rho^t + (1-[\rho^t]^2) \cdot \eta\beta^*(\mathbb{E}[\tanh(\alpha^t X + \nu^t)] - \alpha^t\mathbb{E}[\tanh^2(\alpha^t X + \nu^t)X])/\mathbb{E}[\tanh(\alpha^t X + \nu^t)X]$$

$$+ \mathcal{O}\left(\left(\frac{\eta}{C'_\eta}\right)^2\right) + \rho^t\mathcal{O}\left(\left(\frac{\eta}{C_\eta}\right)^2\right) + (1-[\rho^t]^2) \cdot \mathcal{O}\left(\left(\frac{\eta}{C_\eta}\right)^3\right),$$

where $X$ is a random variable with the density function $X \sim f_X(x) = K_0(|x|)/\pi$ involving the Bessel function $K_0$.

*Proof.* To simplify the notations, we let $\alpha^t, \beta^t, \rho^t, \vec{e}_1^t, \vec{e}_2^t$ denote $\alpha, \beta, \rho, \vec{e}_1, \vec{e}_2$ at iteration $t$. We start by using Lemma H.3 and Lemma H.4, we define $s^t := [\alpha^t]^2(1-[\beta^t]^2)$, then:

$$\begin{aligned}
I_0^t &:= \mathbb{E}[\tanh(\alpha^t X + \nu^t)] = \beta^t(1 - s^t[1 + \mathcal{O}([\alpha^t]^2)]), \\
I_1^t &:= \mathbb{E}[\tanh(\alpha^t X + \nu^t)X] = (s^t/\alpha^t)[1 - 3[\alpha^t]^2(1 - 3[\beta^t]^2) + \mathcal{O}([\alpha^t]^4)], \\
I_2^t &:= \mathbb{E}[\tanh(\alpha^t X + \nu^t)X^2] = \beta^t(1 - 9s^t[1 + \mathcal{O}([\alpha^t]^2)]), \\
J^t &:= \mathbb{E}[\tanh(\alpha^t X + \nu^t)] - \alpha^t\mathbb{E}[\tanh^2(\alpha^t X + \nu^t)X] = \beta^t(1 - 3s^t[1 + \mathcal{O}([\alpha^t]^2)]).
\end{aligned}$$

In the low SNR regime where $\eta = \|\theta^*\|/\sigma \lesssim 1$, by invoking the proven Theorem in Subsection H.2, we have:

$$\begin{aligned}
M(\theta^t, \nu^t)/\sigma &= \vec{e}_1^t\left[I_1^t + \eta\beta^*\rho^t I_2^t + \mathcal{O}(\eta^2\alpha^t)\right] \\
&+ \vec{e}_2^t\left[\eta\beta^*\sqrt{1-[\rho^t]^2}J^t + \mathcal{O}(\eta^2\alpha^t)\right], \\
N(\theta^t, \nu^t) &= I_0^t + \eta\beta^*\rho^t I_1^t + \mathcal{O}(\eta^2[\alpha^t]^2).
\end{aligned}$$

For $\alpha^{t+1} = \|M(\theta^t, \nu^t)\|/\sigma, \beta^{t+1} = N(\theta^t, \nu^t)$, by applying the triangle inequality, we have:

$$\begin{aligned}
\alpha^{t+1} &= \sqrt{(I_1^t + \eta\beta^*\rho^t I_2^t)^2 + (1-[\rho^t]^2)(\eta\beta^* J^t)^2} + \mathcal{O}(\eta^2\alpha^t), \\
\beta^{t+1} &= I_0^t + \eta\beta^*\rho^t I_1^t + \mathcal{O}(\eta^2[\alpha^t]^2).
\end{aligned}$$

The cosine $\rho^{t+1}$ of angle between $\theta^{t+1}$ and $\theta^*$ is determined by the inner product of $M(\theta^t, \nu^t)/\sigma$ and $\theta^*/\|\theta^*\| = \rho^t\vec{e}_1^t + \sqrt{1-[\rho^t]^2}\vec{e}_2^t$:

$$\rho^{t+1}\alpha^{t+1} = \langle M(\theta^t, \nu^t)/\sigma, \theta^*/\|\theta^*\|\rangle = \rho^t I_1^t + \eta\beta^*[\rho^t]^2 I_2^t + \eta\beta^*(1-[\rho^t]^2)J^t + \mathcal{O}(\eta^2\alpha^t).$$

To derive the approximation for $\alpha^{t+1}$ when $\eta$ is small enough, we first note that following inequality:

$$r + \epsilon\cos\gamma \leq \sqrt{r^2 + \epsilon^2 + 2r\epsilon\cos\gamma} \leq r + \epsilon\cos\gamma + \frac{\epsilon^2}{2r} = r + \epsilon\cos\gamma + \mathcal{O}\left(\frac{\epsilon^2}{r}\right).$$

By letting $r = I_1^t + \eta\beta^*\rho^t(I_2^t - J^t), \epsilon = \eta\beta^* J^t, \cos\gamma = \rho^t$, we have:

$$\alpha^{t+1} = I_1^t + \eta\beta^*\rho^t I_2^t + \eta^2\mathcal{O}\left(\frac{[\beta^*]^2[\beta^t]^2/(1-[\beta^t]^2) \vee [\alpha^t]^2}{\alpha^t}\right).$$

where $I_2^t - J^t = -6\beta^t s^t[1 + \mathcal{O}([\alpha^t]^2)]$, and $r = \mathcal{O}(s^t/\alpha^t) = (1-[\beta^t]^2)\mathcal{O}(\alpha^t)$ when $\eta \leq \mathcal{O}(1) \leq 1/(\alpha^t|\beta^t\rho^t\beta^*|)$, namely $I_1 \gtrsim \eta|\beta^*\rho^t(I_2^t - J^t)|$.

Furthurmore, when $\frac{\eta}{\alpha^t} \lesssim \frac{(1-[\beta^t]^2)}{|\beta^t \cdot \beta^*|}$, we have $I_1^t \gtrsim \eta|\beta^* \rho^t I_2^t| \vee \eta^2 [\beta^*]^2 [\beta^t]^2/(\alpha^t(1-[\beta^t]^2))$; when $\eta \lesssim \sqrt{1-[\beta^t]^2}$, we have $I_1^t \gtrsim \eta^2 \alpha^t$. Therefore, if $\eta \lesssim \min\left(\alpha^t \frac{(1-[\beta^t]^2)}{|\beta^t \cdot \beta^*|}, \sqrt{1-[\beta^t]^2}\right) = \alpha^t \frac{(1-[\beta^t]^2)}{|\beta^t \cdot \beta^*|} \wedge \sqrt{1-[\beta^t]^2}$, we have

$$I_1^t + \eta\beta^* \rho^t I_2^t = \mathcal{O}(I_1^t) = \mathcal{O}(s^t/\alpha^t) = (1-[\beta^t]^2)\mathcal{O}(\alpha^t), \quad \alpha^{t+1} = \mathcal{O}(I_1^t) = \mathcal{O}(s^t/\alpha^t) = (1-[\beta^t]^2)\mathcal{O}(\alpha^t).$$

For the simplicity of notations, we define $C_\eta := \alpha^t \frac{(1-[\beta^t]^2)}{|\beta^t \cdot \beta^*|}, C'_\eta := \sqrt{1-[\beta^t]^2}$, then our assumptions on $\eta$ can be rewritten as $\eta \lesssim C_\eta \wedge C'_\eta$. By using such fact $1/(1+x) = 1 + \mathcal{O}(|x|), \forall x \in [-1/2, 1/2]$ and $I_1^t + \eta\beta^* \rho^t I_2^t = (1-[\beta^t]^2)\mathcal{O}(\alpha^t)$, then:

$$(I_1^t + \eta\beta^* \rho^t I_2^t)/\alpha^{t+1} = 1 + \mathcal{O}\left(\left(\frac{\eta}{C_\eta}\right)^2 \vee \left(\frac{\eta}{C'_\eta}\right)^2\right),$$

Similary, by letting $x = (\alpha^{t+1} - I_1^t)/I_1^t$ and note that $I_1^t = (1-[\beta^t]^2)\mathcal{O}(\alpha^t), I_2^t = \mathcal{O}(\beta^t), J^t = \mathcal{O}(|\beta^t|)$, then:

$$1/\alpha^{t+1} = \frac{1}{I_1^t}\left[1 + \rho^t \mathcal{O}\left(\frac{\eta}{C_\eta}\right) + \mathcal{O}\left(\left(\frac{\eta}{C_\eta}\right)^2 \vee \left(\frac{\eta}{C'_\eta}\right)^2\right)\right],$$

$$\eta\beta^* J^t/\alpha^{t+1} = \eta\beta^* J^t/I_1^t + \rho^t \mathcal{O}\left(\left(\frac{\eta}{C_\eta}\right)^2\right) + \mathcal{O}\left(\left(\frac{\eta}{C_\eta}\right)\left[\left(\frac{\eta}{C_\eta}\right)^2 \vee \left(\frac{\eta}{C'_\eta}\right)^2\right]\right).$$

Note that $\alpha^{t+1} = \mathcal{O}(I_1^t) = \mathcal{O}(s^t/\alpha^t) = (1-[\beta^t]^2)\mathcal{O}(\alpha^t)$, therefore we have:

$$\mathcal{O}(\eta^2 \alpha^t)/\alpha^{t+1} = \mathcal{O}\left(\left(\frac{\eta}{C'_\eta}\right)^2\right).$$

Note the following relation that we obtained earlier:

$$\rho^{t+1} = \rho^t \cdot (I_1^t + \eta\beta^* \rho^t I_2^t)/\alpha^{t+1} + (1-[\rho^t]^2) \cdot (\eta\beta^* J^t/\alpha^{t+1}) + \mathcal{O}(\eta^2 \alpha^t)/\alpha^{t+1},$$

by combining the above results and invoking the assumption $\eta \lesssim C_\eta \wedge C'_\eta$, then we have:

$$\rho^{t+1} = \rho^t + (1-[\rho^t]^2) \cdot \eta\beta^* J^t/I_1^t + \mathcal{O}\left(\left(\frac{\eta}{C'_\eta}\right)^2\right) + \rho^t \mathcal{O}\left(\left(\frac{\eta}{C_\eta}\right)^2\right) + (1-[\rho^t]^2) \cdot \mathcal{O}\left(\left(\frac{\eta}{C_\eta}\right)^3\right).$$

In summary, with the notations $C_\eta = \alpha^t \frac{(1-[\beta^t]^2)}{|\beta^t \cdot \beta^*|}, C'_\eta = \sqrt{1-[\beta^t]^2}$, given $\alpha^t < 1/4, \eta \lesssim 1$, we have:

$$\alpha^{t+1} = I_1^t + \eta\beta^* \rho^t I_2^t + \alpha^t(1-[\beta^t]^2)\mathcal{O}\left(\left(\frac{\eta}{C_\eta}\right)^2 \vee \left(\frac{\eta}{C'_\eta}\right)^2\right),$$

$$\beta^{t+1} = I_0^t + \eta\beta^* \rho^t I_1^t + \mathcal{O}(\eta^2 [\alpha^t]^2),$$

with the assumption $\eta \lesssim C_\eta \wedge C'_\eta$, we also have:

$$\rho^{t+1} = \rho^t + (1-[\rho^t]^2) \cdot \eta\beta^* J^t/I_1^t + \mathcal{O}\left(\left(\frac{\eta}{C'_\eta}\right)^2\right) + \rho^t \mathcal{O}\left(\left(\frac{\eta}{C_\eta}\right)^2\right) + (1-[\rho^t]^2) \cdot \mathcal{O}\left(\left(\frac{\eta}{C_\eta}\right)^3\right).$$

$\square$

**Remark.** When $|\rho^t| = 1$, by noting the remark in Subsection H.2, the remainder term of the EM update rules in the direction of $\vec{e}_2^t$ is exactly zero, namely $[\vec{e}_2^t \mathcal{O}(\eta^2 \alpha^t)]_{|\rho^t|=1} = \vec{0}$, thus we have:

$$\left[M(\theta^t, \nu^t)/\sigma\right]_{|\rho^t|=1} = \vec{e}_1^t \left[I_1^t + \eta\beta^* \rho^t I_2^t + \mathcal{O}(\eta^2 \alpha^t)\right].$$

Therefore, we have $|\rho^{t+1}| = 1$ if $|\rho^t| = 1$ since $\theta^*/\|\theta^*\| = \text{sgn}(\rho^t)\vec{e}_1^t$ when $|\rho^t| = 1$ and

$$|\rho^{t+1}| = |\langle M(\theta^t, \nu^t)/\|M(\theta^t, \nu^t)\|, \theta^*/\|\theta^*\|\rangle| = |\langle \vec{e}_1^t, \text{sgn}(\rho^t)\vec{e}_1^t\rangle| = |\langle \vec{e}_1^t, \vec{e}_1^t\rangle| = 1.$$

**Theorem H.8** (Dynamic Equations of EM Update Rules in Low SNR Regime)**.** For the EM iterations of $\alpha^t := \|\theta^t\|/\sigma, \beta^t := \tanh \nu^t, \rho^t := \langle \theta^t, \theta^* \rangle / \|\theta^t\|\|\theta^*\|$, given $\alpha^t < 1/4$ and $\eta = \|\theta^*\|/\sigma \lesssim 1$ in low SNR regime, we have the approximate dynamic equations for $\alpha^{t+1}, \beta^{t+1}$:

$$
\begin{aligned}
\alpha^{t+1} &= \alpha^t(1 - [\beta^2]) + \eta\beta^*\rho^t\beta^t(1 - 9(1 - [\beta^t]^2)[\alpha^t]^2) \\
&+ (1 - [\beta^2])\mathcal{O}([\alpha^t]^3) + \eta^2\mathcal{O}\left(\frac{[\beta^*]^2[\beta^t]^2/(1 - [\beta^t]^2) \vee [\alpha^t]^2}{\alpha^t}\right), \\
\beta^{t+1} &= \beta^t(1 - (1 - [\beta^2])[\alpha^t]^2) + \eta\beta^*\rho^t\alpha^t(1 - [\beta^t]^2) + (1 - [\beta^2])\mathcal{O}([\alpha^t]^3) + \eta^2\mathcal{O}\left([\alpha^t]^2\right)
\end{aligned}
$$

Furthermore, with the notations $C_\eta = \alpha^t\frac{(1 - [\beta^t]^2)}{|\beta^t \cdot \beta^*|}, C'_\eta = \sqrt{1 - [\beta^t]^2}$, when $\eta \lesssim C_\eta \wedge C'_\eta$, we have:

$$
\begin{aligned}
\rho^{t+1} &= \rho^t + (1 - [\rho^t]^2) \cdot \eta\beta^*\frac{\beta^t(1 - 6[\alpha^t]^2[\beta^t]^2)}{\alpha^t(1 - [\beta^t]^2)} \\
&+ (1 - [\rho^t]^2) \cdot \eta\beta^*\beta^t(1/(1 - [\beta^t]^2) + 1)\mathcal{O}([\alpha^t]^3) \\
&+ \mathcal{O}\left(\left(\frac{\eta}{C'_\eta}\right)^2\right) + \rho^t\mathcal{O}\left(\left(\frac{\eta}{C_\eta}\right)^2\right) + (1 - [\rho^t]^2) \cdot \mathcal{O}\left(\left(\frac{\eta}{C_\eta}\right)^3\right),
\end{aligned}
$$

*Proof.* By using Lemma H.3 and Lemma H.4 and defining $s^t := [\alpha^t]^2(1 - [\beta^t]^2)$, we have:

$$
\begin{aligned}
I_0^t &:= \mathbb{E}[\tanh(\alpha^t X + \nu^t)] = \beta^t(1 - s^t[1 + \mathcal{O}([\alpha^t]^2)]), \\
I_1^t &:= \mathbb{E}[\tanh(\alpha^t X + \nu^t)X] = \alpha^t(1 - [\beta^t]^2)[1 - 3[\alpha^t]^2(1 - 3[\beta^t]^2) + \mathcal{O}([\alpha^t]^4)], \\
I_2^t &:= \mathbb{E}[\tanh(\alpha^t X + \nu^t)X^2] = \beta^t(1 - 9s^t[1 + \mathcal{O}([\alpha^t]^2)]), \\
J^t &:= \mathbb{E}[\tanh(\alpha^t X + \nu^t)] - \alpha^t\mathbb{E}[\tanh^2(\alpha^t X + \nu^t)X] = \beta^t(1 - 3s^t[1 + \mathcal{O}([\alpha^t]^2)]).
\end{aligned}
$$

By substituting the above results into the approximate dynamic equations in proven Theorem H.7, note that $(1 - [\beta^2])\mathcal{O}([\alpha^t]^3) + \eta\beta^*\rho^t\beta^t(1 - [\beta^t]^2)\mathcal{O}([\alpha^t]^4) = (1 - [\beta^t]^2)\mathcal{O}([\alpha^t]^3)$, we have:

$$
\begin{aligned}
\alpha^{t+1} &= \alpha^t(1 - [\beta^2]) + \eta\beta^*\rho^t\beta^t(1 - 9(1 - [\beta^t]^2)[\alpha^t]^2) \\
&+ (1 - [\beta^2])\mathcal{O}([\alpha^t]^3) + \eta^2\mathcal{O}\left(\frac{[\beta^*]^2[\beta^t]^2/(1 - [\beta^t]^2) \vee [\alpha^t]^2}{\alpha^t}\right).
\end{aligned}
$$

Note that $\beta^t(1 - [\beta^t]^2)\mathcal{O}([\alpha^t]^4) + \eta\beta^*\rho^t(1 - [\beta^t]^2)\mathcal{O}([\alpha^t]^3) = (1 - [\beta^t]^2)\mathcal{O}([\alpha^t]^3)$, we have:

$$
\beta^{t+1} = \beta^t(1 - (1 - [\beta^2])[\alpha^t]^2) + \eta\beta^*\rho^t\alpha^t(1 - [\beta^t]^2) + (1 - [\beta^2])\mathcal{O}([\alpha^t]^3) + \eta^2\mathcal{O}\left([\alpha^t]^2\right).
$$

Furthermore, with the notations $C_\eta = \alpha^t\frac{(1 - [\beta^t]^2)}{|\beta^t \cdot \beta^*|}, C'_\eta = \sqrt{1 - [\beta^t]^2}$, when $\eta \lesssim C_\eta \wedge C'_\eta$, we have:

$$
\begin{aligned}
J^t/I_1^t &= \frac{\beta^t/(1 - [\beta^t]^2)}{\alpha^t} \cdot \frac{1}{1 - 3[\alpha^t]^2(1 - 3[\beta^t]^2) + \mathcal{O}([\alpha^t]^4)} - 3\alpha^t\beta^t\frac{1 + \mathcal{O}([\alpha^t]^2)}{1 - 3[\alpha^t]^2(1 - 3[\beta^t]^2) + \mathcal{O}([\alpha^t]^4)} \\
&= \left(\frac{\beta^t/(1 - [\beta^t]^2)}{\alpha^t} + 3\frac{\alpha^t\beta^t}{1 - [\beta^t]^2} - 9\frac{\alpha^t[\beta^t]^3}{1 - [\beta^t]^2} + \frac{\beta^t}{1 - [\beta^t]^2}\mathcal{O}([\alpha^t]^3)\right) - 3\alpha^t\beta^t + \beta^t\mathcal{O}([\alpha^t]^3) \\
&= \frac{\beta^t(1 - 6[\alpha^t]^2[\beta^t]^2)}{\alpha^t(1 - [\beta^t]^2)} + \beta^t(1/(1 - [\beta^t]^2) + 1)\mathcal{O}([\alpha^t]^3),
\end{aligned}
$$

hence,

$$
\begin{aligned}
\rho^{t+1} &= \rho^t + (1 - [\rho^t]^2) \cdot \eta\beta^*\frac{\beta^t(1 - 6[\alpha^t]^2[\beta^t]^2)}{\alpha^t(1 - [\beta^t]^2)} \\
&+ (1 - [\rho^t]^2) \cdot \eta\beta^*\beta^t(1/(1 - [\beta^t]^2) + 1)\mathcal{O}([\alpha^t]^3) \\
&+ \mathcal{O}\left(\left(\frac{\eta}{C'_\eta}\right)^2\right) + \rho^t\mathcal{O}\left(\left(\frac{\eta}{C_\eta}\right)^2\right) + (1 - [\rho^t]^2) \cdot \mathcal{O}\left(\left(\frac{\eta}{C_\eta}\right)^3\right).
\end{aligned}
$$

$\square$

