# OpenReview forum: "Characterizing Evolution in Expectation-Maximization Estimates for  Overspecified Mixed Linear Regression"
_TMLR — Accepted by TMLR_

### Review · Reviewer_8cnp · 2025-09-30

**Summary Of Contributions:**

This paper offers theoretical analysis of the Expectation-Maximization (EM) algorithm's performance in the context of overspecified two-component Mixed Linear Regression. The authors explore the complex situation where the model includes more components than the actual data-generating distribution. A significant contribution is the examination of how the initial estimate of the mixing weights critically influences the convergence rate of the regression parameters. The theoretical findings are corroborated by numerical experiments and linked to practical applications such as haplotype assembly and phase retrieval.

Strengths:

1. The paper delivers an overview of the evolution of EM for both regression parameters and mixing weights in the overspecified context.

2. It establishes precise bounds for convergence rates and final statistical error, enhancing previous results for 2GMM.

3. The analysis extends beyond the overspecified context to include the finite low SNR regime.

Weaknesses:

1. The theoretical guarantees are specifically derived for two-component mixtures with Gaussian covariates and noise.

2. The paper lacks practical strategies for achieving a favorable unbalanced initialization in real-world situations where the true parameters are not known.

**Audience:**

Yes

**Audience Explanation:**

The paper's rigorous characterization of EM's convergence dichotomy in a fundamental model like mixed linear regression, along with its improved theoretical bounds, will be of great interest to TMLR's audience.

**Claims And Evidence:**

Yes

**Claims Explanation:**

The paper's claims are supported by a novel theoretical framework that incorporates exact EM derivations using Bessel functions, along with numerical experiments that validate the theoretical findings.

**Requested Changes:**

1. In practice, how would you suggest initializing $\pi^0$? Would a strategy that intentionally unbalances the initial guess be advantageous, even if the true weights are balanced?

2. Including simulations related to haplotype assembly or phase retrieval, along with an explanation of how your results contribute, would be beneficial.

---

> ### Author Response · Authors · 2025-11-21
> **Response to Reviewer 8cnp (part 1)**
>
> **Initialization Strategy for $\pi^0$**
>
> We thank the reviewer for this important practical question. We will add a subsection on "Initialization Strategies" in our revised manuscript.
>
> Mathematically, the benefit of unbalancing stems from the local curvature of the likelihood function. In the overspecified regime ($\theta^* = \vec{0}$), the specific value of the mixing weights is structurally irrelevant to the global optimum (since $\theta_1 = \theta_2 = \vec{0}$ satisfies the likelihood regardless of $\pi$). However, initializing with balanced weights ($\pi^0 = (0.5, 0.5)$) places the iterate on a saddle-like region where the first-order gradient signal vanishes (governed by third-order terms, see Eq. 12 in the paper). By intentionally selecting an unbalanced prior (e.g., $\pi^0 = (0.7, 0.3)$), one forces the algorithm into the linear convergence regime defined by the contraction map $\alpha^{t+1} \approx \alpha^t (1 - [\beta^t]^2)$, bypassing the sublinear plateau.
>
> Building on this, our theoretical findings suggest a specific, non-saddle-point-inducing range for the initialization. We recommend an unbalanced initialization of $\pi^0$ satisfying: $\left(\frac{d}{n}\right)^{1 / 4} \lesssim\left\|\pi^0-\frac{1}{2}\right\|_1<1$. This range is derived from the dual requirements of escaping the slow sublinear convergence regime and avoiding singularities. According to Theorem 6.1, if the initialization is sufficiently balanced $\left(\left\|\pi^0-\frac{1}{2}\right\|_1 \lesssim(d / n)^{1 / 4}\right)$ while the SNR is extremely low, the EM algorithm indeed suffers from slow sublinear convergence. Conversely, we must also avoid fully unbalanced weights $\left(\left\|\pi^0-\frac{1}{2}\right\|_1=1\right)$, as this leads to $\nu^0= \pm \infty$ and a vanishing EM update rule for regression parameters $M(\theta, \nu^0)=\mathbb{E}[\tanh(\pm\infty)x y]= \pm\mathbb{E}[x y] = \vec{0}$ when the SNR is not low and the true mixing weights are balanced.
>
> **Non-standard Gaussian and General Settings**
>
> We thank the reviewer for highlighting this limitation. While our analysis focuses on the standard Gaussian setting common in the literature (Balakrishnan et al., 2017; Klusowski et al., 2019; Reisizadeh et al., 2024; Luo and Hashemi, 2024), we will add a discussion on non-standard Gaussian and non-Gaussian extensions to the revised manuscript.
>
> In particular, suppose the covariate $x\sim \mathcal{N}(\vec{0}, C)$ follows an elliptical distribution with the SVD decomposition of covariance matrix $C=Q \Lambda Q^\top$, where $\Lambda=\text{diag}(\lambda_1, \lambda_2, \ldots, \lambda_d)$ is the matrix of non-negative eigenvalues of $C$, and $Q$ is the matrix of orthonormal eigenvectors of $C$. By defining $\alpha^t := || \sqrt{\Lambda} Q^\top \theta^t||/\sigma$ for the $t$-th iteration $\theta^t$ of regression parameters, instead of $\alpha^t:= ||\theta^t||/\sigma$ under the assumption of $x\sim \mathcal{N}(0, I_d)$, the **recurrence relations** for $\alpha^t, \beta^t$ in Eq. 11 **still hold**. Crucially, the direction of population EM iterations remains the same as the previous EM iteration (see Fig. 1(a)), and the analysis for non-standard Gaussian cases can therefore proceed in the same way.
>
> Specifically, we note that handling heavy-tailed noise (e.g., Cauchy distribution) often requires modifying the EM update rules (Zilber et al., 2023), while bounded noise has primarily been analyzed under convex formulations rather than EM (Chen et al., 2017).
>
> Establishing convergence guarantees for the standard EM algorithm with unknown mixing weights and regression parameters under general settings remains an important open problem for future research.
>
> Reference:
>
> (Balakrishnan et al., 2017) Balakrishnan, Sivaraman, Martin J. Wainwright, and Bin Yu. "Statistical guarantees for the EM algorithm: From population to sample-based
> analysis." (2017): 77-120.
>
> (Klusowski et al., 2019) Klusowski, Jason M., Dana Yang, and W. D. Brinda. "Estimating the coefficients of a mixture of two linear regressions by expectation maximization." IEEE Transactions on Information Theory 65.6 (2019): 3515-3524.
>
> (Reisizadeh et al., 2024) Reisizadeh, Amirhossein, Khashayar Gatmiry, and Asuman Ozdaglar. "EM for mixture of linear regression with clustered data." International Conference on Artificial Intelligence and Statistics. PMLR, 2024.
>
> (Luo and Hashemi, 2024) Luo, Zhankun, and Abolfazl Hashemi. "Unveiling the Cycloid Trajectory of EM Iterations in Mixed Linear Regression." *International Conference on Machine Learning*. PMLR, 2024.
>
> (Zilber et al., 2023) Zilber, Pini, and Boaz Nadler. "Imbalanced mixed linear regression." Advances in Neural Information Processing Systems 36 (2023): 54540-54553.
>
> (Chen et al., 2017) Chen, Yudong, Xinyang Yi, and Constantine Caramanis. "Convex and nonconvex formulations for mixed regression with two components: Minimax optimal rates." IEEE Transactions on Information Theory 64.3 (2017): 1738-1766.

---

> ### Author Response · Authors · 2025-11-21
> **Response to Reviewer 8cnp (part 2): Applied Examples and Theoretical Contributions**
>
> Theoretical and Empirical Comparison with Phase Retrieval Problem
>
> We thank the reviewers for their constructive feedback. We will explicitly connect our theoretical contributions to real-world applications such as phase retrieval in our revised manuscript. We provide a theoretical and empirical comparison of our convergence rates with existing results in the phase retrieval literature.
>
> **Comparison with Theoretical Bounds for Phase Retrieval**
>
> For phase retrieval problems, several theoretical guarantees for the parameter estimate $\hat{\theta}$ have been established:
>
> 1. **Convex Formulation** (Chen et al., 2015): For $n \gtrsim d$ samples, the relative error bound is $\|\hat{\theta} - \theta^\ast\|/\sigma \lesssim \sqrt{d/n} \log^4 n + \min(\sqrt{d/n}/\eta, \left(d/n\right)^{1/4}) \log^4 n$, where $\eta=\|\theta^\ast\|/\sigma$ represents the signal-to-noise ratio, as shown on page 10 of (Chen et al., 2015).
>
> 2. **Agnostic Estimation** (Neykov et al., 2020): For $n \gtrsim d^2 \log d$ samples, the error bound of the estimate $\hat{\theta}$ satisfies $\|\hat{\theta} - \theta^\ast\|/\|\theta^\ast\| \lesssim \sqrt{(d \log d)/n}$ under the constraint $\|\hat{\theta}\| = \|\theta^\ast\|$, as established on pages 3 and 9 of (Neykov et al., 2020).
>
>
> For the **EM algorithm** when the sample size $n$ is sufficiently greater than the dimension $d$ ($n \gtrsim d$), the relative error bound for the estimated vector $\hat{\theta}$ depends on the initialization of the mixing weight $\pi^0$. With **unbalanced initialization** of mixing weights $\|\pi^0 - \frac{1}{2}\|_1 \gtrsim (d/n)^{1/4}$, the relative error is bounded by $\|\hat{\theta} - \theta^\ast\|/\sigma \lesssim \sqrt{d/n}$. Conversely, with **balanced initialization** of mixing weights $\|\pi^0 - \frac{1}{2}\|_1 \lesssim (d/n)^{1/4}$, the relative error bound is less favorable, given by $\|\hat{\theta} - \theta^\ast\|/\sigma \lesssim (d/n)^{1/4}$.
>
> **Sample Complexity**: Our results require $n \gtrsim d$ samples, matching the sample complexity requirement established in (Chen et al., 2015) and improving upon the $n \gtrsim d^2 \log d$ requirement in (Neykov et al., 2020).
>
> **Error Rates**: With balanced initialization, our rate $(d/n)^{1/4}$ matches the second term in (Chen et al., 2015), demonstrating that the EM algorithm achieves a better rate when SNR $\eta$ is sufficiently small. With unbalanced initialization, our rate $\sqrt{d/n}$ matches the leading term $\sqrt{d/n} \log^4 n$ in (Chen et al., 2015) and improves upon the $\sqrt{d \log d/n}$ rate in (Neykov et al., 2020) by removing the logarithmic factor when SNR $\eta \asymp 1$.
>
> These theoretical comparisons demonstrate that our EM-based approach achieves competitive or improved error rates compared to existing phase retrieval methods, while providing explicit characterization of the initialization-dependent convergence behavior.
>
> **Empirical Results**:
>
> The following table summarizes the recovery errors for three different formulations tested on mixed regression problems. All experiments use the same data generation parameters ($n=1000$, $d=10$) with balanced $\pi^*=(\frac{1}{2}, \frac{1}{2})$. The EM formulation varies the initial mixing weight $\pi^0=(\pi^0(1), \pi^0(2))$ (values: (0.5, 0.5), (0.7, 0.3), (0.8, 0.2), (0.9, 0.1), (0.99, 0.01)), while Convex Formulation and Agnostic Estimation do not use initialization parameters $\pi^0$. The relative error results $\|\hat{\theta} - \theta^\ast\|/\sigma$ are shown for different SNR values below.
>
> | SNR | EM with $\pi^0(1)\in \{0.5, 0.7, 0.8, 0.9, 0.99\}$ | Convex Formulation | Agnostic Estimation |
> | --- | --- | --- | --- |
> | 10  | 0.0708, 0.0708, 0.0708, 0.0708, 0.0708 | 0.2141 | 2.0122 |
> | 1   | 0.0880, 0.0868, 0.0861, 0.0851, 0.0820 | 0.1751 | 0.2742 |
> | 0.1 | 0.2394, 0.2046, 0.1653, 0.1360, 0.1206 | 0.3701 | 0.1383 |
> | 0.01 | 0.2240, 0.2021, 0.1713, 0.1399, 0.1113 | 0.3736 | 0.0134 |
>
> The Convex formulation maintains moderate recovery errors that are relatively stable across all SNR values. The Agnostic formulation requires the constraint that the norm of the estimated vector equals the norm of the true signal, $\|\hat{\theta}\| = \|\theta^\ast\|$, and its performance deteriorates significantly at high SNR. Finally, the EM algorithm demonstrates consistent performance at high SNR regardless of the initial mixing weights, yet its efficacy is highly sensitive to $\pi^0$ when SNR is lower, showing improvement when $\pi^0$ is away from $(\frac{1}{2}, \frac{1}{2})$. See also additional numerical results with our anonymous code at https://anonymous.4open.science/r/code_tmlr-6984/additional_experiments.ipynb

---

> ### Author Response · Authors · 2025-11-21
> **Response to Reviewer 8cnp (part 3): Applied Examples and Theoretical Contributions**
>
> #### Problem Formulation of Haplotype Assembly
>
> We thank the reviewers for their constructive feedback. We would like to clarify and revise the problem setting of **haplotype assembly for diploids** (Cai et al., 2016) in our revised manuscript. In this model, let $\theta^\ast \in \\{-1, +1 \\}^d$ represent one **haplotype**, and the other haplotype is its negative, $-\theta^\ast$. The binary variable $z_i \in \\{-1, +1\\}$ indicates the haplotype origin of the $i$-th read, where the probability of the read originating from $\theta^\ast$ is $\mathbb{P}[z_i = +1] = \pi^\ast(1)$, and the probability of it originating from $-\theta^\ast$ is $\mathbb{P}[z_i = -1] = \pi^\ast(2)$. Furthermore, the $j$-th entry, $\varepsilon_i[j]$ for $j \in [d]$, of the noise vector $\varepsilon_i$ follows a distribution defined by a fixed error probability $p_e$: specifically, the noise causes an error (a flip) with probability $p_e$, meaning $\mathbb{P}(\varepsilon_i[j] = -2 z_i \theta^\ast[j]) = p_e$, and the noise is zero (correct reading) with probability $1 - p_e$, meaning $\mathbb{P}(\varepsilon_i[j] = 0) = 1 - p_e$. Given this framework, the read signal $y_i$ can be modeled by the following **two-mixture model**: $y_i = z_i \theta^\ast + \varepsilon_i$. The primary goal is to estimate the unknown ground truth parameters, which are the mixing probabilities $\pi^\ast = (\pi^\ast(1), \pi^\ast(2))$ and the haplotype $\theta^\ast$, using the dataset of read signals $\\{y_i\\}_{i=1}^n$. It should be noted that while this is also a two-mixture model, it features a distinct formulation and noise structure.
>
> **Reference:**
>
> (Chen et al., 2015) Chen, Yudong, Xinyang Yi, and Constantine Caramanis. "A Convex
> Formulation for Mixed Regression with Two Components: Minimax Optimal
> Rates." *arXiv preprint arXiv:1312.7006v2* (2015).
>
> (Neykov et al., 2020) Neykov, Matey, Zhaoran Wang, and Han Liu. "Agnostic Estimation for Misspecified Phase Retrieval Models." *Journal of Machine Learning Research* 21 (2020): 1-39.
>
> (Cai et al., 2016) Cai, Changxiao, Sujay Sanghavi, and Haris Vikalo. "Structured low-rank matrix factorization for haplotype assembly." *IEEE Journal of Selected Topics in Signal Processing* 10.4 (2016): 647-657.

---

### Review · Reviewer_HWto · 2025-10-13

**Summary Of Contributions:**

This paper establishes theoretical guarantees for the Expectation–Maximization (EM) algorithm applied to two-component mixture of linear regression (2MLR) models. The main contributions are Theorem 5.1, which characterizes the convergence rate of the population-level EM algorithm, and Theorem 6.1, which provides finite-sample guarantees by quantifying the statistical accuracy as a function of sample size. Moreover, Theorem 6.1 can be linked to Theorem 5.1 to derive bounds on the iteration complexity. The paper also extends the analysis to the low–signal-to-noise ratio (SNR) regime. Technically, it derives approximate dynamic equations governing the evolution of the regression parameters and mixing weights under population-level EM updates, offering deeper insight into the algorithm’s convergence behavior.

**Audience:**

Yes

**Audience Explanation:**

Considering the broad readership of TMLR, I believe that some readers may find this work of interest. However, to be honest, the topic feels somewhat narrow, as it focuses on the theoretical properties of a very specific model applied to a particular setting with a specific learning algorithm—one among many possible alternatives.

**Broader Impact Concerns:**

I don't find any broader impact concerns regarding the submission.

**Claims And Evidence:**

Yes

**Claims Explanation:**

The paper is clearly written, with a well-defined problem setup, explicit assumptions, and carefully stated theoretical guarantees. Although I did not examine all technical details, particularly the proofs, the authors effectively convey the key ideas and the overall proof strategy throughout the paper. The proposed theoretical analysis is further supported by empirical verification on synthetic data.

**Requested Changes:**

If feasible, it would be valuable to include empirical demonstrations on real-world datasets. For example, as the authors mention in the main paper, applications such as haplotype assembly or phase retrieval could serve as compelling examples to illustrate the practical relevance of the theoretical findings. While the current synthetic experiments are informative for validating the analysis, experiments on real data would better highlight the algorithm’s strengths and potential limitations in realistic settings. Since I am not an expert on this particular algorithmic framework, it is somewhat difficult for me to fully assess its practical utility.

---

> ### Author Response · Authors · 2025-11-21
> **Response to Reviewer HWto (part 1): Applied Examples and Theoretical Contributions**
>
> #### Theoretical and Empirical Comparison with Phase Retrieval Problem
>
> We thank the reviewers for their constructive feedback. We will explicitly connect our theoretical contributions to real-world applications such as phase retrieval in our revised manuscript. We provide a theoretical and empirical comparison of our convergence rates with existing results in the phase retrieval literature.
>
> **Comparison with Theoretical Bounds for Phase Retrieval**
>
> For phase retrieval problems, several theoretical guarantees for the parameter estimate $\hat{\theta}$ have been established:
>
> 1. **Convex Formulation** (Chen et al., 2015): For $n \gtrsim d$ samples, the relative error bound is $\|\hat{\theta} - \theta^\ast\|/\sigma \lesssim \sqrt{d/n} \log^4 n + \min(\sqrt{d/n}/\eta, \left(d/n\right)^{1/4}) \log^4 n$, where $\eta=\|\theta^\ast\|/\sigma$ represents the signal-to-noise ratio, as shown on page 10 of (Chen et al., 2015).
>
> 2. **Agnostic Estimation** (Neykov et al., 2020): For $n \gtrsim d^2 \log d$ samples, the error bound of the estimate $\hat{\theta}$ satisfies $\|\hat{\theta} - \theta^\ast\|/\|\theta^\ast\| \lesssim \sqrt{(d \log d)/n}$ under the constraint $\|\hat{\theta}\| = \|\theta^\ast\|$, as established on pages 3 and 9 of (Neykov et al., 2020).
>
>
> For the **EM algorithm** when the sample size $n$ is sufficiently greater than the dimension $d$ ($n \gtrsim d$), the relative error bound for the estimated vector $\hat{\theta}$ depends on the initialization of the mixing weight $\pi^0$. With **unbalanced initialization** of mixing weights $\|\pi^0 - \frac{1}{2}\|_1 \gtrsim (d/n)^{1/4}$, the relative error is bounded by $\|\hat{\theta} - \theta^\ast\|/\sigma \lesssim \sqrt{d/n}$. Conversely, with **balanced initialization** of mixing weights $\|\pi^0 - \frac{1}{2}\|_1 \lesssim (d/n)^{1/4}$, the relative error bound is less favorable, given by $\|\hat{\theta} - \theta^\ast\|/\sigma \lesssim (d/n)^{1/4}$.
>
> **Sample Complexity**: Our results require $n \gtrsim d$ samples, matching the sample complexity requirement established in (Chen et al., 2015) and improving upon the $n \gtrsim d^2 \log d$ requirement in (Neykov et al., 2020).
>
> **Error Rates**: With balanced initialization, our rate $(d/n)^{1/4}$ matches the second term in (Chen et al., 2015), demonstrating that the EM algorithm achieves a better rate when SNR $\eta$ is sufficiently small. With unbalanced initialization, our rate $\sqrt{d/n}$ matches the leading term $\sqrt{d/n} \log^4 n$ in (Chen et al., 2015) and improves upon the $\sqrt{d \log d/n}$ rate in (Neykov et al., 2020) by removing the logarithmic factor when SNR $\eta \asymp 1$.
>
> These theoretical comparisons demonstrate that our EM-based approach achieves competitive or improved error rates compared to existing phase retrieval methods, while providing explicit characterization of the initialization-dependent convergence behavior.
>
> **Empirical Results**:
>
> The following table summarizes the recovery errors for three different formulations tested on mixed regression problems. All experiments use the same data generation parameters ($n=1000$, $d=10$) with balanced $\pi^*=(\frac{1}{2}, \frac{1}{2})$. The EM formulation varies the initial mixing weight $\pi^0=(\pi^0(1), \pi^0(2))$ (values: (0.5, 0.5), (0.7, 0.3), (0.8, 0.2), (0.9, 0.1), (0.99, 0.01)), while Convex Formulation and Agnostic Estimation do not use initialization parameters $\pi^0$. The relative error results $\|\hat{\theta} - \theta^\ast\|/\sigma$ are shown for different SNR values below.
>
> | SNR | EM with $\pi^0(1)\in \{0.5, 0.7, 0.8, 0.9, 0.99\}$ | Convex Formulation | Agnostic Estimation |
> | --- | --- | --- | --- |
> | 10  | 0.0708, 0.0708, 0.0708, 0.0708, 0.0708 | 0.2141 | 2.0122 |
> | 1   | 0.0880, 0.0868, 0.0861, 0.0851, 0.0820 | 0.1751 | 0.2742 |
> | 0.1 | 0.2394, 0.2046, 0.1653, 0.1360, 0.1206 | 0.3701 | 0.1383 |
> | 0.01 | 0.2240, 0.2021, 0.1713, 0.1399, 0.1113 | 0.3736 | 0.0134 |
>
> The Convex formulation maintains moderate recovery errors that are relatively stable across all SNR values. The Agnostic formulation requires the constraint that the norm of the estimated vector equals the norm of the true signal, $\|\hat{\theta}\| = \|\theta^\ast\|$, and its performance deteriorates significantly at high SNR. Finally, the EM algorithm demonstrates consistent performance at high SNR regardless of the initial mixing weights, yet its efficacy is highly sensitive to $\pi^0$ when SNR is lower, showing improvement when $\pi^0$ is away from $(\frac{1}{2}, \frac{1}{2})$. See also additional numerical results with our anonymous code at https://anonymous.4open.science/r/code_tmlr-6984/additional_experiments.ipynb

---

> ### Author Response · Authors · 2025-11-21
> **Response to Reviewer HWto (part 2): Applied Examples and Theoretical Contributions**
>
> #### Problem Formulation of Haplotype Assembly
>
> We thank the reviewers for their constructive feedback. We would like to clarify and revise the problem setting of **haplotype assembly for diploids** (Cai et al., 2016) in our revised manuscript. In this model, let $\theta^\ast \in \\{-1, +1 \\}^d$ represent one **haplotype**, and the other haplotype is its negative, $-\theta^\ast$. The binary variable $z_i \in \\{-1, +1\\}$ indicates the haplotype origin of the $i$-th read, where the probability of the read originating from $\theta^\ast$ is $\mathbb{P}[z_i = +1] = \pi^\ast(1)$, and the probability of it originating from $-\theta^\ast$ is $\mathbb{P}[z_i = -1] = \pi^\ast(2)$. Furthermore, the $j$-th entry, $\varepsilon_i[j]$ for $j \in [d]$, of the noise vector $\varepsilon_i$ follows a distribution defined by a fixed error probability $p_e$: specifically, the noise causes an error (a flip) with probability $p_e$, meaning $\mathbb{P}(\varepsilon_i[j] = -2 z_i \theta^\ast[j]) = p_e$, and the noise is zero (correct reading) with probability $1 - p_e$, meaning $\mathbb{P}(\varepsilon_i[j] = 0) = 1 - p_e$. Given this framework, the read signal $y_i$ can be modeled by the following **two-mixture model**: $y_i = z_i \theta^\ast + \varepsilon_i$. The primary goal is to estimate the unknown ground truth parameters, which are the mixing probabilities $\pi^\ast = (\pi^\ast(1), \pi^\ast(2))$ and the haplotype $\theta^\ast$, using the dataset of read signals $\\{y_i\\}_{i=1}^n$. It should be noted that while this is also a two-mixture model, it features a distinct formulation and noise structure.
>
> **Reference:**
>
> (Chen et al., 2015) Chen, Yudong, Xinyang Yi, and Constantine Caramanis. "A Convex
> Formulation for Mixed Regression with Two Components: Minimax Optimal
> Rates." *arXiv preprint arXiv:1312.7006v2* (2015).
>
> (Neykov et al., 2020) Neykov, Matey, Zhaoran Wang, and Han Liu. "Agnostic Estimation for Misspecified Phase Retrieval Models." *Journal of Machine Learning Research* 21 (2020): 1-39.
>
> (Cai et al., 2016) Cai, Changxiao, Sujay Sanghavi, and Haris Vikalo. "Structured low-rank matrix factorization for haplotype assembly." *IEEE Journal of Selected Topics in Signal Processing* 10.4 (2016): 647-657.

---

### Review · Reviewer_wKF4 · 2025-11-06

**Summary Of Contributions:**

The paper presents a theoretical analysis of the expectation–maximization (EM) algorithm for mixed linear regression when the fitted model has two components but the true data-generating process is single-component. It establishes, for instance, convergence rate results for both balanced and unbalanced cases and supports them with numerical experiments.

*Remark*: I'm sorry about the delay. To avoid holding up the process further, I'll share my feedback without having checked all derivations in detail.

**Audience:**

Yes

**Audience Explanation:**

The work is specialized and primarily of interest to researchers studying the theoretical underpinnings of EM-type algorithms or mixture models. While the scope is narrow, the results contribute to understanding failure modes and convergence guarantees in over-specified mixture models. Including a clearer discussion of the relevance and broader implications would increase accessibility to a wider segment of the audience. Clarifying the connections to broader themes such as overparameterization or robustness in iterative algorithms would further enhance its appeal to the TMLR audience.

**Claims And Evidence:**

Yes

**Claims Explanation:**

The theoretical claims are soundly developed and presented with mathematical rigor. The exposition is coherent, and the included numerical experiments are well structured and consistent with the stated results, providing credible support for the main theoretical arguments. The overview of related work is thorough and situates the contribution clearly within the existing literature.

However, the presentation is dense, and several results would benefit from additional commentary explaining their intuition, scope, and implications. The paper currently provides limited discussion of how the theoretical insights relate to practical applications; the two motivating examples are illustrative but not empirically explored.

**Requested Changes:**

- Add brief intuitive explanations after each main theorem to clarify its significance and relation to practical algorithmic behavior.
- Include at least one applied or simulated example demonstrating how the derived convergence rates manifest in a more realistic setting, or explicitly clarify that the work is purely theoretical and adjust claims accordingly.

Optional suggestions:
- In eq. (1): for notational simplicity, consider defining $z\in \{-1, 1\}$ instead of using $(-1)^{z+1}$.
- Expand the discussion section to connect findings to current trends in overparameterized models and algorithmic stability.

---

> ### Author Response · Authors · 2025-11-21
> **Response to Reviewer wKF4 (part 1)**
>
> We thank the reviewer for their thorough review and valuable suggestions for improving the accessibility and broader impact of our work.
>
> **Intuitive Explanations After Theorems**
>
> **Main Theorem 5.1: EM Convergence Rate**
>
> For GMM/MLR models, running the EM algorithm is equivalent to performing a step of **Gradient Descent** on the negative log-likelihood (Kwon et al., 2024).
>
> When an **unbalanced** estimate of mixing weights is given, the negative log-likelihood retains its dominant **quadratic** term, $[\alpha]^2$. This quadratic form ensures that the EM algorithm behaves like a Gradient Descent step on a strongly convex function, suggesting a fast, **linear** convergence rate for $\alpha^t$.
>
> In contrast, when a **balanced** estimate of mixing weights is given, the $[\alpha]^2$ term in the negative population log-likelihood cancels out (page 22 of (Luo and Hashemi, 2024) and Lemma A.3). Specifically, the likelihood term can be expanded as $[\alpha]^2/2 - \mathbb{E}[\ln \cosh(\alpha X)] = [\alpha]^2/2 - \mathbb{E}[(\alpha X)^2/2 - (\alpha X)^4/12 +\ldots]\approx 3[\alpha]^4/4$. The EM algorithm, being equivalent to Gradient Descent, then follows a path approximated by $\alpha^{t+1} \approx \alpha^t - \nabla (3[\alpha^t]^4/4) = \alpha^t - 3[\alpha^t]^3$, leading to a much slower, **sublinear** convergence rate where $\alpha^t$ is proportional to $1/\sqrt{t}$.
>
> **Main Theorem 6.1: MLE Estimation Accuracy**
>
> The ultimate statistical accuracy of the **MLE estimate** is governed by the invertibility of the **Fisher Information** matrix, which is the second-order derivative of the negative population log-likelihood.
>
> With an **unbalanced** estimate of mixing weights, the negative log-likelihood maintains the $[\alpha]^2= ||\theta||^2/\sigma^2$ term, which ensures the **Fisher Information** matrix (w.r.t. the regression parameters) is **invertible**. As a well-established result (Van der Vaart, 2000), this invertibility guarantees that the MLE estimate achieves the standard parametric rate of order $n^{-1/2}$, and therefore the final accuracy of the EM algorithm is also proportional to $n^{-1/2}$.
>
> However, when a **balanced** estimate is given, the critical term $[\alpha]^2 $ vanishes, causing the **Fisher Information** matrix to be **singular**. This singularity slows down the standard parametric rate of final accuracy (Dwivedi et al., 2020b), resulting in a converged $\alpha^T$ that exhibits a rate proportional to $n^{-1/4}$.
>
> Reference:
>
> (Kwon et al., 2024) Kwon, Jeongyeol, et al. "Global optimality of the em algorithm for mixtures of two-component linear regressions." *IEEE Transactions on Information Theory* (2024).
>
> (Van der Vaart, 2000) Van der Vaart, Aad W. Asymptotic statistics. Vol. 3. Cambridge university press, 2000.
>
> (Dwivedi et al., 2020b) Dwivedi, Raaz, et al. "Singularity, misspecification and the convergence rate of EM." *The Annals of Statistics* 48.6 (2020): 3161-3182.
>
> **Robustness in Iterative Algorithms**
>
> We thank the reviewer for this insightful question regarding the robustness and stability of the iterative algorithm. We will certainly add a discussion on the robustness and stability of iterative algorithms in our revised manuscript. First, we demonstrate that the EM update rules are robust to perturbations from previous iterations by establishing the Lipschitz continuity of the updates (starting from Eq. 11 and utilizing Lemmas H.3–H.4 in Appendix H). In particular, for balanced initial weights, we can quantify the variation in the $t$-th iteration resulting from the perturbation of the initialization $\alpha^0$ via Proposition 5.3. Furthermore, it is worth noting that running the EM algorithm for GMM/MLR models is equivalent to performing gradient descent on the population negative log-likelihood with respect to the regression parameters and the mixing weight imbalance. This crucial equivalence is explicitly established in Eq. 11 of (Xu et al., 1996), Lemma 2 of (Kwon et al., 2024), and the derivations in Appendix B of (Luo and Hashemi, 2024).
>
> Reference:
>
> (Xu et al., 1996) Xu, Lei, and Michael I. Jordan. "On convergence properties of the EM algorithm for Gaussian mixtures." *Neural computation* 8.1 (1996): 129-151.
>
> (Kwon et al., 2024) Kwon, Jeongyeol, et al. "Global optimality of the em algorithm for mixtures of two-component linear regressions." *IEEE Transactions on Information Theory* (2024).
>
> (Luo and Hashemi, 2024) Luo, Zhankun, and Abolfazl Hashemi. "Unveiling the Cycloid Trajectory of EM Iterations in Mixed Linear Regression." *International Conference on Machine Learning*. PMLR, 2024.

---

> ### Author Response · Authors · 2025-11-21
> **Response to Reviewer wKF4 (part 2)**
>
> **Connection to Overparameterization**
>
> We thank the reviewer for this valuable suggestion. In our theoretical results, we exhibit linear convergence with an unbalanced guess of mixing weights, while showing sublinear convergence ($\alpha^t \asymp 1/\sqrt{t}$) with a balanced guess of mixing weights.
>
> Interestingly, for the problem of **low-rank matrix factorization** (Xiong et al., 2023) in the overparameterization regime, an exponentially faster linear convergence rate is achieved using gradient descent with an **asymmetric** parameterization, while gradient descent with symmetric parameterization exhibits a sublinear rate of $1/t^2$. This suggests that **imbalance** accelerates the convergence rate.
>
> Overparameterization also has an impact on the convergence rate of gradient descent for **learning a single neuron** in neural networks (Xu et al., 2023). The method exhibits linear convergence in the exact-parameterization regime, but shows a sublinear convergence rate of $1/t^3$ in the overparameterization regime. Therefore, overparameterization can **exponentially slow down** the convergence rate of Gradient Descent (GD).
>
> While the EM algorithm has been shown to be a powerful tool for learning MoE models (Fruytier et al., 2025), establishing the convergence rates of the Maximum Likelihood Estimator (MLE) for these complex mixture models under exact-specified and over-specified settings remains a significant open challenge (Ho et al., 2022; Nguyen et al., 2023; Nguyen et al., 2024).
>
> In this context, our theoretical analysis of the EM algorithm for the 2-component Mixed Linear Regression (2MLR) model serves as a **fundamental example**. By establishing rigorous guarantees in this setting, our work provides the necessary theoretical groundwork to deepen the understanding of convergence behaviors in more complex architectures, such as MoE and deep mixture models.
>
> Reference:
>
> (Xu et al., 2023) Xu, Weihang, and Simon Du. "Over-parameterization exponentially slows down gradient descent for learning a single neuron." The Thirty Sixth Annual Conference on Learning Theory. PMLR, 2023.
>
> (Xiong et al., 2023) Xiong, Nuoya, Lijun Ding, and Simon S. Du. "How over-parameterization slows down gradient descent in matrix sensing: The curses of symmetry and initialization." arXiv preprint arXiv:2310.01769 (2023).
>
> (Quentin et al., 2025) Fruytier, Quentin, Aryan Mokhtari, and Sujay Sanghavi. "Learning Mixtures of Experts with EM: A Mirror Descent Perspective." *International Conference on Machine Learning*. PMLR, 2025.
>
> (Nhat et al., 2022) Ho, Nhat, Chiao-Yu Yang, and Michael I. Jordan. "Convergence rates for Gaussian mixtures of experts." *Journal of Machine Learning Research* 23.323 (2022): 1-81.
>
> (Nguyen et al., 2023) Nguyen, Huy, TrungTin Nguyen, and Nhat Ho. "Demystifying softmax gating function in Gaussian mixture of experts." *Advances in Neural Information Processing Systems* 36 (2023): 4624-4652.
>
> (Nguyen et al., 2024) Nguyen, Huy, et al. "Statistical Perspective of Top-K Sparse Softmax Gating Mixture of Experts." *International Conference on Learning Representations*. 2024.

---

> ### Author Response · Authors · 2025-11-21
> **Response to Reviewer wKF4 (part 3)**
>
> Theoretical and Empirical Comparison with Phase Retrieval Problem
>
> We thank the reviewers for their constructive feedback. We will explicitly connect our theoretical contributions to real-world applications such as phase retrieval in our revised manuscript. We provide a theoretical and empirical comparison of our convergence rates with existing results in the phase retrieval literature.
>
> **Comparison with Theoretical Bounds for Phase Retrieval**
>
> For phase retrieval problems, several theoretical guarantees for the parameter estimate $\hat{\theta}$ have been established:
>
> 1. **Convex Formulation** (Chen et al., 2015): For $n \gtrsim d$ samples, the relative error bound is $\|\hat{\theta} - \theta^\ast\|/\sigma \lesssim \sqrt{d/n} \log^4 n + \min(\sqrt{d/n}/\eta, \left(d/n\right)^{1/4}) \log^4 n$, where $\eta=\|\theta^\ast\|/\sigma$ represents the signal-to-noise ratio, as shown on page 10 of (Chen et al., 2015).
>
> 2. **Agnostic Estimation** (Neykov et al., 2020): For $n \gtrsim d^2 \log d$ samples, the error bound of the estimate $\hat{\theta}$ satisfies $\|\hat{\theta} - \theta^\ast\|/\|\theta^\ast\| \lesssim \sqrt{(d \log d)/n}$ under the constraint $\|\hat{\theta}\| = \|\theta^\ast\|$, as established on pages 3 and 9 of (Neykov et al., 2020).
>
>
> For the **EM algorithm** when the sample size $n$ is sufficiently greater than the dimension $d$ ($n \gtrsim d$), the relative error bound for the estimated vector $\hat{\theta}$ depends on the initialization of the mixing weight $\pi^0$. With **unbalanced initialization** of mixing weights $\|\pi^0 - \frac{1}{2}\|_1 \gtrsim (d/n)^{1/4}$, the relative error is bounded by $\|\hat{\theta} - \theta^\ast\|/\sigma \lesssim \sqrt{d/n}$. Conversely, with **balanced initialization** of mixing weights $\|\pi^0 - \frac{1}{2}\|_1 \lesssim (d/n)^{1/4}$, the relative error bound is less favorable, given by $\|\hat{\theta} - \theta^\ast\|/\sigma \lesssim (d/n)^{1/4}$.
>
> **Sample Complexity**: Our results require $n \gtrsim d$ samples, matching the sample complexity requirement established in (Chen et al., 2015) and improving upon the $n \gtrsim d^2 \log d$ requirement in (Neykov et al., 2020).
>
> **Error Rates**: With balanced initialization, our rate $(d/n)^{1/4}$ matches the second term in (Chen et al., 2015), demonstrating that the EM algorithm achieves a better rate when SNR $\eta$ is sufficiently small. With unbalanced initialization, our rate $\sqrt{d/n}$ matches the leading term $\sqrt{d/n} \log^4 n$ in (Chen et al., 2015) and improves upon the $\sqrt{d \log d/n}$ rate in (Neykov et al., 2020) by removing the logarithmic factor when SNR $\eta \asymp 1$.
>
> These theoretical comparisons demonstrate that our EM-based approach achieves competitive or improved error rates compared to existing phase retrieval methods, while providing explicit characterization of the initialization-dependent convergence behavior.
>
> **Empirical Results**:
>
> The following table summarizes the recovery errors for three different formulations tested on mixed regression problems. All experiments use the same data generation parameters ($n=1000$, $d=10$) with balanced $\pi^*=(\frac{1}{2}, \frac{1}{2})$. The EM formulation varies the initial mixing weight $\pi^0=(\pi^0(1), \pi^0(2))$ (values: (0.5, 0.5), (0.7, 0.3), (0.8, 0.2), (0.9, 0.1), (0.99, 0.01)), while Convex Formulation and Agnostic Estimation do not use initialization parameters $\pi^0$. The relative error results $\|\hat{\theta} - \theta^\ast\|/\sigma$ are shown for different SNR values below.
>
> | SNR | EM with $\pi^0(1)\in \{0.5, 0.7, 0.8, 0.9, 0.99\}$ | Convex Formulation | Agnostic Estimation |
> | --- | --- | --- | --- |
> | 10  | 0.0708, 0.0708, 0.0708, 0.0708, 0.0708 | 0.2141 | 2.0122 |
> | 1   | 0.0880, 0.0868, 0.0861, 0.0851, 0.0820 | 0.1751 | 0.2742 |
> | 0.1 | 0.2394, 0.2046, 0.1653, 0.1360, 0.1206 | 0.3701 | 0.1383 |
> | 0.01 | 0.2240, 0.2021, 0.1713, 0.1399, 0.1113 | 0.3736 | 0.0134 |
>
> The Convex formulation maintains moderate recovery errors that are relatively stable across all SNR values. The Agnostic formulation requires the constraint that the norm of the estimated vector equals the norm of the true signal, $\|\hat{\theta}\| = \|\theta^\ast\|$, and its performance deteriorates significantly at high SNR. Finally, the EM algorithm demonstrates consistent performance at high SNR regardless of the initial mixing weights, yet its efficacy is highly sensitive to $\pi^0$ when SNR is lower, showing improvement when $\pi^0$ is away from $(\frac{1}{2}, \frac{1}{2})$. See also additional numerical results with our anonymous code at https://anonymous.4open.science/r/code_tmlr-6984/additional_experiments.ipynb

---

> ### Author Response · Authors · 2025-11-21
> **Response to Reviewer wKF4 (part 4)**
>
> #### Problem Formulation of Haplotype Assembly
>
> We thank the reviewers for their constructive feedback. We would like to clarify and revise the problem setting of **haplotype assembly for diploids** (Cai et al., 2016) in our revised manuscript. In this model, let $\theta^\ast \in \\{-1, +1 \\}^d$ represent one **haplotype**, and the other haplotype is its negative, $-\theta^\ast$. The binary variable $z_i \in \\{-1, +1\\}$ indicates the haplotype origin of the $i$-th read, where the probability of the read originating from $\theta^\ast$ is $\mathbb{P}[z_i = +1] = \pi^\ast(1)$, and the probability of it originating from $-\theta^\ast$ is $\mathbb{P}[z_i = -1] = \pi^\ast(2)$. Furthermore, the $j$-th entry, $\varepsilon_i[j]$ for $j \in [d]$, of the noise vector $\varepsilon_i$ follows a distribution defined by a fixed error probability $p_e$: specifically, the noise causes an error (a flip) with probability $p_e$, meaning $\mathbb{P}(\varepsilon_i[j] = -2 z_i \theta^\ast[j]) = p_e$, and the noise is zero (correct reading) with probability $1 - p_e$, meaning $\mathbb{P}(\varepsilon_i[j] = 0) = 1 - p_e$. Given this framework, the read signal $y_i$ can be modeled by the following **two-mixture model**: $y_i = z_i \theta^\ast + \varepsilon_i$. The primary goal is to estimate the unknown ground truth parameters, which are the mixing probabilities $\pi^\ast = (\pi^\ast(1), \pi^\ast(2))$ and the haplotype $\theta^\ast$, using the dataset of read signals $\\{y_i\\}_{i=1}^n$. It should be noted that while this is also a two-mixture model, it features a distinct formulation and noise structure.
>
> **Reference:**
>
> (Chen et al., 2015) Chen, Yudong, Xinyang Yi, and Constantine Caramanis. "A Convex
> Formulation for Mixed Regression with Two Components: Minimax Optimal
> Rates." *arXiv preprint arXiv:1312.7006v2* (2015).
>
> (Neykov et al., 2020) Neykov, Matey, Zhaoran Wang, and Han Liu. "Agnostic Estimation for Misspecified Phase Retrieval Models." *Journal of Machine Learning Research* 21 (2020): 1-39.
>
> (Cai et al., 2016) Cai, Changxiao, Sujay Sanghavi, and Haris Vikalo. "Structured low-rank matrix factorization for haplotype assembly." *IEEE Journal of Selected Topics in Signal Processing* 10.4 (2016): 647-657.

---

### Decision · Action_Editor_Kjw9 · 2025-12-12

**Recommendation:** Accept with minor revision

**Audience:**

Yes

**Audience Explanation:**

Statistics/theory community will find this paper interesting.

Reviewers have pointed out that the scope of the paper is "narrow" or the area being "niche". It will really be good if the authors can revise their introduction section to give some broader context of the problem, and perhaps include some more motivation and/or applications.

**Claims And Evidence:**

Yes

**Claims Explanation:**

This paper provides a careful analysis of the convergence rate of the expectation maximization algorithm for 2 component high dimensional mixed linear regression. The reviewers all agree that the paper is theoretically solid with well-specified theorems and clearly written proofs. The proofs seem to be correct.